# Exploring the Optimal Choice for Generative Processes in Diffusion Models: Ordinary vs Stochastic Differential Equations

**Yu Cao** [*]          **Jingrun Chen** [†]          **Yixin Luo** [‡]          **Xiang Zhou** [§]

## Abstract

The diffusion model has shown remarkable success in computer vision, but it remains unclear whether the ODE-based probability flow or the SDE-based diffusion model is more superior and under what circumstances. Comparing the two is challenging due to dependencies on data distributions, score training, and other numerical issues. In this paper, we study the problem mathematically for two limiting scenarios: the zero diffusion (ODE) case and the large diffusion case. We first introduce a pulse-shape error to perturb the score function and analyze error accumulation of sampling quality, followed by a thorough analysis for generalization to *arbitrary error*. Our findings indicate that when the perturbation occurs at the end of the generative process, the ODE model outperforms the SDE model with a large diffusion coefficient. However, when the perturbation occurs earlier, the SDE model outperforms the ODE model, and we demonstrate that the error of sample generation due to such a pulse-shape perturbation is exponentially suppressed as the diffusion term's magnitude increases to infinity. Numerical validation of this phenomenon is provided using Gaussian, Gaussian mixture, and Swiss roll distribution, as well as realistic datasets like MNIST and CIFAR-10.

## 1  Introduction

Diffusion models have achieved remarkable success in various artificial intelligence context generation tasks, particularly in computer vision [13]. This technique is rapidly evolving with industrial-level products like DALL·E series. The diffusion model was first proposed and studied by Sohl-Dickstein et al. [26] in 2015. Later, Song and Ermon [29] proposed score matching with Langevin dynamics (SMLD) and Ho et al. [16] further explored the Denoising Diffusion Probabilistic Models (DDPM). Both formalisms can be interpreted as time-discretization of stochastic differential equations (SDEs) [30]. Since the publication of these seminal works, many techniques have been proposed to improve the efficiency and accuracy of diffusion models, such as DDIM [28], Analytic-DPM [4], gDDIM [36], EDM [19], and consistency model [31], among others.

The score-based diffusion model involves two steps [30, 17]. Firstly, one estimates the score function, which is the gradient of the logarithm of the probability density function, in the form of a neural network. This step uses trajectories of an Ornstein-Uhlenbeck (OU) process starting with given data

---

[*]Institute of Natural Sciences and School of Mathematical Sciences, Shanghai Jiao Tong University, Shanghai 200240, China, `yucao@sjtu.edu.cn`

[†]University of Science and Technology of China, Hefei 230026, China; Suzhou Institute of Advanced Research, University of Science and Technology of China, Suzhou 215123, China, `jingrunchen@ustc.edu.cn`

[‡]University of Science and Technology of China, Hefei 230026, China; Suzhou Institute of Advanced Research, University of Science and Technology of China, Suzhou 215123, China, `seeing@mail.ustc.edu.cn`

[§]School of Data Science and Department of Mathematics, City University of Hong Kong, Kowloon, Hong Kong SAR, `xizhou@cityu.edu.hk`

37th Conference on Neural Information Processing Systems (NeurIPS 2023).

samples. This process of injecting noise into structured data is usually referred to as the *inference process*. Secondly, new samples are generated by simulating a time-reversed SDE, with a drift term depending on the learned score function from the first step. This is known as the *generative process*.

In general, there are two diffusion coefficients $g$ in the inference process and $h$ in the generative process; see § 2. Regardless of the choice of $h$ and $g$, it is always possible to design an SDE in the generative process that matches the forward inference process in the weak sense, i.e., the probability density functions match for both processes. We highlight that this function $h$ (unlike $g$) does not only appear in the diffusion term, but also enters the drift term in the generative SDE. The choice of $g$ is equivalent to time re-scaling (see Appx. B.2), while the choice of $h$ is an important topic in practice. Two common choices of $h$ are Probability Flow $h = 0$ [9, 32], which refers to as an ODE, and an SDE-based diffusion model with $h = g$ [16, 29, 30]. When the score training is accurate, the choice of this function $h$ does not affect the sample generation quality in the continuous-time setting.

In practice, numerical error is inevitable during training the score function. Recent theoretical works [10, 7] have shown that the sample generation quality are affected by three aspects: (1) the truncation of simulation time to a finite $T$; (2) the inexact score function; (3) the time-discretization error. The first error is not significantly since the forward OU process converges to the equilibrium Gaussian measure exponentially fast in $T$. The third error can be reduced systematically by more efficient numerical schemes [21], such as exponential integrator proposed in [35, 36]. The inexact training of score function has a few important but subtle consequences. Recent works [10, 7] analyzed the convergence rate of diffusion models, provided that the score training error is sufficient small. However, once the score training is not accurate, the nice equivalence of the generated distribution free of the generative diffusion coefficient $h$ no longer holds as in the idealized situation of exact score function. This raises a key question of our interest about how the choice of $h$ can affect the sample quality in the face of the inexact score training error. Qualitatively, there are two distinctive cases: $h = 0$ or $h$ is large. An important question to ask is: **in the presence of non-negligible score training errors, which $h$ will produce better sampling quality?** Is it the probability flow ($h = 0$) or the SDE? More quantitatively, what magnitude of $h$ is optimal?

**Related works** The impact of $h$ on the generative process seems not yet fully investigated in recent literature, as most experiments used the default choice of this parameter. However, some authors have reported related empirical observations. For example, Song et al. [30] empirically observed that the choice of $h = g$ produces better sample generation quality than the ODE case ($h = 0$) with real datasets. On the other hand, Denoising Diffusion Implicit Models (DDIM) in [28] includes both deterministic and stochastic samplers and points out that the probability flow ($h = 0$) can produce better samples with improved numerical schemes for the generative process. [36] generalized the DDIM and tried to explain the advantages of a deterministic sampling scheme over the stochastic one for fast sampling. Moreover, Karras et. al., [19] had empirically searched for optimal coefficients which had shown to bring practical advantages. None of these empirical results delivered comprehensive investigations on the influence of the diffusion coefficient, and a consistent and affirmative answer to our question still awaits. Recently, there has been rapid progress in theoretical works on error analysis for diffusion models, as seen in [10, 7] and references therein. However, these analyses usually assume specific settings of $h$, such as $h = g$. Furthermore, it seems that directly analyzing upper bounds based on these error estimations cannot provide adequate information about choosing the optimal $h$; see Appx. B.4. Albergo et al. proposed a unified framework known as stochastic interpolants and slightly discussed the optimal choice between the probability flow and diffusion models [1, Sec. 2.4]. It is interesting to see how our theoretical analysis below can generalize to their promising unified settings [1].

**Our approach** To investigate the effect of the diffusion coefficient $h$ on sampling quality, we adopt the continuous-time framework, which precludes time discretization errors. We measure sample quality by the KL divergence between the data distribution $p_0$ and the distribution of the generative SDE at the terminal time $T$. Given the assumption that the score function carries numerical errors, we consider $h$ as a controller and aim to minimize the KL divergence with respect to $h$. While the optimization problem is straightforward to set up, it is challenging to draw valuable theoretical insights in a general setting of approximate score functions. Therefore, we choose the asymptotic approach, assuming that the error from the training score is reasonably small with a magnitude of $\epsilon$. Under this assumption, the leading-order term of the KL divergence takes the form

$$\text{error of sample generation in KL divergence} = L(h)\,\epsilon^2 + \mathcal{O}(\epsilon^3).$$

This $\epsilon^2$ order is known in [7, 10], but the dependence of this Gateaux differential $L(h)$ on $h$ and other factors has yet to be understood at all. Our contribution is to analyze how $L(h)$ behaves as $h$ varies; in particular, by considering the constant $h$ in two limiting situations: $h = 0$ and $h \gg 1$.

**Main Contributions**  We summarize main contributions below:

- We prove that when the error in score function approximation is a time-localized function only at the beginning of the inference step (i.e., at the end of the generative process), the ODE case ($h = 0$) outperforms the SDE case ($h \to \infty$); see Prop. 3.5. If this (time-localized) error occurs in the middle, then the SDE case has an *exponentially smaller error* than the ODE case ($h = 0$), as $h \to \infty$ (see Prop. 3.4). See Appx. E.4 for reasons behind the time-localized choice.

- For a **general** score training error, we prove that as $h \to \infty$, the leading-order term $L(h)$ above converges exponentially fast to a constant, which only depends on the distribution $p_0$ and the score training error at the end of the generative process; see Prop. 3.6. The conclusion about the optimal $h$ depends on how the score training error is distributed over the time horizon $[0, T]$.

Numerically, we validate the above phenomenon for 1D Gaussian, 2D Gaussian mixture, and Swiss roll distribution, as well as realistic datasets like MNIST and CIFAR-10. Due to the tight connection between the distribution of score training error and $h$, our results may suggest backwardly modifying loss functions during training to adapt to a particular diffusion coefficient of interest. This is a topic of independent interest and we report some preliminary experiments in Appx. I to validate potential applications of our theoretical analysis. A comprehensive investigation will be left as future works.

**Notation convention**  The time duration $T > 0$ is a fixed parameter. For any time-dependent function $(t, x) \mapsto f_t(x)$, where $x \in \mathbb{R}^d$ and $t \in [0, T]$, we denote $f_t^{\leftarrow}(x) \equiv f_{T-t}(x)$. Sometimes we directly use the "arrowed" variables $f_t^{\leftarrow}(x)$ to highlight the direction of time is from reference noise to the data distribution (i.e., the generative direction) even without referring to $f$ first. The notation $f \lesssim g$ means that $f \leq cg$ where $c \to 1$ in a certain limit, i.e., $\limsup f/g \leq 1$; $f \sim g$ means $\lim(f/g) = 1$. The asymptotic parameter will be explained below explicitly. When two matrices $A \preceq B$, it means $B - A$ is positive semidefinite. $\boldsymbol{I}_d$ is the $d$-dimensional identity matrix; $\mathbb{I}_A$ means an indicator function of a set $A$; Id is the identity operator. For a random variable $X$, law$(X)$ means the distribution of $X$. Some important quantities are summarized in Appx. A.

## 2  Background

**Score-based generative models**  Suppose we have a collection of data from an unknown distribution with density $p_0$, we can inject noise into data via the following SDEs:

$$\mathrm{d}X_t = f_t(X_t)\,\mathrm{d}t + g_t\,\mathrm{d}W_t, \qquad \mathrm{law}(X_0) = p_0, \tag{1}$$

where the drift coefficient $f_{(\cdot)}(\cdot) : \mathbb{R}^d \times \mathbb{R} \to \mathbb{R}^d$ is a time-dependent vector field, and the diffusion coefficient $g_{(\cdot)} : \mathbb{R} \to \mathbb{R}$ is a scalar-valued function (for simplicity). A widely used example is variance-preserving SDE (VP-SDE) with $f_t(x) = -1/2\, g_t^2\, x$ and $g > 0$ is typically chosen as a non-decreasing function in literature [30]. Without loss of generality, we can assume $g_t = 1$ since for any non-zero $g$, its effect is simply to re-scale the time; see Appx. B.2.

Denote the probability density of $X_t$ as $p_t$ and the score function is defined as $\nabla \log p_t$. One main innovation in diffusion models is to find a "backward" SDE $Y_t$ such that $Y_t$ drives the state with distribution $p_T$ back to $p_0$. We adopt the arrow of time in this backward direction now and write $Y_t$ as

$$\mathrm{d}Y_t = A_t^{\leftarrow}(Y_t)\,\mathrm{d}t + h_t^{\leftarrow}\,\mathrm{d}W_t, \qquad \mathrm{law}(Y_0) = p_T, \tag{2}$$

where $h_t^{\leftarrow}$ is an arbitrary real-valued function of time. The distribution of $Y_t$ is denoted as $q_t$. Provided that the score function is available, we can select $A^{\leftarrow}$ such that $q_t$ is the same as $p_{T-t}$, in particular, $q_T = p_0$. It is not hard to derive that we can ensure $q_t \equiv p_{T-t}$ if we choose

$$A_t^{\leftarrow}(x) = -f_t^{\leftarrow}(x) + \frac{(g_t^{\leftarrow})^2 + (h_t^{\leftarrow})^2}{2} \nabla \log p_t^{\leftarrow}(x). \tag{3}$$

A self-contained proof is provided in Appx. B.1. When $h^{\leftarrow} = g^{\leftarrow}$, it refers to the backward SDE used in [30]; when $h^{\leftarrow} = 0$, it refers to the probability flow therein. More general interpolation of diffusion and flow can be found in, e.g., [1].

**Training of score functions**    The above score function $(t, x) \mapsto \nabla \log p_t^{\leftarrow}(x)$ needs to be trained from data. Denoising score matching [33] refers to the following score-matching loss (SML) function to train the score whose parameterized architecture is denoted as $\mathfrak{S}$:

$$\min_{\mathfrak{S}} \int_0^T \omega_t \, \mathbb{E}_{X_0 \sim p_0} \mathbb{E}_{X_t \sim p_{t|0}(\cdot|X_0)} \Big[ \big\| \mathfrak{S}_t(X_t) - \nabla \log p_{t|0}(X_t|X_0) \big\|^2 \Big] \mathrm{d}t, \tag{4}$$

where $p_{t|0}(x_t|x_0)$ is the transition probability of the state $x_0$ at time $0$ towards the state $x_t$ at time $t$ for the forward process (1). The function $\omega_t \geq 0$ is a weight function. The default choice in many literature is that $g_t = \sqrt{\beta_0 + (\beta_1 - \beta_0)t}$, $0 < \beta_0 < \beta_1$ are parameters, and one chooses the weight function as follows:

$$\textbf{Default weight:} \qquad \omega_t = \varpi_t^2, \qquad \varpi_t = \sqrt{1 - e^{-\frac{1}{2}t^2(\beta_1 - \beta_0) - t\beta_0}}. \tag{5}$$

The quantity $\varpi_t$ has the meaning as the standard deviation of $X_t$ conditioned on a fixed $X_0$ in the forward process. See § 3.7 and Appx. I for more weight functions.

**Source of errors**    There is usually intrinsic error due to an inexact score function. It is not negligible in many scenarios, e.g., there is only a finite amount of samples of $p_0$ available or only a small neural network architecture is achievable. However, it is reasonable to assume that this non-negligible error is reasonably small, and we decompose the trained score function $\mathfrak{S}_t^{\leftarrow}$ into

$$\mathfrak{S}_t^{\leftarrow}(x) = \nabla \log p_t^{\leftarrow}(x) + \epsilon \mathscr{E}_t^{\leftarrow}(x), \tag{6}$$

where $\epsilon$ is small, $\mathscr{E}_t^{\leftarrow}$ is assumed to be $\mathcal{O}(1)$ and the total error is $\epsilon \mathscr{E}_t^{\leftarrow}$. The generative process used in practice has to use $\mathrm{law}(Y_0) = \mathcal{N}(0, I_d)$ instead since $p_T$ is intractable in (2). By choosing $T$ large enough so that $p_T \approx \mathcal{N}(0, I_d)$, we can neglect this error due to the finite $T$. Besides, we also need some numerical schemes to simulate this generative SDE, which also leads into discretization errors. **In summary**, there are three sources of errors (1) $p_T \neq \mathcal{N}(0, I_d)$: this is the source of errors in the initial distribution of the generative process; (2) $\mathscr{E}^{\leftarrow} \neq 0$: error from imperfect score function from training; (3) numerical discretization of the generative process. The third error can be systemically eliminated by choosing a high-order scheme [21] or an extremely small time step. It has been observed that by choosing a more accurate numerical scheme, e.g., exponential integrator, one can reduce the computational costs [35, 36]. As for the first error, if one chooses the OU process for (1), $p_T$ converges to $\mathcal{N}(0, I_d)$ exponentially fast and thus $T = \mathcal{O}(\ln(\delta))$ where $\delta$ is the error between $p_0$ and the distribution of generated samples. Therefore, the choice of $T$ is, in practice, not hard to manage. More details about these three error sources can be found in, e.g., [7, 10] or Appx. B.3.

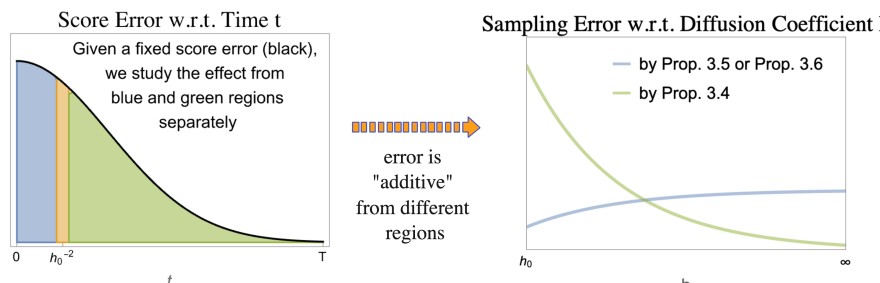

Figure 1: *Schematic illustration* of the main message: the distribution of the score error $\mathscr{E}_t^{\leftarrow}$ w.r.t. time also matters, in addition to the score-matching loss (4). Asymptotically, the score error can be viewed as "additive" and the error from two time regions (blue and green) might decay or magnify as the magnitude of diffusion coefficient $h^{\leftarrow}$ increases (see the right picture). The yellow region is a transitional region whether the effect of h is not easy to decide.

## 3    Asymptotic analysis of terminal errors

We use the KL divergence between the data distribution $p_0$ and the distribution of generated samples to quantify the performance of generative model, which depends on the error of score function $\epsilon \mathscr{E}^{\leftarrow}$,

the diffusion coefficient $h^{\leftarrow}$, and data distribution $p_0$. To extract the main feature, we first let $\epsilon \to 0$ and estimate

$$\text{sample generation error in KL divergence} = L(h^{\leftarrow}, \mathscr{E}^{\leftarrow}, p_0)\epsilon^2 + \mathcal{O}(\epsilon^3).$$

Whenever $p_0$ and $\mathscr{E}^{\leftarrow}$ are obvious from the context, we simply write $L(h^{\leftarrow}) \equiv L(h^{\leftarrow}, \mathscr{E}^{\leftarrow}, p_0)$. Next, we formulate the main problem setup and assumptions in § 3.1. The expression of $L$ is shown in Prop. 3.2. Then we let $h_t^{\leftarrow} \equiv h$ be independent of time, and study how the leading order function $L$ depends on h in various settings of the error function $\mathscr{E}_t^{\leftarrow}$. Firstly, we consider $\mathscr{E}_t^{\leftarrow}(x) = \delta_{t-s}E(x)$ as a time-localized function and two limiting scenarios: $h^{\leftarrow} = $ h where h $= 0$ (ODE case) and h $\to \infty$ (SDE case with large diffusion). When $\mathscr{E}_t^{\leftarrow}(x) = \delta_{t-s}E(x)$ is a time-localized function at the end of the generative process (i.e., $s$ is close to $T$), the ODE case will outperform the SDE case (see Prop. 3.5); otherwise, the SDE case has an exponentially smaller error than the ODE case as h $\to \infty$ (see Prop. 3.4). Secondly, by combing Prop. 3.4 and Prop. 3.5, the tail behavior of h $\mapsto L($h$)$ for a general $\mathscr{E}^{\leftarrow}$ is described in Prop. 3.6. The reasons behind considering this pulse-shape $\mathscr{E}_t^{\leftarrow}$ will be discussed in Appx. E.4.

## 3.1 Set-up and assumptions

We fix the time $T$ and consider the following SDE for the generative process in $t \in [0, T]$,

$$\mathrm{d}\widetilde{Y}_t = \left( -f_t^{\leftarrow}(\widetilde{Y}_t) + \frac{(g_t^{\leftarrow})^2 + (h_t^{\leftarrow})^2}{2}\mathfrak{S}_t^{\leftarrow}(\widetilde{Y}_t) \right) \mathrm{d}t + h_t^{\leftarrow} \, \mathrm{d}W_t, \ \text{law}(\widetilde{Y}_0) = p_T, \qquad (7)$$

which can be regarded as a perturbed equation (2) of $Y_t$. Denote the distribution of $\widetilde{Y}_t$ as $\widetilde{q}_t$. Note that $\widetilde{q}_T$ depends on both $h^{\leftarrow}$ and $\epsilon\mathscr{E}_t^{\leftarrow}$ (hidden inside $\mathfrak{S}_t^{\leftarrow} \equiv \nabla \log p_t^{\leftarrow} + \epsilon\mathscr{E}_t^{\leftarrow}$); however, when $\epsilon = 0$, by (3), $\widetilde{q}_T \equiv q_T \equiv p_0$ for *any* $h^{\leftarrow}$. We quantify the sample generation quality via

$$\text{KL}(p_0||\widetilde{q}_T) = \int p_0 \log(p_0/\widetilde{q}_T).$$

Due to the presence of $\epsilon\mathscr{E}^{\leftarrow}$ with non-zero $\epsilon$, in general $\text{KL}(p_0||\widetilde{q}_T) > 0$.

**Assumption 3.1.** *Throughout this section, we assume that:*

*(1) For the forward process, we assume $f_t(x) = -\frac{1}{2}x$, $g_t = 1$ without loss of generality.*

*(2) The data distribution has the density $p_0$.*

*(3) There exists $c_U \in \mathbb{R}$ such that $U_0(x) - |x|^2/2 \geq c_U$, for any $x \in \mathbb{R}^d$, where $U_0 := -\log p_0$.*

Recall that a generic $g_t$ is equivalent to the time re-scaling (Appx. B.2). So this choice of $g_t = 1$ refers to the generic choice in VP-SDE [30]. The second assumption is also mild; in practice, if $p_0$ is a delta distribution, a common practice is that one tries to learn the mollified version $p_\sigma(x) := \int_{\mathbb{R}^d} \frac{1}{(2\pi\sigma^2)^{d/2}} \exp\left( -(x-y)^2/2\sigma^2 \right)p_0(y)\mathrm{d}y$ instead, as in the GAN [27] and early-stop techniques [7, 10]. The third assumption is not restrict, for example, $U_0(x) = |x|^2$ and $c_U = 0$. As many realistic datasets are almost compactly supported, we expect that $\rho_0 = e^{-U_0}$ decays faster than a Gaussian, namely, $\rho_0(x) \leq \widetilde{C}e^{-|x|^2/2}$ for some $\widetilde{C} > 0$, which reduces to the third one.

## 3.2 Asymptotic expansion of the KL divergence with respect to $\epsilon$

We introduce a time-dependent operator

$$\mathcal{L}_t^{(h^{\leftarrow})}(\mu)(x) := -\nabla \cdot \left( \left(\frac{1}{2}x + \frac{1 + h_t^{\leftarrow 2}}{2}\nabla \log p_t^{\leftarrow}(x)\right)\mu(x) \right) + \frac{h_t^{\leftarrow 2}}{2}\Delta\mu(x), \qquad (8)$$

which is the generator in the Fokker-Planck equation of $q_t$ for (2). Define an operator $\Phi_{s,t}^{(h^{\leftarrow})}$ as follows: given any function $\mu$, define $\Phi_{s,t}^{(h^{\leftarrow})}(\mu)$ to be the solution at time $t$ of the following Fokker-Planck equation with $r \in [s, t]$: $\partial_r\theta_r = \mathcal{L}_r^{(h^{\leftarrow})}(\theta_r)$, and with initial condition $\theta_s = \mu$. We define $\Phi_{s,t}^{(h^{\leftarrow})}(\mu) := \theta_t$. Properties of this operator are collected in Appx. C.1.

We use the notation $\widetilde{q}^\epsilon$ to refer to $\widetilde{q}$ since we need to calculate the its derivative for the small parameter $\epsilon$. The dependence of $\widetilde{q}$ on $h^{\leftarrow}$ is suppressed for short notations. We have the following asymptotic result with the proof given in Appx. C.3.

**Proposition 3.2.** *Define* $v_T := \partial_\epsilon \widetilde{q}_T^\epsilon|_{\epsilon=0}$ *as the first-order perturbation of* $\widetilde{q}_T^\epsilon$. *We have*

$$KL\big(p_0||\widetilde{q}_T^\epsilon\big) = L(h^\leftarrow)\epsilon^2 + \mathcal{O}\big(\epsilon^3\big), \tag{9}$$

*where*

$$L(h^\leftarrow) = \frac{1}{2}\int_{\mathbb{R}^d} \frac{v_T^2(x)}{p_0(x)}\,\mathrm{d}x, \qquad v_T = -\frac{1}{2}\int_0^T \big(1 + h_t^{\leftarrow 2}\big)\Phi_{t,T}^{(h^\leftarrow)}\big(\boldsymbol{\nabla}\cdot\big(p_t^\leftarrow \mathscr{E}_t^\leftarrow\big)\big)\,\mathrm{d}t. \tag{10}$$

## 3.3 The role of $h^\leftarrow$ in the Fokker-Planck operator $\mathcal{L}_t^{(h^\leftarrow)}$

Let the potential $U_t := -\log p_t$, and by the notation of time-reversal, $U_t^\leftarrow \equiv U_{T-t}$. When $h^\leftarrow > 0$, we can rewrite

$$\mathcal{L}_t^{(h^\leftarrow)}(\mu) = h_t^{\leftarrow 2}/2\Big(\triangle\mu + \boldsymbol{\nabla}\cdot\big(\nabla V_t^\leftarrow \mu\big)\Big), \qquad V_t^\leftarrow(x) := \big(1 + 1/h_t^{\leftarrow 2}\big)U_t^\leftarrow(x) - \frac{|x|^2}{2h_t^{\leftarrow 2}}. \tag{11}$$

We now introduce a probability distribution induced by the potential $V_t^\leftarrow$:

$$\rho_t^\leftarrow \propto \exp(-V_t^\leftarrow). \tag{12}$$

By convection, we also have $V_t = V_{T-t}^\leftarrow$ and $\rho_t = \rho_{T-t}^\leftarrow$. Note that $V_t^\leftarrow$ depends on $h^\leftarrow$ and when $h^\leftarrow \to \infty$, we have $V_t^{\leftarrow,\infty} = U_t^\leftarrow$. The role of $h^\leftarrow$ in $\mathcal{L}_t^{(h^\leftarrow)}$ now can be viewed as the time re-scaling and the effect of $\mathcal{L}_t^{(h^\leftarrow)}$ at a local time $t$ can be viewed as evolving the Fokker-Planck equation associated with the overdamped Langevin dynamics in the time-dependent potential $V_t^\leftarrow$ for $\mathcal{O}\big(h_t^{\leftarrow 2}/2\big)$ amount of time. When $h^\leftarrow \to \infty$, $\mathcal{L}_t^{(h^\leftarrow)}$ can be roughly viewed as constructing an "almost quasi-static" thermodynamics [6] bridging the initial $p_T$ and the (quasi-)equilibrium $p_0 = e^{-U_T^\leftarrow}$: for any distribution $\mu_t^\leftarrow$ (probably far away from $\rho_t^\leftarrow$), within a short time period $\Delta t$ slightly larger than $\mathcal{O}\big(1/h_t^{\leftarrow 2}\big)$, we have $\mu_{t+\Delta t}^\leftarrow \approx \rho_{t+\Delta t}^\leftarrow$, provided that $s \mapsto \mu_s^\leftarrow$ evolves according to $\mathcal{L}_s^{(h^\leftarrow)}$; see Appx. C.2. This key finding will guide our analysis of the solution operator $\Phi_{t,T}^{(h^\leftarrow)}$.

## 3.4 Score function is perturbed by a pulse

From (10), we know that $v_T$ combines the averaged effect of $\Phi_{t,T}^{(h^\leftarrow)}\big(\boldsymbol{\nabla}\cdot\big(p_t^\leftarrow \mathscr{E}_t^\leftarrow\big)\big)$ for various $t$. As a first result, we consider $\mathscr{E}_t^\leftarrow(x) = E(x)\delta_{t-s}$ for a fixed time instance $s \in [0,T)$, where $\delta_{t-s}$ is the Dirac function. In this case, $v_T$ no longer involves time integration and we have

$$v_T = -\frac{(1 + h_s^{\leftarrow 2})}{2}\Phi_{s,T}^{(h^\leftarrow)}\big(\boldsymbol{\nabla}\cdot\big(p_s^\leftarrow E\big)\big),$$

$$L(h^\leftarrow) = \frac{(1 + h_s^{\leftarrow 2})^2}{8}\int_{\mathbb{R}^d} \frac{\big(\Phi_{s,T}^{(h^\leftarrow)}\big(\boldsymbol{\nabla}\cdot\big(p_s^\leftarrow E\big)\big)\big)^2}{p_0}. \tag{13}$$

To proceed, we need to make additional assumptions:

**Assumption 3.3.**

*(1) For any $t \in [0,T]$, $U_t = -\log p_t$ is assumed to be strongly convex and the Hessian of $U_t$ is bounded by two positive numbers $m_t$ and $M_t$ as below*

$$m_t \boldsymbol{I}_d \preceq \nabla^2 U_t(x) \preceq M_t \boldsymbol{I}_d, \qquad \forall x \in \mathbb{R}^d. \tag{14}$$

*Moreover, assume that*

$$m_t \geq 1, \qquad t \in [0,T], \qquad and \qquad m_0 > 1. \tag{15}$$

*(2) For all $t \in [0,T]$, we choose $h_t^\leftarrow = \mathsf{h}$ as constant.*

Introduce

$$\kappa_t^\leftarrow := (1 + \mathsf{h}^{-2})m_t^\leftarrow - \mathsf{h}^{-2} \equiv (1 + \mathsf{h}^{-2})m_{T-t} - \mathsf{h}^{-2}, \tag{16}$$

which characterizes the Hessian lower bound of $V_t^\leftarrow$ (11). Note $\kappa_t^\leftarrow \approx m_t^\leftarrow$ when $\mathsf{h} \gg 1$. We would like to explain the reason behind the above assumptions, in particular, their practical relevance. **Part (1) Strong convexity:** this is a common assumption for Langevin sampling analysis [11, 14]. As the role of $\mathcal{L}_t^{(\mathsf{h}^\leftarrow)}$ is essentially simulating a Langevin dynamics with time-dependent potential, it is reasonable to use this assumption as a starting point. Moreover, the algorithmic improvement in gDDIM [10] is highly inspired by a form with assuming the data distribution as a Gaussian; Fréchet inception distance (FID) [15], a widely used metric to evaluate the quality of generative model, essentially treats the data (in the feature space) as Gaussians. Therefore, we believe that this assumption can still capture some main features of realistic datasets. **Part (2) $m_t \geq 1$ for any** $t \in [0, T]$**:** The second assumption $m_t \geq 1$ means that $p_t$ is more localized (smaller variance) than the standard Gaussian (unit variance), which is compatible with Assumption 3.1 (3). It can also ensure that $V_t$ is strongly convex with positive Hessian lower bounds, i.e., $\kappa_t^\leftarrow \geq 1$, for any $t \in [0, T]$ and $\mathsf{h} \in (0, \infty)$.

**Proposition 3.4.** *Under Assumptions 3.1 and 3.3, suppose that $\mathscr{E}_t^\leftarrow(x) = E(x)\delta_{t-s}$ for some fixed $s \in [0, T)$. If $\mathsf{h} \geq \mathsf{h}_{lb} := \max\left\{ 1/2, \mathsf{h}_0(1/2), \sqrt{\max\{0, -\frac{c_U}{\ln(2)}, \sup_{t\in[s,T]} C_t^{\leftarrow,(2)}\}} \right\}$, we have the upper bound of $L(\mathsf{h})$ in (13):*

$$L(\mathsf{h}) \leq C_\mathsf{h}(1 + \mathsf{h}^2)^2 \exp\left( -\int_s^T (\mathsf{h}^2 - C_r^{\leftarrow,(2)})\kappa_r^\leftarrow - C_r^{\leftarrow,(1)} dr \right), \tag{17}$$

*where $C_\mathsf{h} = \frac{1}{2}\int \left( \boldsymbol{\nabla}\cdot(p_s^\leftarrow E) \right)^2 / \rho_s^\leftarrow$, $C_t^{\leftarrow,(2)} = \frac{20 + 30M_t^{\leftarrow 2}}{m_t^{\leftarrow 2}}$; see Appx. E.1 for details about $C_t^{\leftarrow,(1)}$ and $\mathsf{h}_0(1/2)$; $c_U \in \mathbb{R}$ comes from Assumption 3.1. Moreover, $\lim_{\mathsf{h}\to\infty} C_\mathsf{h}$, $\lim_{\mathsf{h}\to\infty} C_t^{\leftarrow,(1)}$ exist.*

See Appx. E.1 for proofs. We remark that the above bound focuses on capturing the scaling with respect to $\mathsf{h}^2$ but may not be tight for other parameters. It remains interesting to see how we can improve the above upper bound. The *main conclusion* is that: if the error function $\mathscr{E}_t^\leftarrow$ is a pulse at time $t = s$, then for a large $\mathsf{h}$, $L(\mathsf{h})$ will decay to zero exponentially fast with respect to $\mathsf{h}$. For 1D Gaussian case, we can clearly observe such an exponential decay in Fig. 2a, where we pick $\mathscr{E}_t^\leftarrow = \mathbb{I}_{t\leq 0.95T}\nabla \log p_t^\leftarrow$. The intuition behind this exponential suppressed prefactor is that for large $\mathsf{h}$, $\Phi_{s,T}^{(\mathsf{h}^\leftarrow)}$ can be viewed as an almost quasi-static thermodynamics dragging any positive measure towards $\rho_T^\leftarrow \approx p_0$, as mentioned above in § 3.3. As $\nu = \boldsymbol{\nabla}\cdot\left(p_s^\leftarrow E\right)$ has measure zero, we can split it into positive and negative parts: $\nu = \nu^+ - \nu^-$ ( assume $\int \nu^\pm = 1$ WLOG). Each term $\Phi_{s,T}^{(\mathsf{h}^\leftarrow)}(\nu^+) \approx \rho_T^\leftarrow \approx \Phi_{s,T}^{(\mathsf{h}^\leftarrow)}(\nu^-)$, which explains that $\Phi_{s,T}^{(\mathsf{h}^\leftarrow)}(\nu) \approx 0$ for large $\mathsf{h}$.

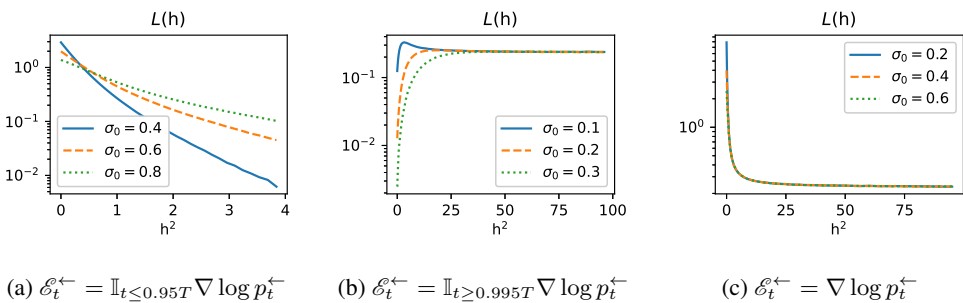

(a) $\mathscr{E}_t^\leftarrow = \mathbb{I}_{t\leq 0.95T}\nabla \log p_t^\leftarrow$    (b) $\mathscr{E}_t^\leftarrow = \mathbb{I}_{t\geq 0.995T}\nabla \log p_t^\leftarrow$    (c) $\mathscr{E}_t^\leftarrow = \nabla \log p_t^\leftarrow$

Figure 2: 1D Gaussian $p_0 = \mathcal{N}(0, \sigma_0^2)$ (with different $\sigma_0$ smaller than one) and $T = 2$. Panel (a) validates the exponential decay of $L(\mathsf{h})$ when the score function has no error near $t \approx T$, similar to Prop. 3.4. Panel (b) validates Prop. 3.5 that the ODE model ($h = 0$) outperforms the SDE model when there is a large score error at $t \approx T$. Panel (c) validates Prop. 3.6 that $\lim_{\mathsf{h}\to\infty} L(\mathsf{h})$ exists.

## 3.5 Score function is only perturbed near the end of the generative process

**Proposition 3.5.** *Under Assumptions 3.1 and 3.3, suppose that $\mathscr{E}_t^\leftarrow(x) = \mathbb{I}_{t\in[T-a,T]}E(x)$ where $a \ll 1$. Then when $a \ll 1$ and $\mathsf{h} \gg 1$, asymptotically ,*

$$L(0) \sim \frac{a^2}{8}\int_{\mathbb{R}^d} \frac{(\boldsymbol{\nabla}\cdot(p_0 E))^2}{p_0}, \qquad L(\mathsf{h}) \lesssim \frac{\left(1 - e^{-a\frac{\mathsf{h}^2}{2}\kappa_0}\right)^2}{2\kappa_0^2}\int_{\mathbb{R}^d} \frac{(\boldsymbol{\nabla}\cdot(p_0 E))^2}{p_0},$$

with $\kappa_0 = (1 + {}^1\!/\mathsf{h}^2)m_0 - {}^1\!/\mathsf{h}^2$ as in (16). *The upper bound of $L(\mathsf{h})$ is tight asymptotically.*

We remark that we made *no assumption on how* $a\mathsf{h}^2$ *scales*. If $\mathsf{h} \gg 1$, ${}^{L(\mathsf{h})}\!/_{L(0)} \lesssim 4\big(1 - e^{-a\mathsf{h}^2\kappa_0/2}\big)^2/a^2\kappa_0^2$. **Case (I):** If $a\mathsf{h}^2 \gg 1$, then ${}^{L(\mathsf{h})}\!/_{L(0)} \lesssim {}^4\!/a^2\kappa_0^2$, which is large as $a \ll 1$. **Case (II):** If $a\mathsf{h}^2 \ll 1$, then ${}^{L(\mathsf{h})}\!/_{L(0)} \lesssim \mathsf{h}^4$, which is still large. In either case, ${}^{L(\mathsf{h})}\!/_{L(0)}$ is expected to be large for a general $E$ and the ODE model is preferred in this case. The intuition is that there is almost no time for the operator $\mathcal{L}_t^{(h^\leftarrow)}$ to suppress the error $E$, so the prefactor $1 + (h_t^\leftarrow)^2$ in $v_T$ (10) dominates (which is the key difference compared with Prop. 3.4). The proof is postponed to Appx. E.3. In Fig. 2b, we consider 1D Gaussian and only perturb the score function at the end of the generative process ($\mathscr{E}_t^\leftarrow = \mathbb{I}_{t \geq 0.995T}\nabla \log p_t^\leftarrow$); clearly, the SDE-based models have comparatively larger error.

### 3.6 General error in score function

We can generalize Prop. 3.4 and Prop. 3.5 to a general error function $\mathscr{E}^\leftarrow$, and observe that $L(\mathsf{h})$ will converge to a constant exponentially fast as $\mathsf{h} \to \infty$.

**Proposition 3.6.** *Under Assumptions 3.1 and 3.3, we consider a general error function* $(t, x) \mapsto \mathscr{E}_t^\leftarrow(x)$. *Let* $\gamma = \inf_{\mathsf{h} \geq \mathsf{h}_{lb}} \inf_{t \in [0,T]} \kappa_t^\leftarrow$. *For any* $\alpha \in (0, 1)$ *and* $\beta \in (0, 2)$, *when* $\mathsf{h} \gg 1$,

$$L(\mathsf{h}) \lesssim (1 + \alpha^2)\mathcal{T} + (1 + \alpha^{-2})\, C\, \gamma^{-1}(1 + \mathsf{h}^2)\exp\big(-\mathsf{h}^{2-\beta}\gamma\big),$$
$$L(\mathsf{h}) \gtrsim (1 - \alpha^2)\mathcal{T} - (\alpha^{-2} - 1)C\, \gamma^{-1}(1 + \mathsf{h}^2)\exp\big(-\mathsf{h}^{2-\beta}\gamma\big),$$

*where $C$ is given in Appx. F.1 and $\mathcal{T}$ (only depending on $p_0$ and $\mathscr{E}_T^\leftarrow$) is upper bounded by*

$$0 \leq \mathcal{T} \lesssim \frac{1}{2m_0^2}\int_{\mathbb{R}^d}\frac{\big(\boldsymbol{\nabla} \cdot (p_0\mathscr{E}_T^\leftarrow)\big)^2}{p_0} \equiv \frac{1}{2m_0^2}\int_{\mathbb{R}^d}\Big(\nabla \log p_0 \cdot \mathscr{E}_T^\leftarrow + \boldsymbol{\nabla} \cdot \mathscr{E}_T^\leftarrow\Big)^2 p_0. \tag{18}$$

In the limit $\mathsf{h} \to \infty$, $(1 - \alpha^2)\mathcal{T} \lesssim L(\mathsf{h}) \lesssim (1 + \alpha^2)\mathcal{T}$, where $\alpha \in (0, 1)$ is arbitrary. Hence, the tail behavior is that $L(\mathsf{h})$ converges to $\mathcal{T}$ exponentially fast as $\mathsf{h} \to \infty$. If we assume that $\mathscr{E}_T^\leftarrow = \nabla \log p_0$, $p_0 = \mathcal{N}(0, \sigma_0^2 \boldsymbol{I}_d)$ in $d$-dimension, then the above upper bound is simply $\mathcal{T} \lesssim d$, which is independent of $\sigma_0$ (see Appx. F.2). For 1D Gaussian in Fig. 2c, we can indeed observe that $\lim_{\mathsf{h} \to \infty} L(\mathsf{h})$ exists, and is bounded above by $d = 1$; see Appx. G for more types of error functions.

The above upper bound has an interesting tight connection to the generator of (overdamped) Langevin dynamics with drift $-\nabla U_0 \equiv \nabla p_0$. If we adopt constrained score models [23, 25], namely, parameterizing $\log p_t$ instead of the score function $\nabla \log p_t$ during training, the error $\mathscr{E}_T^\leftarrow = \nabla\varphi$ for some scalar-valued function $\varphi$. Then the above upper bound becomes

$$\frac{1}{2m_0^2}\int_{\mathbb{R}^d}(\triangle\varphi - \nabla U_0 \cdot \nabla\varphi)^2 e^{-U_0} = \frac{1}{2m_0^2}\int_{\mathbb{R}^d}(\mathcal{L}^*\varphi)^2 e^{-U_0}, \tag{19}$$

where $\mathcal{L}^*(\varphi) := \triangle\varphi - \nabla U_0 \cdot \nabla\varphi$ whose adjoint operator $\mathcal{L}(\mu) = \nabla \cdot (\nabla U_0\mu) + \triangle\mu$ is the Fokker-Planck generator of the following Langevin dynamics $\mathrm{d}X_t = -\nabla U_0(X_t)\,\mathrm{d}t + \sqrt{2}\,\mathrm{d}W_t$ where $W_t$ is the standard Brownian motion. We remark that this formula (19) is general for constrained score models [23, 25]; see also F.2 for elaborations. An interesting open question is whether and how we can take the above upper bound into consideration when designing the loss function.

### 3.7 An application: exploring the effect of training weight

The above theoretical discussions suggest that diffusion models with large diffusion coefficients are more negatively affected by score error near data's side, whereas the ODE model is more negatively affected by the score error near the noise end. This leads us to conjecture that if we can control the training (e.g., by optimizing the training weight $\omega_t$), so that the score error distribution near the noise end is reduced and meanwhile the score-matching loss is not significantly impacted, then it will very likely improve the ODE models. We report preliminary numerical experiments to support this idea in Appx. I, whereas a comprehensive study will be left as future works.

# 4 Numerical experiments

We present experiments on 2D Gaussian mixture model, Swiss roll, CIFAR10 to support our theoretical results: when numerical discretization error is not dominating, the sampling quality increases as h increases, a reminiscent of Prop. 3.4. Experimental details are postponed to Appx. H, as well as more numerical results (e.g., results about 1D Gaussian mixture and MNIST). Results by adopting various weight functions, a technique arising from theoretical predictions, are postponed to Appx. I. Source codes are available at `https://github.com/yucaoyc/OptimalDiffusion`.

**Example 1: 2D 4-mode Gaussian mixture.** We verify the theoretical results on 2D Gaussian mixture with a specified score error. In Fig. 3, a clear trend is that a higher h produces generated distributions that better match the marginal densities of $p_0$, and it is numerically verified by the purple line of Fig. 4a. In Fig. 4, with multiple settings of $\mathscr{E}_t^{\leftarrow}$ and $\epsilon$, we observe a consistent phenomenon that as h increases, the KL divergence of true data and generated data converges exponentially fast, thus validating Prop. 3.6. It worths noticing that in all three settings of $\mathscr{E}_t^{\leftarrow}$, by simply adopting a larger diffusion coefficient $h$ in (3), we can obtain better generative models without any extra training costs.

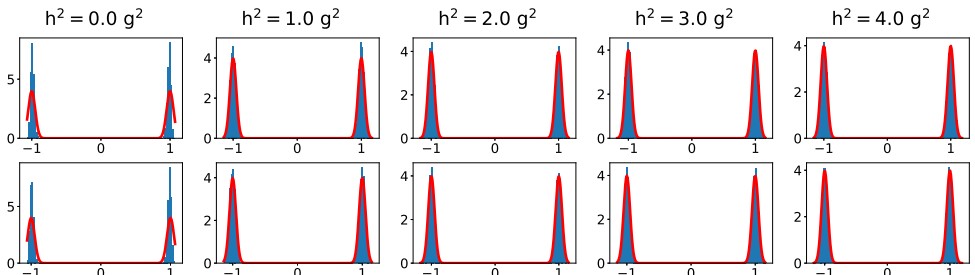

Figure 3: Visualization of marginal densities of 2D 4-mode Gaussian mixture, where $\mathscr{E}_t^{\leftarrow} = \nabla \log p_t^{\leftarrow}$ and $\epsilon = 0.2$. The top row shows the marginal distribution of the first coordinate and the bottom row for the second coordinate. The red lines are the exact marginal distributions of $p_0$ and the histograms (blue) visualize the empirical densities of generated samples.

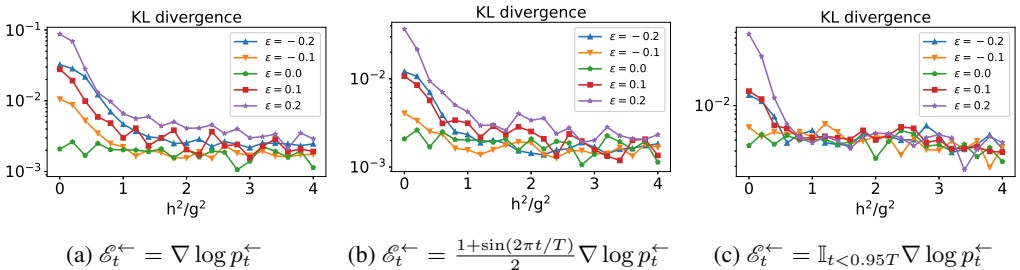

(a) $\mathscr{E}_t^{\leftarrow} = \nabla \log p_t^{\leftarrow}$     (b) $\mathscr{E}_t^{\leftarrow} = \frac{1+\sin(2\pi t/T)}{2} \nabla \log p_t^{\leftarrow}$     (c) $\mathscr{E}_t^{\leftarrow} = \mathbb{I}_{t<0.95T} \nabla \log p_t^{\leftarrow}$

Figure 4: Numerical results of 2D 4-mode Gaussian mixture. The above panels show that the KL divergences between the true distribution and the generated samples overall decay as h increases for various types of error perturbation of score functions.

**Example 2: Swiss roll.** We consider Swiss roll, a more complex distribution without exact score function available. We train the score function with the denoising score-matching objective [33] (Appx. H). The first plot in Fig. 5a shows the difference between true data and generated data measured by Wasserstein distance, which decays to zero exponentially fast, verifying Prop. 3.6. In the second plot of Fig. 5a, we tested various $h$ and time steps for the generative process. The ODE model ($h = 0$) does not improve, as the number of time discretization steps increases, but near the continuous-time limit, all SDE cases ($h > 0$) are better than the ODE model. It suggests that our conclusions here is limited to the continuous-time setting. The exploration of time discretization errors will be future works. The generated data results in Fig. 5b demonstrate that with increasing $h$, the sample quality is improved; see Appx. H.6 for more results.

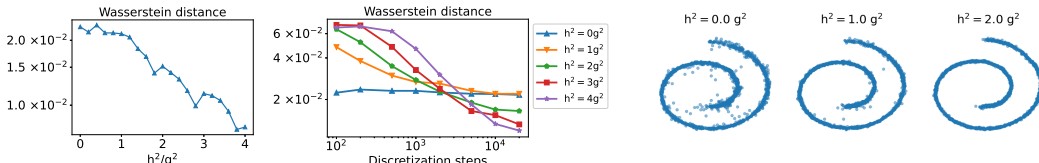

(a) Improved model performance with larger h  (b) Visualization results for selective h

Figure 5: Numerical results of Swiss roll. Panel (a) shows the decay of Wasserstein distance between the true distribution and the generated samples with increasing $h$ and 20,000 time-discretization steps, and the decay of Wasserstein distance with the increasing number of time-discretization steps and different $h$. Panel (b) shows generated samples with different $h$.

**Example 3: CIFAR-10.**  When using a large amount of parameters for score matching in practice, we observe that SDEs appear to perform better than ODEs as discretization error descreases, a result similarly observed on Swiss roll. This implicates the practical applications on generating samples of better quality under a given (pre-trained) score-matching model.

Table 1: **CIFAR-10:** We evaluate FIDs with different discretization steps and $^{h^2}/_{g^2}$ on a pre-trained checkpoint entitled "vp/cifar10_ddpmpp_continuous" in [30]. We do not use any correctors and evaluate FIDs on $10^4$ samples, thus the results for $^{h^2}/_{g^2} = 0, 1$ are different from the reported values.

| Discretization steps | 100 | 200 | 500 | 1000 | 2000 | 3000 | 4000 |
|---|---|---|---|---|---|---|---|
| $h^2/g^2 = 0$ | **22.43** | **8.12** | **7.20** | 6.89 | 6.98 | 7.27 | 7.33 |
| $h^2/g^2 = 1$ | 31.72 | 15.72 | 7.23 | **6.70** | 6.71 | 6.95 | 7.08 |
| $h^2/g^2 = 2$ | 52.77 | 26.68 | 10.78 | 6.99 | **6.70** | **6.69** | **6.98** |
| $h^2/g^2 = 4$ | 92.83 | 46.11 | 20.47 | 10.17 | 7.20 | 7.09 | 7.01 |

# 5  Summary and outlook

In this work, we study the effect of the diffusion coefficient on the quality of overall sample generation in the generative process. Theoretically, we provide understandings of scenarios in which the ODE-based model and the SDE-based model is superior than the other; see Prop. 3.4 and Prop. 3.5. Numerically, these results are validated via toy examples as well as benchmark datasets.

There are many interesting directions for continuing works. (1) As we focus on the asymptotic case, a time-dependent $h_t^{\leftarrow}$ with large magnitude (i.e., $h_t^{\leftarrow} \gg 1$ for all $t$) is essentially no different from a constant $h_t^{\leftarrow} \equiv h$ with $h \gg 1$. Whether it is possible to use time-dependent $t \mapsto h_t^{\leftarrow}$ to combine the advantages of ODE and large diffusion cases in dealing with different types of error of score functions in the non-asymptotic region is still an open question. (2) Can we design a practical criterion that directly learn the optimal magnitude of the noise-level function $h_t^{\leftarrow}$ by looking at the score-training error distribution? Can we develop certain theoretical understanding of the empirical results in [19]? (3) How can we find a stable and accurate numerical scheme to deal with the fast diffusion case? (4) How can we generalize the above theoretical results by removing the convexity assumption, and including the low-dimensional feature of datasets into the theory [8, 5, 18]?

Concerning the broad impact, though we don't foresee any negative social impact of this work, the potential improvement of generative model might relate to creation of "deep fakes".

## Acknowledgments and Disclosure of Funding

YC is sponsored by Shanghai Pujiang Program, and acknowledges the support from City University of Hong Kong during his visit. JC and YL are supported by the NSFC Major Research Plan - Interpretable and General-purpose Next-generation Artificial Intelligence (92370205). XZ is supported by General Research Funds from the Research Grants Council of the Hong Kong Special Administrative Region, China (Project No. 11308121, 11318522) and the NSFC/RGC Joint Research Scheme [RGC Project No. N-CityU102/20 and NSFC Project No. 12061160462].

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

# Supplementary Material for "Exploring the Optimal Choice for Generative Processes in Diffusion Models: Ordinary vs Stochastic Differential Equations"

## A  Notation Convention.

Table 2: Summary of important quantities in this paper

| Quantity | Notation | Notes |
|---|---|---|
| Forward process | $X_t$ | $t = 0$: data distribution |
| Backward/generative process | $Y_t$ | $t = 0$: noise distribution |
| Backward process with inexact score | $\widetilde{Y}_t$ | |
| Distribution of forward process | $p_t$ | $p_t := \mathrm{law}(X_t)$ |
| Distribution of backward process | $q_t$ | $\mathrm{law}(Y_t) := q_t \equiv p_{T-t} \equiv p_t^{\leftarrow}$ |
| Distribution of approximated backward process | $\widetilde{q}_t$ | $\widetilde{q}_t := \mathrm{law}(\widetilde{Y}_t)$, $\widetilde{q}_t = q_t$ when error $\epsilon = 0$ |
| Error function of the score | $\epsilon \mathscr{E}_t^{\leftarrow}$ | $0 \le \epsilon \ll 1$ and $\mathscr{E}_t^{\leftarrow} = \mathcal{O}(1)$ |
| The exact potential | $U_t$ | $p_t := e^{-U_t}$ |
| Modified potential | $V_t^{\leftarrow}$ | see (11) |
| Normalizing constants | $Z_V := \int e^{-V}$ | $V$ is arbitrary |

## B  Discussion and proof for § 2

### B.1  Proof of (3)

We re-state the conclusion in (3) in the following lemma:

**Lemma B.1.** *For any function $h_t^{\leftarrow}$, if one chooses $A^{\leftarrow}$ as in (3), then we have $q_t(x) = p_{T-t}(x)$ for any $t \in [0, T]$ and $x \in \mathbb{R}^d$.*

*Proof.* We can easily write down the Fokker-Planck equation of (1)

$$\partial_t p_t(x) = -\boldsymbol{\nabla} \cdot \big(f_t(x)p_t(x)\big) + \frac{g_t^2}{2} \triangle p_t(x).$$

Since we need to ensure $q_t = p_{T-t}$, we require

$$\begin{aligned}
\partial_t q_t(x) &= \boldsymbol{\nabla} \cdot \big(f_t^{\leftarrow}(x)q_t(x)\big) - \frac{g_t^{\leftarrow 2}}{2} \triangle q_t(x) \\
&= \boldsymbol{\nabla} \cdot \Big(f_t^{\leftarrow}(x)q_t(x) - \frac{g_t^{\leftarrow 2} + h_t^{\leftarrow 2}}{2} \nabla q_t(x)\Big) + \frac{h_t^{\leftarrow 2}}{2} \triangle q_t(x) \\
&= \boldsymbol{\nabla} \cdot \Big(\big(f_t^{\leftarrow}(x) - \frac{g_t^{\leftarrow 2} + h_t^{\leftarrow 2}}{2} \nabla \log q_t(x)\big)q_t(x)\Big) + \frac{h_t^{\leftarrow 2}}{2} \triangle q_t(x) \\
&= \boldsymbol{\nabla} \cdot \Big(\big(f_t^{\leftarrow}(x) - \frac{g_t^{\leftarrow 2} + h_t^{\leftarrow 2}}{2} \nabla \log p_t^{\leftarrow}(x)\big)q_t(x)\Big) + \frac{h_t^{\leftarrow 2}}{2} \triangle q_t(x) \\
&= -\boldsymbol{\nabla} \cdot \Big(A_t^{\leftarrow}(x)q_t(x)\Big) + \frac{h_t^{\leftarrow 2}}{2} \triangle q_t(x),
\end{aligned}$$

by noting that $A_t^{\leftarrow}(x)$ is specified in (3). This equation is exactly the Fokker-Planck equation of (2). □

### B.2  The role of $g_t$

The function $t \mapsto g_t$ as the diffusion coefficient in the forward Fokker-Planck equation (1), in fact, plays a role as time re-scaling as long as $g_t > 0$ for any $t$. Indeed, if we have an SDE in the following

form (the VP-SDE in [30])

$$dX_t = -\frac{g_t^2}{2}X_t \, dt + g_t \, dW_t, \qquad t \in [0, T],$$

then by introducing $\tau : \mathbb{R}^+ \to \mathbb{R}^+$ via the ODE $\tau'(t) = \big(g_{\tau(t)}\big)^{-2}$, $\tau(0) = 0$, we know that $\underline{X}_t := X_{\tau(t)}$ satisfies the following SDE [20]:

$$d\underline{X}_t = -\frac{1}{2}\underline{X}_t \, dt + dW_t, \qquad t \in [0, \theta],$$

where $\theta := \tau^{-1}(T)$ means the inverse function of $\tau$ at time $T$. The Brownian motion $W$ in $\underline{X}_t$ is not the same Brownian motion in $X_t$, i.e., the driven-noise in the last two equations are not the same technically; we slightly abuse the notation for the simplicity of notations. By Lem. B.1, this SDE has the backward process as follows:

$$
\begin{aligned}
d\underline{Y}_t &\overset{(3)}{=} \left(\frac{1}{2}\underline{Y}_t + \frac{1 + \underline{h}_t^{\leftarrow 2}}{2}\nabla \log \underline{p}_t^{\leftarrow}(\underline{Y}_t)\right) dt + \underline{h}_t^{\leftarrow} \, dW_t \qquad t \in [0, \theta] \\
&= \left(\frac{1}{2}\underline{Y}_t + \frac{1 + \underline{h}_{\theta-t}^2}{2}\nabla \log \underline{p}_{\theta-t}(\underline{Y}_t)\right) dt + \underline{h}_{\theta-t} \, dW_t,
\end{aligned}
\tag{20}
$$

where $\underline{p}_t$ is the density function of $\underline{X}_t$ by notation conventions in § 2. By Lem. B.1, so far we know that

$$\mathrm{law}(\underline{Y}_{\theta-t}) \overset{\text{Lem. B.1}}{=} \mathrm{law}(\underline{X}_t) = \mathrm{law}(X_{\tau(t)}).$$

Let $f_s := \theta - (\tau^{-1})(T - s)$ and $Y_s := \underline{Y}_{f_s}$, where $s \in [0, T]$. By straightforward computation, we know

$$
\begin{aligned}
dY_s &= f_s'\left(\frac{1}{2}\underline{Y}_{f_s} + \frac{1 + \underline{h}_{\theta-f_s}^2}{2}\nabla \log \underline{p}_{\theta-f_s}(\underline{Y}_{f_s})\right) dt + \sqrt{f_s' \, \underline{h}_{\theta-f_s}^2} \, dW_s \\
&= f_s'\left(\frac{1}{2}Y_s + \frac{1 + \underline{h}_{\theta-f_s}^2}{2}\nabla \log p_{T-s}(Y_s)\right) dt + \sqrt{f_s' \, \underline{h}_{\theta-f_s}^2} \, dW_s.
\end{aligned}
$$

To get the second line, we used the fact that

$$\underline{p}_{\theta-f_s} = \mathrm{law}\big(\underline{X}_{\theta-f_s}\big) = \mathrm{law}\big(\underline{X}_{\tau^{-1}(T-s)}\big) = \mathrm{law}(X_{T-s}) \equiv p_{T-s}.$$

By chain rules, it is easy to compute that

$$f_s' = \frac{1}{\tau'\big(\tau^{-1}(T-s)\big)} = \frac{1}{\big(g_{T-s}\big)^{-2}} = g_{T-s}^2 > 0.$$

Therefore, the above SDE for $Y_s$ has the form

$$dY_s = \left(\frac{g_{T-s}^2}{2}Y_s + g_{T-s}^2\frac{1 + \underline{h}_{\theta-f_s}^2}{2}\nabla \log p_s^{\leftarrow}(Y_s)\right) dt + g_{T-s}\underline{h}_{\theta-f_s} \, dW_s.$$

This matches the form in Lem. B.1 by choosing

$$h_s^{\leftarrow} = g_{T-s}\underline{h}_{\theta-f_s} \equiv g_s^{\leftarrow}\underline{h}_{\tau^{-1}(T-s)}.$$

Hence, if we simply pick $\underline{h} = c$ as a constant function in (20), it has the same effect as choosing $h_s^{\leftarrow} = cg_s^{\leftarrow}$ where $c \in \mathbb{R}^+$. The former ($\underline{h} = c$) is used in § 3 for simplicity in theoretical analysis, and the latter ($h_s^{\leftarrow} = cg_s^{\leftarrow}$) is used in numerical experiments in § 4 to match previous literature in practice (namely, a general $g$). **In conclusion,** the above discussion justifies the consistency of notation and set-up between our theoretical analysis and numerical experiments.

### B.3 Existing analysis of sample generation quality

In practice, one simulates the following SDE:

$$dZ_t = \left(-f_t^{\leftarrow}(Z_t) + \frac{(g_t^{\leftarrow})^2 + (h_t^{\leftarrow})^2}{2}\mathfrak{S}_t^{\leftarrow}(Z_t)\right) dt + h_t^{\leftarrow} \, dW_t, \qquad \mathrm{law}(Z_0) = \mathcal{N}(0, I_d). \tag{21}$$

From the analysis by Chen et al. [7, Theorem 2.1] for the case $h^{\leftarrow} = g^{\leftarrow}$, one has

$$\mathrm{KL}\big(p_0||\widehat{q}\big) \leq \mathrm{KL}\big(p_T||\mathcal{N}(0, I_d)\big) + \mathcal{O}\big(T\epsilon^2\big) + \mathcal{O}\big(T^2 d/N\big),$$

where $\widehat{q}$ is the distribution of $Z_T$ after applying the exponential integrator to (21) and $N$ is the number of time-discretization steps. The first term $\mathrm{KL}\big(p_T||\mathcal{N}(0, I_d)\big) \leq (\mathcal{M}_2 + d)e^{-T}$, where $\mathcal{M}_2 = \mathbb{E}_{p_0}|x|^2$ is the second moment of data distribution.

Therefore, to ensure that $\mathrm{KL}\big(p_0||\widehat{q}\big) \leq \delta$, it is sufficient to choose

$$T = \mathcal{O}\big(\log\big((\mathcal{M}_2 + d)/\delta\big)\big).$$

The time-discretization error can be eliminated by choosing $N \to \infty$. What is so far less clear in literature is the term $\mathcal{O}\big(T\epsilon^2\big)$.

### B.4  Existing error bounds appear to fail to characterize the optimal $h^{\leftarrow}$

Note that the target dynamics $q_t$ in (2) and the approximated dynamics $\widetilde{q}_t$ in (7) only differ in the drift term. Recall that $q_T = p_0$, and to estimate $\mathrm{KL}\big(p_0||\widetilde{q}_T\big) \equiv \mathrm{KL}\big(q_T||\widetilde{q}_T\big)$, we can simply estimate how the quantity $\mathrm{KL}\big(q_t||\widetilde{q}_t\big)$ changes for $t \in (0, T)$. By [7, Lemma C.1], for any $t \in \mathbb{R}$,

$$\partial_t \mathrm{KL}\big(q_t||\widetilde{q}_t\big) = -\frac{h_t^{\leftarrow 2}}{2}\mathscr{J}\big(q_t||\widetilde{q}_t\big) + \frac{g_t^{\leftarrow 2} + h_t^{\leftarrow 2}}{2}\mathbb{E}\Big[\big\langle -\epsilon\mathscr{E}_t^{\leftarrow}(Y_t), \nabla\log\frac{q_t(Y_t)}{\widetilde{q}_t(Y_t)}\big\rangle\Big], \qquad (22)$$

where the Fisher information $\mathscr{J}\big(p||q\big) := \int \mathrm{d}p\left|\nabla\log\frac{p}{q}\right|^2$. By Cauchy-Schwarz inequality,

$$\begin{aligned}
&\partial_t \mathrm{KL}\big(q_t||\widetilde{q}_t\big) \\
&\leq -\frac{h_t^{\leftarrow 2}}{2}\mathscr{J}\big(q_t||\widetilde{q}_t\big) + \frac{g_t^{\leftarrow 2} + h_t^{\leftarrow 2}}{2}\Big(\frac{c^2}{2}\mathbb{E}\Big[\|\epsilon\mathscr{E}_t^{\leftarrow}(Y_t)\|^2\Big] + \frac{1}{2c^2}\mathbb{E}\Big[\Big\|\nabla\log\frac{q_t(Y_t)}{\widetilde{q}_t(Y_t)}\Big\|^2\Big]\Big) \\
&= \frac{(h_t^{\leftarrow 2} + g_t^{\leftarrow 2})^2}{8h_t^{\leftarrow 2}}\epsilon^2\mathbb{E}\big[\|\mathscr{E}_t^{\leftarrow}(Y_t)\|^2\big],
\end{aligned} \qquad (23)$$

where we chose $c^2 = \frac{h_t^{\leftarrow 2} + g_t^{\leftarrow 2}}{2h_t^{\leftarrow 2}}$ in the last line. This bound captures the scaling extremely well when $h^{\leftarrow} = g^{\leftarrow}$, which helps to establish the effectiveness of score-based diffusion method in [7]. However, this bound is not directly applicable for a general $h^{\leftarrow}$, as it is clear that this bound fails to provide useful information when $h^{\leftarrow} \approx 0$ (namely, the probability flow), as well as the large diffusion case ($h^{\leftarrow} \to \infty$). From directly optimizing the upper bound, i.e., minimizing $\frac{(h_t^{\leftarrow 2} + g_t^{\leftarrow 2})^2}{8h_t^{\leftarrow 2}}$, one ends up with the choice $h^{\leftarrow} = g^{\leftarrow}$, which has been used in many literature. We acknowledge that $h^{\leftarrow} = g^{\leftarrow}$ is an effective choice; however, as we show in § 4, this is not really the optimal case in general, and the above argument cannot justify choosing the diffusion model over the probability flow.

## C  Discussion for § 3 and proof of Prop. 3.2

### C.1  The operator $\Phi_{s,t}^{(h^{\leftarrow})}$

**Lemma C.1.**

- $\Phi_{(\cdot),(\cdot)}^{(h^{\leftarrow})}$ satisfies the following property, i.e., for any $s, t, r \in \mathbb{R}$,

$$\Phi_{t,r}^{(h^{\leftarrow})} \circ \Phi_{s,t}^{(h^{\leftarrow})} = \Phi_{s,r}^{(h^{\leftarrow})}. \qquad (24)$$

  Moreover, $\Phi_{t,t}^{(h^{\leftarrow})} = Id$ is an identity operator for any $t$.
- For any $s, t \in \mathbb{R}$,

$$\partial_s\big(\Phi_{s,t}^{(h^{\leftarrow})}(\mu)\big) = -\Phi_{s,t}^{(h^{\leftarrow})}\big(\mathcal{L}_s^{(h^{\leftarrow})}\mu\big) \qquad \partial_t\big(\Phi_{s,t}^{(h^{\leftarrow})}(\mu)\big) = \mathcal{L}_t^{(h^{\leftarrow})}\big(\Phi_{s,t}^{(h^{\leftarrow})}(\mu)\big). \qquad (25)$$

*Proof.* The structure in (24) is easy to imagine and is thus omitted herein. Next, we shall only prove the first result in (25) for illustration:

$$
\begin{aligned}
\partial_s\big(\Phi_{s,t}^{(h^\leftarrow)}(\mu)\big) &= \lim_{\delta s \to 0^+} \frac{\Phi_{s+\delta s,t}^{(h^\leftarrow)}(\mu) - \Phi_{s,t}^{(h^\leftarrow)}(\mu)}{\delta s} \\
&\stackrel{(24)}{=} \lim_{\delta s \to 0^+} \frac{\Phi_{s,t}^{(h^\leftarrow)}\Phi_{s+\delta s,s}^{(h^\leftarrow)}(\mu) - \Phi_{s,t}^{(h^\leftarrow)}(\mu)}{\delta s} \\
&= \lim_{\delta s \to 0^+} \frac{\Phi_{s,t}^{(h^\leftarrow)}\big(\Phi_{s+\delta s,s}^{(h^\leftarrow)} - \mathrm{Id}\big)(\mu)}{\delta s} \\
&= \lim_{\delta s \to 0^+} \frac{\Phi_{s,t}^{(h^\leftarrow)}\big(\mathrm{Id} - \delta s \mathcal{L}_s^{(h^\leftarrow)} - \mathrm{Id}\big)(\mu)}{\delta s} \\
&= -\Phi_{s,t}^{(h^\leftarrow)}\big(\mathcal{L}_s^{(h^\leftarrow)}(\mu)\big).
\end{aligned}
$$

$\square$

**Lemma C.2.** *Suppose* $(t,x) \mapsto \mu_t(x)$, $(t,x) \mapsto \theta_t(x)$ *are time-dependent functions. Suppose* $\partial_t \mu_t = \mathcal{L}_t^{(h^\leftarrow)}(\mu_t) + \theta_t$ *with* $\mu_0 = 0$, *then*

$$
\mu_T = \int_0^T \Phi_{t,T}^{(h^\leftarrow)}(\theta_t)\, \mathrm{d}t.
$$

*Proof.* Note that

$$
\partial_t\big(\Phi_{t,0}^{(h^\leftarrow)}\mu_t\big) \stackrel{(25)}{=} \Phi_{t,0}^{(h^\leftarrow)}\big(-\mathcal{L}_t^{(h^\leftarrow)}\mu_t\big) + \Phi_{t,0}^{(h^\leftarrow)}\big(\mathcal{L}_t^{(h^\leftarrow)}\mu_t + \theta_t\big) = \Phi_{t,0}^{(h^\leftarrow)}(\theta_t).
$$

Therefore,

$$
\Phi_{T,0}^{(h^\leftarrow)}(\mu_T) = \int_0^T \Phi_{t,0}^{(h^\leftarrow)}(\theta_t)\, \mathrm{d}t.
$$

By applying the operator $\Phi_{0,T}^{(h^\leftarrow)}$ to both sides, we have

$$
\mu_T = \int_0^T \Phi_{0,T}^{(h^\leftarrow)} \circ \Phi_{t,0}^{(h^\leftarrow)}(\theta_t)\, \mathrm{d}t \stackrel{(24)}{=} \int_0^T \Phi_{t,T}^{(h^\leftarrow)}(\theta_t)\, \mathrm{d}t.
$$

$\square$

## C.2 Time re-scaling of $\mathcal{L}^{(h^\leftarrow)}$ and connections to almost quasi-static Langevin process

By time re-scaling, it is immediate to obtain the following result:

**Lemma C.3.** *Suppose* $h_t^\leftarrow = \mathsf{h}$ *for any* $t \in [0,T]$ *for simplicity. For any probability distribution* $\mu$, *the probability distribution* $\Phi_{s,t}^{(h^\leftarrow)}(\mu)$ *(with* $s < t$*) is the final state of the following PDE on the time interval* $\widetilde{r} \in \big[0, {}^{(t-s)\mathsf{h}^2}/_2\big]$:

$$
\partial_{\widetilde{r}}\widetilde{\mu}_{\widetilde{r}} = \triangle\widetilde{\mu}_{\widetilde{r}} + \boldsymbol{\nabla}\cdot\Big(\nabla V_{s+2\widetilde{r}/\mathsf{h}^2}^\leftarrow \widetilde{\mu}_{\widetilde{r}}\Big), \qquad \widetilde{\mu}_0 = \mu. \tag{26}
$$

*It corresponds to the following Langevin with time-dependent potential:*

$$
\mathrm{d}X_{\widetilde{r}} = -\nabla V_{s+2\widetilde{r}/\mathsf{h}^2}^\leftarrow(X_{\widetilde{r}})\, \mathrm{d}\widetilde{r} + \sqrt{2}\, \mathrm{d}W_{\widetilde{r}}, \qquad law(X_0) = \mu.
$$

*Proof.* To obtain $\Phi_{s,t}^{(h^\leftarrow)}(\mu)$, we simply solve the following PDE:

$$
\partial_r \mu_r \stackrel{(11)}{=} \frac{\mathsf{h}^2}{2}\big(\triangle\mu_r + \boldsymbol{\nabla}\cdot\big(\nabla V_r^\leftarrow \mu_r\big) \qquad \mu_s = \mu.
$$

By change of variables $\widetilde{r} = (r-s)\mathsf{h}^2/2$ and $\widetilde{\mu}_{\widetilde{r}} := \mu_r \equiv \mu_{s+2\widetilde{r}/\mathsf{h}^2}$, we immediately have (26), and the corresponding Langevin dynamics easily follows. $\square$

When $\mathsf{h} \to \infty$, the potential in the Langevin dynamics $\widetilde{r} \mapsto V_{s+2\widetilde{r}/\mathsf{h}^2}^\leftarrow$ evolves extremely slowly. From (11), we also know that $V_{s+2\widetilde{r}/\mathsf{h}^2}^\leftarrow \approx U_s^\leftarrow$ when $\widetilde{r} = 0$ and is approximately $U_t^\leftarrow$ when $\widetilde{r} = \frac{(t-s)\mathsf{h}^2}{2}$. Therefore, the Langevin dynamics can be viewed as an almost *quasi-static process* [6] approximately transforming the state $p_s^\leftarrow$ to the state $p_t^\leftarrow$ over an extremely long time period though.

## C.3  Proof of Prop. 3.2

Denote

$$v_t(x) := \partial_\epsilon \widetilde{q}_t^\epsilon(x)|_{\epsilon=0}, \qquad \zeta_t(x) := \partial_\epsilon^2 \widetilde{q}_t^\epsilon(x)|_{\epsilon=0}.$$

Namely, we expand $\widetilde{q}_t^\epsilon$ via the following

$$\widetilde{q}_t^\epsilon(x) = \underbrace{\widetilde{q}_t^0(x)}_{\equiv q_t(x)} + \epsilon v_t(x) + \frac{\epsilon^2}{2}\zeta_t(x) + \mathcal{O}(\epsilon^3).$$

Recall that when $\epsilon = 0$, $\widetilde{q}_t^0 \equiv q_t$; in particular, $\widetilde{q}_T^0 = q_T \overset{\text{Lem. B.1}}{=} p_0$. The cost function can be easily expanded via Taylor's formula:

$$\text{KL}(p_0||\widetilde{q}_T^\epsilon) = -\epsilon \int_{\mathbb{R}^d} v_T(x)\,\mathrm{d}x - \frac{\epsilon^2}{2}\int_{\mathbb{R}^d}\big(\zeta_T(x) - \frac{v_T^2(x)}{p_0(x)}\big)\,\mathrm{d}x + \mathcal{O}(\epsilon^3).$$

Since $\int \widetilde{q}_T^\epsilon \equiv 1$, it is easy to know that $\int v_T = \int \zeta_T = 0$ and thus

$$\text{KL}(p_0||\widetilde{q}_T^\epsilon) = \frac{\epsilon^2}{2}\int_{\mathbb{R}^d}\frac{v_T^2(x)}{p_0(x)}\,\mathrm{d}x + \mathcal{O}(\epsilon^3).$$

Next we need to study $v_T$. Recall from (7) that the Fokker-Planck equation of $\widetilde{q}_t$ is

$$\partial_t \widetilde{q}_t^\epsilon = \mathcal{L}_t^{(h^\leftarrow)}(\widetilde{q}_t^\epsilon) - \epsilon \boldsymbol{\nabla}\cdot\big(\frac{1 + h_t^{\leftarrow 2}}{2}\mathscr{E}_t^\leftarrow \widetilde{q}_t^\epsilon\big).$$

By taking derivatives with respect to $\epsilon$ on both sides of this equation and then passing $\epsilon \to 0$,

$$\partial_t v_t = \mathcal{L}_t^{(h^\leftarrow)}(v_t) - \boldsymbol{\nabla}\cdot\big(\frac{1 + h_t^{\leftarrow 2}}{2}\mathscr{E}_t^\leftarrow q_t\big).$$

By Lem. C.2,

$$v_T = -\frac{1}{2}\int_0^T (1 + h_t^{\leftarrow 2})\Phi_{t,T}^{(h^\leftarrow)}\big(\boldsymbol{\nabla}\cdot\big(\underbrace{q_t}_{\equiv p_t^\leftarrow}\mathscr{E}_t^\leftarrow\big)\big)\,\mathrm{d}t. \tag{27}$$

## D   Preliminary results

We collect two lemmas which will be useful in proving Prop. 3.5 and Prop. 3.6 later.

**Lemma D.1.**

1. *The operator $\mathcal{K}(f) := \triangle f + \boldsymbol{\nabla}\cdot(\nabla V \cdot f)$ is a Hermitian/self-adjoint operator in the space $\langle\cdot,\cdot\rangle_{L^2(\rho^{-1})}$ where $\rho \propto e^{-V}$. Therefore, $\mathcal{K}$ has the eigen-decomposition in this weighted $L^2(\rho^{-1})$ space.*

2. *Assume that $\nabla^2 V(x) \geq m\boldsymbol{I}_d$ for any $x \in \mathbb{R}^d$ for some $m > 0$. Then the operator $-\mathcal{K}$ is a positive operator on the space $\{f : \int f = 0\}$ with spectrum gap at least $m$.*

*Proof.* For any $f, g$, note that

$$\langle f, \mathcal{K}g\rangle_{L^2(\rho^{-1})} = -\int_{\mathbb{R}^d}\nabla(f/\rho)\cdot\nabla(g/\rho)\rho.$$

Therefore, $\mathcal{K}$ is a Hermitian operator. In the space $\{f : \int f = 0\}$, we know

$$-\langle f, \mathcal{K}f\rangle_{L^2(\rho^{-1})} = \int_{\mathbb{R}^d}|\nabla(f/\rho)|^2\rho$$

$$\overset{\text{Poincaré ineq.}}{\geq} m\int_{\mathbb{R}^d}(f^2/\rho^2)\rho = m\langle f, f\rangle_{L^2(\rho^{-1})}.$$

The validity of Poincaré inequality under the strong convexity assumption is well-known in literature; see, e.g., [2, 24] and [3, Corollary 4.8.2]. $\qquad\square$

**Lemma D.2.** *Suppose the operator $\mathcal{K}$ is defined as $\mathcal{K}(f) := \triangle f + \boldsymbol{\nabla} \cdot \big(\nabla V \cdot (f)\big)$ with $\nabla^2 V(x) \geq m\boldsymbol{I}_d$ for any $x \in \mathbb{R}^d$. Then for any $c, a \in \mathbb{R}^+$, and any function $\varphi$ with $\int \varphi = 0$,*

$$\int_{\mathbb{R}^d} \frac{1}{\rho} \Big( \int_0^a \exp\big(ct\mathcal{K}\big)(\varphi) \, \mathrm{d}t \Big)^2 = \sum_{k=1}^{\infty} \big(\frac{1 - e^{-ac\lambda_k}}{c\lambda_k}\big)^2 \alpha_k^2$$

$$\leq \Big(\frac{1 - e^{-acm}}{cm}\Big)^2 \int_{\mathbb{R}^d} \frac{\varphi^2}{\rho},$$

*where $\rho \propto e^{-V}$, $\{(\lambda_k, \phi_k)\}_{k=1}^{\infty}$ are eigen pairs for the operator $-\mathcal{K}$, and $\varphi = \sum_{k=1}^{\infty} \alpha_k \phi_k$.*

**Remark.** Note that as $\mathcal{K}$ is an operator, $\exp\big(ct\mathcal{K}\big) := \sum_{k=0}^{\infty} \frac{(ct\mathcal{K})^k}{k!}$ is the operator exponential.

*Proof.* Let us first denote the eigenvalue decomposition of $\mathcal{K}$ as $\mathcal{K}(\phi_k) = -\lambda_k \phi_k$ where $\lambda_k \geq m$ for $k \in \mathbb{N}$ and $\langle \phi_j, \phi_k \rangle_{L^2(\rho^{-1})} = \delta_{j,k}$ for any $j, k \in \mathbb{N}$ by Lem. D.1. Then we can decompose $\varphi$ by $\varphi = \sum_k \alpha_k \phi_k$. It is not hard to verify that

$$\exp\big(ct\mathcal{K}\big)(\varphi) = \sum_{k=1}^{\infty} e^{-ct\lambda_k} \alpha_k \phi_k.$$

Then

$$\int_0^a \exp\big(ct\mathcal{K}\big)(\varphi) \, \mathrm{d}t = \sum_{k=1}^{\infty} \frac{1 - e^{-ac\lambda_k}}{c\lambda_k} \alpha_k \phi_k.$$

Hence,

$$\int_{\mathbb{R}^d} \frac{1}{\rho} \Big( \int_0^a \exp\big(ct\mathcal{K}\big)(\varphi) \, \mathrm{d}t \Big)^2 = \int_{\mathbb{R}^d} \frac{1}{\rho} \Big( \sum_{k=1}^{\infty} \frac{1 - e^{-ac\lambda_k}}{c\lambda_k} \alpha_k \phi_k \Big)^2$$

$$= \sum_{k=1}^{\infty} \big(\frac{1 - e^{-ac\lambda_k}}{c\lambda_k}\big)^2 \alpha_k^2$$

$$\leq \Big(\frac{1 - e^{-acm}}{cm}\Big)^2 \sum_{k=1}^{\infty} \alpha_k^2$$

$$= \Big(\frac{1 - e^{-acm}}{cm}\Big)^2 \int_{\mathbb{R}^d} \frac{\varphi^2}{\rho}.$$

$\square$

# E  Proof of Prop. 3.4, Prop. 3.5, and a discussion on the pulse-shape error

## E.1  Proof of Prop. 3.4

For convenience, we summarize some notations below; see also Appx. A.

- Denote the global minimum of $U_t^{\leftarrow}$ as $\mathcal{X}_t^{\leftarrow}$ and denote the global minimum of $V_t^{\leftarrow}$ as $\mathcal{Y}_t^{\leftarrow}$. When $\mathsf{h} \to \infty$, we know

$$\lim_{\mathsf{h} \to \infty} \mathcal{Y}_t \to \mathcal{X}_t.$$

- Denote the normalizing constant $Z_V := \int e^{-V}$ for an arbitrary potential $V$.
- Recall that the distribution of the exact dynamics $Y_t$ is $q_t \equiv p_{T-t} \equiv p_t^{\leftarrow}$ and the distribution of the approximated dynamics $\widetilde{Y}_t$ is $\widetilde{q}_t$.
- Recall that $p_t := e^{-U_t}$ and $\rho_t \propto e^{-V_t}$ in (12).

**Lemma E.1.** *Under Assumption 3.3, we have:*

*(i) Assumption 3.1 (3) is valid, i.e., the existence of $c_U$.*

*(ii) For any $t \in [0, T]$, the probability distribution $\rho_t$ defined in (12) satisfies the Poincaré inequality with constant $\kappa_t$ (16):*

$$\int_{\mathbb{R}^d} |\nabla\varphi|^2 \, \mathrm{d}\rho_t \geq \kappa_t \int_{\mathbb{R}^d} \varphi^2 \, \mathrm{d}\rho_t, \qquad \forall \varphi \text{ with } \int_{\mathbb{R}^d} \varphi\rho_t = 0. \tag{28}$$

*(iii) $Z_{V_0} := \int_{\mathbb{R}^d} e^{-V_0}$ is both upper and lower bounded: for any $\delta \in (0, 1)$, there exists $\mathsf{h}_0(\delta) > 0$ such that whenever $\mathsf{h} \geq \mathsf{h}_0(\delta)$,*

$$1 - \delta \leq Z_{V_0} \leq e^{-\mathsf{h}^{-2} c_U}.$$

*and*

$$\frac{\rho_0}{p_0} \leq \frac{e^{-\mathsf{h}^{-2} c_U}}{1 - \delta}.$$

*(iv) If we pick $\delta = 1/2$, for $\mathsf{h} \geq \max\{\mathsf{h}_0(1/2), \sqrt{\max\{0, -\frac{c_U}{\ln(2)}\}}\}$, the function $\rho_0/p_0$ is uniformly bounded:*

$$\frac{\rho_0}{p_0} \leq 4. \tag{29}$$

**Proposition E.2.** *For any $\mathsf{h} \geq 1/2$, we have*

$$\left| \int_{\mathbb{R}^d} \varphi^2 \frac{\mathrm{d}}{\mathrm{d}t} \rho_t^{\leftarrow} \right| \leq C_t^{\leftarrow,(1)} \int_{\mathbb{R}^d} \varphi^2 \, \rho_t^{\leftarrow} + C_t^{\leftarrow,(2)} \int_{\mathbb{R}^d} |\nabla\varphi|^2 \rho_t^{\leftarrow}, \qquad \forall \varphi \in C_0^2(\mathbb{R}^d).$$

*where*

$$\begin{cases} C_t^{\leftarrow,(1)} := \xi_t^{\leftarrow} + 2\varsigma_t^{\leftarrow} |\mathcal{Y}_t^{\leftarrow} - \mathcal{X}_t^{\leftarrow}|^2 + \dfrac{20 d M_t^{\leftarrow} \varsigma_t^{\leftarrow}}{m_t^{\leftarrow 2}}, \\[2mm] C_t^{\leftarrow,(2)} := \dfrac{8\varsigma_t^{\leftarrow}}{m_t^{\leftarrow 2}} \equiv \dfrac{20 + 30 M_t^{\leftarrow 2}}{m_t^{\leftarrow 2}}, \\[2mm] \xi_t^{\leftarrow} := \left| \partial_t \log Z_{V_t^{\leftarrow}} \right| + \dfrac{5d(1 + M_t^{\leftarrow})}{2} + \dfrac{5}{2} |\mathcal{X}_t^{\leftarrow}|^2, \\[2mm] \varsigma_t^{\leftarrow} := 5 \left( \dfrac{1}{2} + \dfrac{3 M_t^{\leftarrow 2}}{4} \right). \end{cases} \tag{30}$$

$Z_{V_t^{\leftarrow}} = \int e^{-V_t^{\leftarrow}}$, $M_t^{\leftarrow}$ and $m_t^{\leftarrow}$ are shown in **Assumption** *3.3*, and $\mathcal{X}_t^{\leftarrow}$ and $\mathcal{Y}_t^{\leftarrow}$ are the global minimum points of functions $x \mapsto U_t^{\leftarrow}(x)$ and $x \mapsto V_t^{\leftarrow}(x)$, respectively.

**Remark.** A similar result holds for any $\mathsf{h} > 0$; we choose $\mathsf{h} \geq 1/2$ in order to simplify the above constants (30).

Note that the constant $C_t^{\leftarrow,(2)}$ is independent of $h$. We remark that when $\mathsf{h} \gg 1$, $C_t^{\leftarrow,(1)}$ approximately behaves as follows:

**Lemma E.3.** *When $\mathsf{h} \to \infty$, we have*

$$\lim_{\mathsf{h} \to \infty} C_t^{\leftarrow,(1)} = \frac{5d(1 + M_t^{\leftarrow})}{2} + \frac{5}{2} |\mathcal{X}_t^{\leftarrow}|^2 + \frac{(50 + 75 M_t^{\leftarrow 2}) d M_t^{\leftarrow}}{m_t^{\leftarrow 2}}.$$

We now proceed to finish the proof of Prop. 3.4. The detailed proofs of Lem. E.1, Prop. E.2, and Lem. E.3 are postponed to Appx. E.2.

*Proof of Prop. 3.4.* By Lem. E.1,

$$L(\mathsf{h}) \overset{(13)}{=} \frac{(1 + \mathsf{h}^2)^2}{8} \int_{\mathbb{R}^d} \frac{\left( \Phi_{s,T}^{(h^{\leftarrow})} (\boldsymbol{\nabla} \cdot (p_s^{\leftarrow} E)) \right)^2}{\rho_0} \frac{\rho_0}{p_0}$$

$$\overset{(29)}{\leq} \frac{1}{2} (1 + \mathsf{h}^2)^2 \int_{\mathbb{R}^d} \frac{\left( \Phi_{s,T}^{(h^{\leftarrow})} (\boldsymbol{\nabla} \cdot (p_s^{\leftarrow} E)) \right)^2}{\rho_0}. \tag{31}$$

To simplify notations, let

$$\Lambda_t^{\leftarrow} := \Phi_{s,t}^{(h^{\leftarrow})}\big(\boldsymbol{\nabla} \cdot (p_s^{\leftarrow} E)\big), \qquad J(\mathsf{h}, t) := \int_{\mathbb{R}^d} \frac{\Lambda_t^{\leftarrow 2}}{\rho_t^{\leftarrow}}, \qquad t \in [s, T].$$

After taking the time-derivative and using Lemma C.1 (in the first line below), and expressions (11) in the second line below, and integration by parts in the third line, the expression of $\rho_t^{\leftarrow}$ (12) in the fourth line, we have

$$
\begin{aligned}
\frac{\mathrm{d}}{\mathrm{d}t} J(\mathsf{h}, t) =& 2\int_{\mathbb{R}^d} \frac{\Lambda_t^{\leftarrow} \mathcal{L}_t^{(h^{\leftarrow})}(\Lambda_t^{\leftarrow})}{\rho_t^{\leftarrow}} - \int_{\mathbb{R}^d} \frac{\Lambda_t^{\leftarrow 2}}{\left(\rho_t^{\leftarrow}\right)^2} \frac{\mathrm{d}}{\mathrm{d}t} \rho_t^{\leftarrow} \\
\overset{(11)}{=}& \mathsf{h}^2 \int_{\mathbb{R}^d} \frac{\Lambda_t^{\leftarrow}}{\rho_t^{\leftarrow}} \big( \triangle \Lambda_t^{\leftarrow} + \boldsymbol{\nabla} \cdot (\nabla V_t^{\leftarrow} \Lambda_t^{\leftarrow}) \big) - \int_{\mathbb{R}^d} \frac{\Lambda_t^{\leftarrow 2}}{\left(\rho_t^{\leftarrow}\right)^2} \frac{\mathrm{d}}{\mathrm{d}t} \rho_t^{\leftarrow} \\
=& -\mathsf{h}^2 \int_{\mathbb{R}^d} \nabla \frac{\Lambda_t^{\leftarrow}}{\rho_t^{\leftarrow}} \cdot \big(\nabla \Lambda_t^{\leftarrow} + \nabla V_t^{\leftarrow} \Lambda_t^{\leftarrow}\big) - \int_{\mathbb{R}^d} \frac{\Lambda_t^{\leftarrow 2}}{\left(\rho_t^{\leftarrow}\right)^2} \frac{\mathrm{d}}{\mathrm{d}t} \rho_t^{\leftarrow} \\
=& -\mathsf{h}^2 \int_{\mathbb{R}^d} \left| \nabla \frac{\Lambda_t^{\leftarrow}}{\rho_t^{\leftarrow}} \right|^2 \rho_t^{\leftarrow} - \int_{\mathbb{R}^d} \frac{\Lambda_t^{\leftarrow 2}}{\left(\rho_t^{\leftarrow}\right)^2} \frac{\mathrm{d}}{\mathrm{d}t} \rho_t^{\leftarrow}.
\end{aligned}
\tag{32}
$$

The major challenge is to estimate $\int_{\mathbb{R}^d} \frac{\Lambda_t^{\leftarrow 2}}{\left(\rho_t^{\leftarrow}\right)^2} \frac{\mathrm{d}}{\mathrm{d}t} \rho_t^{\leftarrow}$. By Prop. E.2,

$$
\begin{aligned}
\frac{\mathrm{d}}{\mathrm{d}t} J(\mathsf{h}, t) \leq& -(\mathsf{h}^2 - C_t^{\leftarrow,(2)}) \int_{\mathbb{R}^d} \left| \nabla \frac{\Lambda_t^{\leftarrow}}{\rho_t^{\leftarrow}} \right|^2 \mathrm{d}\rho_t^{\leftarrow} + C_t^{\leftarrow,(1)} \int_{\mathbb{R}^d} \big(\frac{\Lambda_t^{\leftarrow}}{\rho_t^{\leftarrow}}\big)^2 \mathrm{d}\rho_t^{\leftarrow} \\
\overset{(28)}{\leq}& \big( -(\mathsf{h}^2 - C_t^{\leftarrow,(2)})\kappa_t^{\leftarrow} + C_t^{\leftarrow,(1)} \big) \int_{\mathbb{R}^d} \big(\frac{\Lambda_t^{\leftarrow}}{\rho_t^{\leftarrow}}\big)^2 \mathrm{d}\rho_t^{\leftarrow}.
\end{aligned}
$$

To verify the condition in Prop. E.2, we can readily confirm that

$$\int \frac{\Lambda_t^{\leftarrow}}{\rho_t^{\leftarrow}} \rho_t^{\leftarrow} = \int \Lambda_t^{\leftarrow} = \int \Phi_{s,t}^{(h^{\leftarrow})}\big(\boldsymbol{\nabla} \cdot (p_s^{\leftarrow} E)\big) = \int \boldsymbol{\nabla} \cdot (p_s^{\leftarrow} E) = 0,$$

where the third equality in the last equation comes the fact that the Fokker-Planck solution operator $\Phi_{s,t}^{(h^{\leftarrow})}$ preserves the total mass. Note that we need $\mathsf{h}^2 - C_t^{\leftarrow,(2)} \geq 0$ in order to ensure the correct direction when applying the Poincaré inequality above, which explains the lower bound that $\mathsf{h}$ needs to satisfy in Prop. 3.4.

By Gröwnwall's inequality, for any $t \geq s$,

$$J(\mathsf{h}, t) \leq J(\mathsf{h}, s) \exp\Big( -\int_s^t \big((\mathsf{h}^2 - C_r^{\leftarrow,(2)})\kappa_r^{\leftarrow} - C_r^{\leftarrow,(1)}\big)\mathrm{d}r \Big).
\tag{33}$$

Finally, by (31), we have

$$L(\mathsf{h}) \leq \frac{(1+\mathsf{h}^2)^2}{2} J(\mathsf{h}, T) \leq C_{\mathsf{h}}(1+\mathsf{h}^2)^2 \exp\Big( -\int_s^T \big((\mathsf{h}^2 - C_r^{\leftarrow,(2)})\kappa_r^{\leftarrow} - C_r^{\leftarrow,(1)}\big)\mathrm{d}r \Big),$$

where

$$C_{\mathsf{h}} = \frac{1}{2} J(\mathsf{h}, s) = \frac{1}{2} \int_{\mathbb{R}^d} \frac{\big(\boldsymbol{\nabla} \cdot (p_s^{\leftarrow} E)\big)^2}{\rho_s^{\leftarrow}}.
\tag{34}$$

Recall that $C_t^{\leftarrow,(2)}$ (30) is independent of $\mathsf{h}$. By Lem. E.3, we already discussed that $\lim_{\mathsf{h}\to\infty} C_t^{\leftarrow,(1)}$ exists. When $\mathsf{h} \to \infty$, we know that $\rho_s^{\leftarrow} \to p_s^{\leftarrow}$ (11)(12), and thus,

$$\lim_{\mathsf{h}\to\infty} C_{\mathsf{h}} = \frac{1}{2} \int_{\mathbb{R}^d} \frac{\big(\boldsymbol{\nabla} \cdot (p_s^{\leftarrow} E)\big)^2}{p_s^{\leftarrow}}.$$

This completes the proof of Prop. 3.4. $\qquad\square$

### E.2 Proof of Lem. E.1, Prop. E.2, and Lem. E.3

*Proof of Lem. E.1.*

(i) From $m_0 > 1$ in **Assumption** 3.3, $x \mapsto U_0(x) - x^2/2$ is strongly convex and thus is surely bounded from below by some $c_U \in \mathbb{R}$.

(ii) The second result is a classical result by Bakry-Emery criterion [2, 3], as $V_t^{\leftarrow}$ (12) is strongly convex with Hessian lower bound $\kappa_t^{\leftarrow}$ (16).

(iii) To prove the third one, note that

$$Z_{V_0} := \int_{\mathbb{R}^d} e^{-V_0} \overset{(11)}{=} \int_{\mathbb{R}^d} e^{-(1+\mathsf{h}^{-2})U_0(x)+\mathsf{h}^{-2}|x|^2/2} \, \mathrm{d}x$$

$$= \int_{\mathbb{R}^d} e^{-U_0(x)} e^{-\mathsf{h}^{-2}(U_0(x)-|x|^2/2)} \, \mathrm{d}x$$

$$\leq e^{-\mathsf{h}^{-2}c_U} \int_{\mathbb{R}^d} e^{-U_0} = e^{-\mathsf{h}^{-2}c_U}.$$

To prove the lower bound,

$$Z_{V_0} \geq \int_{\mathbb{R}^d} e^{-(1+\mathsf{h}^{-2})U_0(x)} \, \mathrm{d}x = e^{-(1+\mathsf{h}^{-2})U_0(\mathcal{X}_0)} \int_{\mathbb{R}^d} e^{-(1+\mathsf{h}^{-2})\left(U_0(x)-U_0(\mathcal{X}_0)\right)} \, \mathrm{d}x. \quad (35)$$

Recall from the beginning of this Appendix, $\mathcal{X}_0$ is defined as the global minimum of $x \mapsto U_0(x)$. Then, $U_0(x) \geq U_0(\mathcal{X}_0)$ for any $x$ and we know $\mathsf{h} \mapsto e^{-(1+\mathsf{h}^{-2})\left(U_0(x)-U_0(\mathcal{X}_0)\right)}$ is monotone increasing. By monotone convergence theorem,

$$\lim_{\mathsf{h}\to\infty} \int_{\mathbb{R}^d} e^{-(1+\mathsf{h}^{-2})\left(U_0(x)-U_0(\mathcal{X}_0)\right)} \, \mathrm{d}x = \int_{\mathbb{R}^d} \lim_{\mathsf{h}\to\infty} e^{-(1+\mathsf{h}^{-2})\left(U_0(x)-U_0(\mathcal{X}_0)\right)} \, \mathrm{d}x$$

$$= \int_{\mathbb{R}^d} e^{-U_0(x)+U_0(\mathcal{X}_0)} \, \mathrm{d}x = e^{U_0(\mathcal{X}_0)}.$$

Thus, the limit of the right-hand side of (35) is 1 when $\mathsf{h} \to \infty$. Hence, the lower bound of $Z_{V_0}$ follows immediately. Finally, when $\mathsf{h} \geq \mathsf{h}_0(\delta)$,

$$\frac{\rho_0}{p_0}(x) = \frac{e^{-(1+\mathsf{h}^{-2})U_0(x)+\mathsf{h}^{-2}|x|^2/2}}{Z_{V_0}e^{-U_0}} \leq \frac{e^{-\mathsf{h}^{-2}c_U}}{1-\delta}. \quad (36)$$

(iv) When we pick $\delta = 1/2$ and choose $\mathsf{h}$ as specified, we can immediately obtain (29).

$\square$

Before we prove Prop. E.2, we need the following three lemmas.

**Lemma E.4.** *Under* **Assumption** *3.3, we have*

$$|\partial_t U_t(x)| \leq \frac{d(1+M_t)}{2} + \frac{|x|^2}{4} + \frac{3M_t^2}{4}|x - \mathcal{X}_t|^2.$$

*Proof.* Since $p_t$ follows the Fokker-Planck equation $\partial_t p_t = \nabla \cdot \left(\frac{1}{2}xp_t\right) + \frac{1}{2}\triangle p_t$, then $U_t = -\log p_t$ satisfies $\partial_t U_t(x) = -\frac{1}{2}\left(d - x \cdot \nabla U_t(x) - \triangle U_t(x) + |\nabla U_t(x)|^2\right)$. Then by **Assumption** 3.3,

$$|\partial_t U_t(x)| \leq \frac{d}{2} + \frac{1}{2}|x \cdot \nabla U_t(x)| + \frac{1}{2}\triangle U_t(x) + \frac{1}{2}|\nabla U_t(x)|^2 \quad \text{(triangle inequality)}$$

$$\leq \frac{d}{2} + \frac{1}{4}\left(|x|^2 + |\nabla U_t(x)|^2\right) + \frac{1}{2}\triangle U_t(x) + \frac{1}{2}|\nabla U_t(x)|^2 \quad \text{(Cauchy inequality)}$$

$$\leq \frac{d}{2} + \frac{1}{4}|x|^2 + \frac{3}{4}|\nabla U_t(x) - \nabla U_t(\mathcal{X}_t)|^2 + \frac{dM_t}{2} \quad \text{(by } \nabla U_t(\mathcal{X}_t) \equiv 0)$$

$$\leq \frac{d(1+M_t)}{2} + \frac{|x|^2}{4} + \frac{3M_t^2}{4}|x - \mathcal{X}_t|^2.$$

Recall that $\mathcal{X}_t$ is defined as the global minimum of $U_t$ and thus $\nabla U_t(\mathcal{X}_t) = 0$. $\square$

**Lemma E.5.** *When* $\mathsf{h} \geq \frac{1}{2}$,

$$\frac{\left|\frac{\mathrm{d}}{\mathrm{d}t}\rho_t^{\leftarrow}\right|}{\rho_t^{\leftarrow}}(x) \leq \xi_t^{\leftarrow} + \varsigma_t^{\leftarrow}|x - \mathcal{X}_t^{\leftarrow}|^2, \tag{37}$$

*where*

$$\begin{cases} \xi_t^{\leftarrow} := \left|\partial_t \log Z_{V_t^{\leftarrow}}\right| + \dfrac{5d(1 + M_t^{\leftarrow})}{2} + \dfrac{5}{2}|\mathcal{X}_t^{\leftarrow}|^2, \\[2mm] \varsigma_t^{\leftarrow} := 5\left(\dfrac{1}{2} + \dfrac{3M_t^{\leftarrow 2}}{4}\right). \end{cases}$$

*Proof.* By direct calculation from the definitions of $\rho_t^{\leftarrow}$ in (12) and $V_t^{\leftarrow}$ in (11),

$$\partial_t \rho_t^{\leftarrow} = -(\partial_t \log Z_{V_t^{\leftarrow}})\rho_t^{\leftarrow} - (1 + \mathsf{h}^{-2})\rho_t^{\leftarrow}\partial_t U_t^{\leftarrow}.$$

Hence, by Lem. E.4 and $\mathsf{h} \geq \frac{1}{2}$,

$$\begin{aligned} \frac{\left|\partial_t \rho_t^{\leftarrow}\right|}{\rho_t^{\leftarrow}}(x) &\leq \left|\partial_t \log Z_{V_t^{\leftarrow}}\right| + (1 + \mathsf{h}^{-2})\left(\frac{d(1 + M_t^{\leftarrow})}{2} + \frac{|x|^2}{4} + \frac{3M_t^{\leftarrow 2}}{4}|x - \mathcal{X}_t^{\leftarrow}|^2\right) \\ &\leq \left(\left|\partial_t \log Z_{V_t^{\leftarrow}}\right| + \frac{5d(1 + M_t^{\leftarrow})}{2}\right) \\ &\quad + 5\left(\frac{|\mathcal{X}_t^{\leftarrow}|^2}{2} + \frac{|x - \mathcal{X}_t^{\leftarrow}|^2}{2} + \frac{3M_t^{\leftarrow 2}}{4}|x - \mathcal{X}_t^{\leftarrow}|^2\right) \\ &= \xi_t^{\leftarrow} + 5\left(\frac{1}{2} + \frac{3M_t^{\leftarrow 2}}{4}\right)|x - \mathcal{X}_t^{\leftarrow}|^2. \end{aligned}$$

$\square$

**Lemma E.6.** *Assume that $\rho \propto e^{-V}$ and $\varphi$ decays fast enough such that*

$$\int_{B_0(R)} \boldsymbol{\nabla} \cdot \left(\varphi^2 \nabla \rho\right) \to 0,$$

*as the radius $R \to \infty$ ($B_0(R)$ is the ball centered at 0 with radius R), e.g., $\varphi \in C_0^2(\mathbb{R}^d)$. Then*

$$\int_{\mathbb{R}^d} |\nabla V|^2 \varphi^2 \rho \leq 4 \int_{\mathbb{R}^d} |\nabla \varphi|^2 \rho + 2 \int_{\mathbb{R}^d} \varphi^2 \triangle V \rho. \tag{38}$$

*Proof.* The identity $\boldsymbol{\nabla} \cdot \left(\varphi^2 \nabla \rho\right) = \nabla \varphi^2 \cdot \nabla \rho + \varphi^2 \triangle \rho$ and the trivial facts $\nabla \rho = -\rho \nabla V$, $\triangle \rho = (|\nabla V|^2 - \triangle V)\rho$ show that

$$\begin{aligned} 0 &= \int \nabla \varphi^2 \cdot \nabla \rho + \varphi^2 \triangle \rho \\ &= -2\int (\nabla V \cdot \nabla \varphi)\varphi \rho + \int \varphi^2(-\triangle V + |\nabla V|^2)\rho \\ &\overset{\text{Cauchy ineq.}}{\geq} -\frac{1}{2}\int |\nabla V|^2 \varphi^2 \rho - 2\int |\nabla \varphi|^2 \rho - \int \varphi^2 \triangle V \rho + \int |\nabla V|^2 \varphi^2 \rho \\ &= \frac{1}{2}\int |\nabla V|^2 \varphi^2 \rho - 2\int |\nabla \varphi|^2 \rho - \int \varphi^2 \triangle V \rho. \end{aligned}$$

Therefore,

$$\int |\nabla V|^2 \varphi^2 \rho \leq 4 \int |\nabla \varphi|^2 \rho + 2 \int \varphi^2 \triangle V \rho.$$

$\square$

*Proof of Prop. E.2.* The above three lemmas show that

$$\left|\int_{\mathbb{R}^d}\varphi^2\frac{\mathrm{d}}{\mathrm{d}t}\rho_t^{\leftarrow}\right|\overset{(37)}{\leq}\int_{\mathbb{R}^d}\varphi^2(\xi_t^{\leftarrow}+\varsigma_t^{\leftarrow}|x-\mathcal{X}_t^{\leftarrow}|^2)\rho_t^{\leftarrow}$$

$$=\xi_t^{\leftarrow}\int_{\mathbb{R}^d}\varphi^2\,\rho_t^{\leftarrow}+\varsigma_t^{\leftarrow}\int_{\mathbb{R}^d}\varphi^2|x-\mathcal{Y}_t^{\leftarrow}+\mathcal{Y}_t^{\leftarrow}-\mathcal{X}_t^{\leftarrow}|^2\rho_t^{\leftarrow}$$

$$\overset{\text{Cauchy ineq.}}{\leq}\left(\xi_t^{\leftarrow}+2\varsigma_t^{\leftarrow}|\mathcal{Y}_t^{\leftarrow}-\mathcal{X}_t^{\leftarrow}|^2\right)\int_{\mathbb{R}^d}\varphi^2\rho_t^{\leftarrow}+2\varsigma_t^{\leftarrow}\int_{\mathbb{R}^d}\varphi^2|x-\mathcal{Y}_t^{\leftarrow}|^2\rho_t^{\leftarrow}$$

$$\overset{(14)}{\leq}\left(\xi_t^{\leftarrow}+2\varsigma_t^{\leftarrow}|\mathcal{Y}_t^{\leftarrow}-\mathcal{X}_t^{\leftarrow}|^2\right)\int_{\mathbb{R}^d}\varphi^2\rho_t^{\leftarrow}+\frac{2\varsigma_t^{\leftarrow}}{m_t^{\leftarrow 2}}\int_{\mathbb{R}^d}\varphi^2|\nabla V_t^{\leftarrow}(x)-\nabla V_t^{\leftarrow}(\mathcal{Y}_t^{\leftarrow})|^2\rho_t^{\leftarrow}$$

$$=\left(\xi_t^{\leftarrow}+2\varsigma_t^{\leftarrow}|\mathcal{Y}_t^{\leftarrow}-\mathcal{X}_t^{\leftarrow}|^2\right)\int_{\mathbb{R}^d}\varphi^2\rho_t^{\leftarrow}+\frac{2\varsigma_t^{\leftarrow}}{m_t^{\leftarrow 2}}\int_{\mathbb{R}^d}\varphi^2|\nabla V_t^{\leftarrow}(x)|^2\rho_t^{\leftarrow}$$

$$\overset{(38)}{\leq}\left(\xi_t^{\leftarrow}+2\varsigma_t^{\leftarrow}|\mathcal{Y}_t^{\leftarrow}-\mathcal{X}_t^{\leftarrow}|^2\right)\int_{\mathbb{R}^d}\varphi^2\rho_t^{\leftarrow}+\frac{2\varsigma_t^{\leftarrow}}{m_t^{\leftarrow 2}}\left(4\int_{\mathbb{R}^d}|\nabla\varphi|^2\rho_t^{\leftarrow}+2\int_{\mathbb{R}^d}\varphi^2\triangle V_t^{\leftarrow}\rho_t^{\leftarrow}\right)$$

$$\overset{(14)}{\leq}\left(\xi_t^{\leftarrow}+2\varsigma_t^{\leftarrow}|\mathcal{Y}_t^{\leftarrow}-\mathcal{X}_t^{\leftarrow}|^2\right)\int_{\mathbb{R}^d}\varphi^2\rho_t^{\leftarrow}+\frac{2\varsigma_t^{\leftarrow}}{m_t^{\leftarrow 2}}\left(4\int_{\mathbb{R}^d}|\nabla\varphi|^2\rho_t^{\leftarrow}+10M_t^{\leftarrow}d\int_{\mathbb{R}^d}\varphi^2\rho_t^{\leftarrow}\right)$$

$$=\left(\xi_t^{\leftarrow}+2\varsigma_t^{\leftarrow}|\mathcal{Y}_t^{\leftarrow}-\mathcal{X}_t^{\leftarrow}|^2+\frac{20M_t^{\leftarrow}d\varsigma_t^{\leftarrow}}{m_t^{\leftarrow 2}}\right)\int_{\mathbb{R}^d}\varphi^2\rho_t^{\leftarrow}+\frac{8\varsigma_t^{\leftarrow}}{m_t^{\leftarrow 2}}\int_{\mathbb{R}^d}|\nabla\varphi|^2\rho_t^{\leftarrow}.$$

To get the fifth line, we used the definition of $\mathcal{Y}_t^{\leftarrow}$: $\mathcal{Y}_t^{\leftarrow}$ is the global minimum of $V_t^{\leftarrow}$ so that $\nabla V_t^{\leftarrow}(\mathcal{Y}_t^{\leftarrow})\equiv 0$. To get the second last line, we used the fact that

$$\triangle V_t^{\leftarrow}=(1+\mathsf{h}^{-2})\triangle U_t^{\leftarrow}-\mathsf{h}^{-2}\leq (1+\mathsf{h}^{-2})\,dM_t^{\leftarrow}-\mathsf{h}^{-2}\overset{(\text{by }\mathsf{h}\geq 1/2)}{\leq}5dM_t^{\leftarrow}.$$

$\square$

*Proof of Lem. E.3.* Note that $\varsigma_t^{\leftarrow}$ does not depend on $\mathsf{h}$. When $\mathsf{h}\to\infty$, as $V_t^{\leftarrow}\to U_t^{\leftarrow}$ (11), we know $Z_{V_t^{\leftarrow}}$ converges to $Z_{U_t^{\leftarrow}}\equiv 1$ by the definition of $p_t:=e^{-U_t}$, we immediately know that

$$\lim_{\mathsf{h}\to\infty}\left|\partial_t\log Z_{V_t^{\leftarrow}}\right|=0.$$

Then

$$\lim_{\mathsf{h}\to\infty}\xi_t^{\leftarrow}=\frac{5d(1+M_t^{\leftarrow})}{2}+\frac{5}{2}|\mathcal{X}_t^{\leftarrow}|^2.$$

Moreover, $\mathcal{Y}_t\to\mathcal{X}_t$ as $\mathsf{h}\to\infty$, we have

$$\lim_{\mathsf{h}\to\infty}C_t^{\leftarrow,(1)}=\lim_{\mathsf{h}\to\infty}\xi_t^{\leftarrow}+2\lim_{\mathsf{h}\to\infty}\varsigma_t^{\leftarrow}|\mathcal{Y}_t^{\leftarrow}-\mathcal{X}_t^{\leftarrow}|^2+\frac{20dM_t^{\leftarrow}\varsigma_t^{\leftarrow}}{m_t^{\leftarrow 2}},$$

$$=\frac{5d(1+M_t^{\leftarrow})}{2}+\frac{5}{2}|\mathcal{X}_t^{\leftarrow}|^2+\frac{(50+75M_t^{\leftarrow 2})dM_t^{\leftarrow}}{m_t^{\leftarrow 2}}.$$

$\square$

## E.3    Proof of Prop. 3.5

**Case I:** $\mathsf{h}=0$. For this ODE case, (8) becomes

$$\mathcal{L}_t^{(0)}(\mu)(x)=\frac{1}{2}\boldsymbol{\nabla}\cdot\left(\nabla\big(U_t^{\leftarrow}(x)-|x|^2/2\big)\mu(x)\right),\tag{39}$$

and the formula of leading order term $L(0)$ is

$$L(0)=\frac{1}{2}\int_{\mathbb{R}^d}\frac{v_T^2(x)}{p_0(x)}\,\mathrm{d}x,\qquad v_T\overset{(10)}{=}-\frac{1}{2}\int_{T-a}^T\Phi_{t,T}^{(0)}\big(\boldsymbol{\nabla}\cdot(p_t^{\leftarrow}E)\big)\,\mathrm{d}t,$$

since we perturb the $\mathscr{E}_t^{\leftarrow}$ only when $t \approx T$, that is, $\mathscr{E}_t^{\leftarrow}(x) = \mathbb{I}_{t \in [T-a,T]} E(x)$ for a small positive $a$. Then we know that

$$v_T \sim -\frac{1}{2} \int_{T-a}^{T} \boldsymbol{\nabla} \cdot (p_0 E)) \, \mathrm{d}t = -\frac{a}{2} \boldsymbol{\nabla} \cdot (p_0 E).$$

Recall that $p_0 = p_T^{\leftarrow} \approx p_t^{\leftarrow}$ when $t \approx T$; moreover, $\Phi_{t,T}^{(0)} \approx \Phi_{T,T}^{(0)} \equiv \mathrm{Id}$. Therefore, the leading order term of $L(0)$ when $a \ll 1$ is

$$L(0) \sim \frac{a^2}{8} \int_{\mathbb{R}^d} \frac{(\boldsymbol{\nabla} \cdot (p_0 E))^2}{p_0}.$$

**Case II: $\mathsf{h} \to \infty$.** For the SDE case,

$$
\begin{aligned}
v_T &\overset{(10)}{=} -\frac{1+\mathsf{h}^2}{2} \int_0^T \Phi_{t,T}^{(\mathsf{h})}\big(\boldsymbol{\nabla} \cdot (p_t^{\leftarrow} \mathscr{E}_t^{\leftarrow})\big) \, \mathrm{d}t \\
&\sim -\frac{1+\mathsf{h}^2}{2} \int_{T-a}^T \Phi_{t,T}^{(\mathsf{h})}\big(\boldsymbol{\nabla} \cdot (p_0 E)\big) \, \mathrm{d}t \qquad \text{(by } a \ll 1) \\
&\sim -\frac{1+\mathsf{h}^2}{2} \int_{T-a}^T e^{(T-t)\frac{\mathsf{h}^2}{2}\mathcal{K}_T^{\leftarrow}}\big(\boldsymbol{\nabla} \cdot (p_0 E)\big) \, \mathrm{d}t \qquad \text{(by } a \ll 1) \\
&= -\frac{1+\mathsf{h}^2}{2} \int_0^a e^{t\frac{\mathsf{h}^2}{2}\mathcal{K}_T^{\leftarrow}}\big(\boldsymbol{\nabla} \cdot (p_0 E)\big) \, \mathrm{d}t,
\end{aligned}
$$

where

$$\mathcal{K}_t^{\leftarrow}(\mu) := \triangle\mu + \boldsymbol{\nabla} \cdot (\nabla V_t^{\leftarrow} \mu). \tag{40}$$

Then when we further assume $\mathsf{h} \gg 1$,

$$
\begin{aligned}
\frac{1}{2} \int_{\mathbb{R}^d} \frac{(v_T)^2}{p_0} &\lesssim \frac{1}{2} \int_{\mathbb{R}^d} \frac{(v_T)^2}{\rho_0} \qquad \text{(by Lem. E.1 (iii))} \\
&\lesssim \frac{(1+\mathsf{h}^2)^2}{8}\left(\frac{1-e^{-a\frac{\mathsf{h}^2}{2}\kappa_0}}{\frac{\mathsf{h}^2}{2}\kappa_0}\right)^2 \int_{\mathbb{R}^d} \frac{(\boldsymbol{\nabla} \cdot (p_0 E))^2}{\rho_0} \qquad \text{(by Lem. D.2)} \\
&\sim \frac{(1+\mathsf{h}^2)^2}{\mathsf{h}^4} \frac{\left(1-e^{-a\frac{\mathsf{h}^2}{2}\kappa_0}\right)^2}{2\kappa_0^2} \int_{\mathbb{R}^d} \frac{(\boldsymbol{\nabla} \cdot (p_0 E))^2}{\rho_0} \\
&\sim \frac{\left(1-e^{-a\frac{\mathsf{h}^2}{2}\kappa_0}\right)^2}{2\kappa_0^2} \int_{\mathbb{R}^d} \frac{(\boldsymbol{\nabla} \cdot (p_0 E))^2}{\rho_0} \\
&\sim \frac{\left(1-e^{-a\frac{\mathsf{h}^2}{2}\kappa_0}\right)^2}{2\kappa_0^2} \int_{\mathbb{R}^d} \frac{(\boldsymbol{\nabla} \cdot (p_0 E))^2}{p_0}.
\end{aligned}
$$

To apply Lem. D.2 in the second line, we remark that $\mathcal{K}_T^{\leftarrow}(\mu) = \triangle\mu + \boldsymbol{\nabla} \cdot (\nabla V_T^{\leftarrow} \mu)$, $V_T^{\leftarrow}$ has Hessian lower bound $\kappa_T^{\leftarrow} \equiv \kappa_0$ (16).

### E.4 Discussion on the pulse-shape error

Though the error of the score function is time-dependent, we can always divide the error function into a linear combination of step functions:

$$\mathscr{E}_t^{\leftarrow}(x) \approx \sum_j \phi_j(x)\chi_{[t_j,t_{j+1}]}(t),$$

where $\chi$ is the indicator function and $\phi_j$ is the function value of $\mathcal{E}_t^{\leftarrow}$ on the time interval $[t_j, t_{j+1}]$. Without loss of generality, assume that the time-discretization is uniform, i.e., $t_{j+1} - t_j = \Delta t$ for any $j$. If we don't worry about ill-behaved functions, this decomposition can be made arbitrarily accurate by choosing a smaller time interval; it is not difficult to make this approximation mathematically rigorous. It is hard to assume that this error function at time e.g., $t = 0.1$ has some subtle connection

with its error at e.g., $t = 0.8$ in general; how the error function exactly looks like depend on a vast amount of hyper-parameters in training. Therefore, we might as well treat each $\phi_j$ as "mostly independent" in order to handle the *worst case* situation. If we consider how each $\phi_j$ contributes to the final sample generation error (quantified via KL divergence), then we might as well study each one independently. We remark that this is simply a reasonable assumption in order to treat generic error types.

More specifically, recall that in Prop. 3.2, $v_T$ linearly depends on each $\phi_j$

$$
\begin{aligned}
v_T &= -\frac{1}{2} \int_0^T (1 + (h_t^{\leftarrow})^2) \Phi_{t,T}^{(h^{\leftarrow})} \left( \boldsymbol{\nabla} \cdot (p_t^{\leftarrow} \mathscr{E}_t^{\leftarrow}) \right) \, \mathrm{d}t \\
&\approx -\frac{1}{2} \sum_j (1 + (h_{t_j}^{\leftarrow})^2) \Phi_{t_j,T}^{(h^{\leftarrow})} \left( \boldsymbol{\nabla} \cdot \left( p_{t_j}^{\leftarrow} \phi_j \right) \right) \Delta t,
\end{aligned}
$$

and recall that the leading order term (10) is simply

$$
L(h^{\leftarrow}) = \frac{1}{2} \int_{\mathbb{R}^d} \frac{v_T^2(x)}{p_0(x)} \, \mathrm{d}x.
$$

Therefore, after plugging the decomposition of $\mathscr{E}_t^{\circ \leftarrow}$ into $L(h^{\leftarrow})$, we have

$$
L(h^{\leftarrow}) = \frac{(1 + \mathsf{h}^2)^2}{8} (\Delta t)^2 \sum_{j,k} \int_{\mathbb{R}^d} \frac{\Phi_{t_j,T}^{(h^{\leftarrow})} \left( \boldsymbol{\nabla} \cdot \left( p_{t_j}^{\leftarrow} \phi_j \right) \right) \Phi_{t_k,T}^{(h^{\leftarrow})} \left( \boldsymbol{\nabla} \cdot \left( p_{t_k}^{\leftarrow} \phi_k \right) \right)}{p_0}.
$$

We had chosen $h_t^{\leftarrow} = \mathsf{h}$ for all $t \in [0, T]$ for simplicity. Either by Cauchy-Schwartz inequality, or by assumptions on the independence of each $\phi_j$ as discussed above, the main feature is the following term

$$
\frac{(1 + \mathsf{h}^2)^2}{8} \Delta t \int_{\mathbb{R}^d} \frac{1}{p_0} \left[ \Phi_{t_j,T}^{(h^{\leftarrow})} \left( \boldsymbol{\nabla} \cdot \left( p_{t_j}^{\leftarrow} \phi_j \right) \right) \right]^2.
$$

Therefore, it makes sense to study how this quantity scales for each $j$, and for two asymptotic regions $\mathsf{h} = 0$ and $\mathsf{h} \to \infty$. For each fixed $j$, we can easily observe that the above quantity arises from choosing the error function as a pulse-shape error, i.e., $\phi_k = 0$ if $k \neq j$ (with certain normalization rescaling).

# F  Proof of Prop. 3.6 and the discussion of the large diffusion limit

## F.1  Proof of Prop. 3.6

Let us pick an $a = \mathsf{h}^{-\beta} \ll 1$. We split $v_T$ (10) into two parts:

$$
\begin{aligned}
v_T &= -\frac{1 + \mathsf{h}^2}{2} \int_0^T \Phi_{t,T}^{(h^{\leftarrow})} \big( \underbrace{\boldsymbol{\nabla} \cdot (p_t^{\leftarrow} \mathscr{E}_t^{\leftarrow})}_{=:\Gamma_t^{\leftarrow}} \big) \, \mathrm{d}t \\
&= -\frac{1 + \mathsf{h}^2}{2} \int_{T-a}^T \Phi_{t,T}^{(h^{\leftarrow})} \big( \Gamma_t^{\leftarrow} \big) \, \mathrm{d}t - \frac{1 + \mathsf{h}^2}{2} \int_0^{T-a} \Phi_{t,T}^{(h^{\leftarrow})} \big( \Gamma_t^{\leftarrow} \big) \, \mathrm{d}t.
\end{aligned}
$$

**Upper bound**

By Cauchy-Schwartz inequality ($(x + y)^2 = x^2 + y^2 + 2xy \leq (1 + \alpha^2)x^2 + (1 + \alpha^{-2})y^2$ for any $x, y$ and $\alpha > 0$),

$$L(\mathsf{h}) = \frac{1}{2} \int \frac{v_T^2}{p_0}$$

$$\leq (1 + \alpha^2) \underbrace{\frac{(1 + \mathsf{h}^2)^2}{8} \int_{\mathbb{R}^d} \frac{\left( \int_{T-a}^T \Phi_{t,T}^{(\mathsf{h}^\leftarrow)}(\Gamma_t^\leftarrow) \mathrm{d}t \right)^2}{p_0}}_{=: \mathcal{T}_1}$$

$$+ (1 + \alpha^{-2}) \underbrace{\frac{(1 + \mathsf{h}^2)^2}{8} \int_{\mathbb{R}^d} \frac{\left( \int_0^{T-a} \Phi_{t,T}^{(\mathsf{h}^\leftarrow)}(\Gamma_t^\leftarrow) \mathrm{d}t \right)^2}{p_0}}_{=: \mathcal{T}_2}.$$

For the second term $\mathcal{T}_2$,

$$\mathcal{T}_2 = \frac{(1 + \mathsf{h}^2)^2}{8} \int_{\mathbb{R}^d} \frac{\left( \int_0^{T-a} \Phi_{t,T}^{(\mathsf{h}^\leftarrow)}(\Gamma_t^\leftarrow) \mathrm{d}t \right)^2}{p_0}$$

$$\overset{\text{Cauchy ineq.}}{\leq} \frac{(1 + \mathsf{h}^2)^2 (T - a)}{8} \int_0^{T-a} \left( \int_{\mathbb{R}^d} \frac{\Phi_{t,T}^{(\mathsf{h}^\leftarrow)}(\Gamma_t^\leftarrow)^2}{p_0} \right) \mathrm{d}t$$

$$\overset{(29)}{\leq} \frac{(1 + \mathsf{h}^2)^2 T}{2} \int_0^{T-a} \left( \int_{\mathbb{R}^d} \frac{\Phi_{t,T}^{(\mathsf{h}^\leftarrow)}(\Gamma_t^\leftarrow)^2}{\rho_0} \right) \mathrm{d}t$$

$$\overset{(33)}{\leq} \frac{(1 + \mathsf{h}^2)^2 T}{2} \int_0^{T-a} \left( \int_{\mathbb{R}^d} \frac{\Gamma_t^{\leftarrow 2}}{\rho_t^\leftarrow} \right) \exp \left( - \int_t^T \left( (\mathsf{h}^2 - C_r^{\leftarrow,(2)}) \kappa_r^\leftarrow - C_r^{\leftarrow,(1)} \right) \mathrm{d}r \right) \mathrm{d}t$$

$$\leq \frac{(1 + \mathsf{h}^2)^2 T}{2} \sup_{t \in [0,T]} \left( \int_{\mathbb{R}^d} \frac{\Gamma_t^{\leftarrow 2}}{\rho_t^\leftarrow} \right) \exp \left( \int_0^T C_r^{\leftarrow,(2)} \kappa_r^\leftarrow + C_r^{\leftarrow,(1)} \mathrm{d}r \right) \int_0^{T-a} e^{-\mathsf{h}^2 \gamma (T-t)} \, \mathrm{d}t$$

$$\leq \frac{(1 + \mathsf{h}^2) T}{2\mathsf{h}^2} \sup_{t \in [0,T]} \left( \int_{\mathbb{R}^d} \frac{\Gamma_t^{\leftarrow 2}}{\rho_t^\leftarrow} \right) \exp \left( \int_0^T C_r^{\leftarrow,(2)} \kappa_r^\leftarrow + C_r^{\leftarrow,(1)} \mathrm{d}r \right) \frac{(1 + \mathsf{h}^2) e^{-a\mathsf{h}^2 \gamma}}{\gamma},$$

which decays exponentially fast as $\mathsf{h} \to \infty$ as long as $\mathsf{h}^2 \gg a^{-1} \gg 1$. To get the second line above, we used Cauchy-Schwarz inequality and Fubini's theorem. Recall the definition of $\gamma \in \mathbb{R}^+$ in the statement of Prop. 3.6.

Overall, when $a = \mathsf{h}^{-\beta} \ll 1$,

$$L(\mathsf{h}) \leq (1 + \alpha^{-2}) C \frac{(1 + \mathsf{h}^2) \exp \left( -\mathsf{h}^{2-\beta} \gamma \right)}{\gamma} + (1 + \alpha^2) \mathcal{T}_1,$$

where

$$C = \frac{(1 + \mathsf{h}^2)}{2\mathsf{h}^2} T \sup_{t \in [0,T]} \left( \int_{\mathbb{R}^d} \frac{\Gamma_t^{\leftarrow 2}}{\rho_t^\leftarrow} \right) \exp \left( \int_0^T C_r^{\leftarrow,(2)} \kappa_r^\leftarrow + C_r^{\leftarrow,(1)} \mathrm{d}r \right)$$

$$\lesssim \frac{T}{2} \sup_{t \in [0,T]} \left( \int_{\mathbb{R}^d} \frac{\Gamma_t^{\leftarrow 2}}{p_t^\leftarrow} \right) \exp \left( \int_0^T C_r^{\leftarrow,(2)} m_r^\leftarrow + \left( \lim_{\mathsf{h} \to \infty} C_r^{\leftarrow,(1)} \right) \mathrm{d}r \right).$$

Please refer to Lem. E.3 for $\lim_{\mathsf{h} \to \infty} C_r^{\leftarrow,(1)}$.

**Lower bound**

The lower bound can be proved in the same way: by Cauchy-Schwarz inequality again,

$$L(\mathsf{h}) \geq (1 - \alpha^2) \mathcal{T}_1 + (1 - \alpha^{-2}) \mathcal{T}_2$$

$$\geq (1 - \alpha^2) \mathcal{T}_1 - (\alpha^{-2} - 1) C \frac{(1 + \mathsf{h}^2) \exp \left( -\mathsf{h}^{2-\beta} \gamma \right)}{\gamma}.$$

The remaining task is to estimate $\mathcal{T}_1$.

**Asymptotic limit of $\mathcal{T}_1$**

We only provide an asymptotic result below. As $a \ll 1$,

$$
\begin{aligned}
\mathcal{T}_1 &:= \frac{(1+\mathsf{h}^2)^2}{8} \int_{\mathbb{R}^d} \frac{\big(\int_{T-a}^T \Phi_{t,T}^{(h^\leftarrow)}(\Gamma_t^\leftarrow) \mathrm{d}t\big)^2}{p_0} \\
&\sim \frac{(1+\mathsf{h}^2)^2}{8} \int_{\mathbb{R}^d} \frac{1}{p_0} \Big(\big(\int_{T-a}^T \Phi_{t,T}^{(h^\leftarrow)} \, \mathrm{d}t\big)(\Gamma_T^\leftarrow)\Big)^2 \\
&\sim \frac{(1+\mathsf{h}^2)^2}{8} \int_{\mathbb{R}^d} \frac{1}{p_0} \Big(\int_{T-a}^T \exp\big(\frac{\mathsf{h}^2}{2}(T-t)\mathcal{K}_T^\leftarrow\big) \Gamma_T^\leftarrow \, \mathrm{d}t\Big)^2 \\
&\sim \frac{(1+\mathsf{h}^2)^2}{8} \int_{\mathbb{R}^d} \frac{1}{\rho_0} \Big(\int_{T-a}^T \exp\big(\frac{\mathsf{h}^2}{2}(T-t)\mathcal{K}_T^\leftarrow\big) \Gamma_T^\leftarrow \, \mathrm{d}t\Big)^2,
\end{aligned}
$$

where we used $\Gamma_t^\leftarrow \approx \Gamma_T^\leftarrow$, when $t \approx T$ in the second line; we used $\frac{\mathcal{L}_t^{(h^\leftarrow)}}{\mathsf{h}^2/2} \approx \frac{\mathcal{L}_T^{(h^\leftarrow)}}{\mathsf{h}^2/2}$ when $t \approx T$ in the third line; $\mathcal{L}_T^{(h^\leftarrow)} = \mathsf{h}^2/2 \, \mathcal{K}_T^\leftarrow$ (40) herein. In the last line, we used the fact that $\rho_0 \sim p_0$ when $\mathsf{h} \to \infty$.

By Lem. D.2,

$$
\begin{aligned}
\mathcal{T}_1 &\sim \frac{(1+\mathsf{h}^2)^2}{8} \sum_{k=1}^\infty \Big(\frac{1 - e^{-a\frac{\mathsf{h}^2}{2}\lambda_k^{(\mathsf{h})}}}{\frac{\mathsf{h}^2}{2}\lambda_k^{(\mathsf{h})}}\Big)^2 \big(\alpha_k^{(\mathsf{h})}\big)^2 \\
&\sim \frac{1}{2} \sum_{k=1}^\infty \Big(\frac{\alpha_k^{(\mathsf{h})}}{\lambda_k^{(\mathsf{h})}}\Big)^2 \qquad (\text{by } a = \mathsf{h}^{-\beta}, \ \mathsf{h} \gg 1),
\end{aligned}
$$

where $(\lambda_k^{(\mathsf{h})}, \phi_k^{(\mathsf{h})})$ are eigen pairs of $\mathcal{K}_T^\leftarrow$ and $\Gamma_T^\leftarrow = \sum_{k=1}^\infty \alpha_k^{(\mathsf{h})} \phi_k^{(\mathsf{h})}$. When $\mathsf{h} \to \infty$, we know that the eigenvalues of $\mathcal{K}_T^\leftarrow$, which depends on $\mathsf{h}$ and has the form

$$
\mathcal{K}_T^\leftarrow(\mu)(x) = \triangle\mu(x) + \boldsymbol{\nabla} \cdot (\nabla U_0(x)\mu(x)) + \mathsf{h}^{-2}\boldsymbol{\nabla} \cdot \Big(\big(\nabla U_0(x) - x\big)\mu\Big),
$$

should converge to $\mathcal{K}^\infty$, defined as

$$
\mathcal{K}^\infty(\mu) := \triangle\mu + \boldsymbol{\nabla} \cdot (\nabla U_0 \mu).
$$

Therefore, in the limit,

$$
\lim_{\mathsf{h} \to \infty} \mathcal{T}_1 = \frac{1}{2} \sum_{k=1}^\infty \big(\alpha_k^{(\infty)}/\lambda_k^{(\infty)}\big)^2,
$$

where $(\lambda_k^{(\infty)}, \phi_k^{(\infty)})$ are eigen pairs of $\mathcal{K}^\infty$ and $\Gamma_T^\leftarrow = \sum_{k=1}^\infty \alpha_k^{(\infty)} \phi_k^{(\infty)}$.

**Upper bound of $\mathcal{T}_1$**

By the upper bound in Lem. D.2 (together with $a = \mathsf{h}^{-\beta}$ and $\mathsf{h} \gg 1$) or by applying $\lambda_k^{(\mathsf{h})} \geq \kappa_0$ directly to the asymptotic of $\mathcal{T}_1$, when $\mathsf{h} \gg 1$,

$$
\mathcal{T}_1 \lesssim \frac{1}{2\kappa_0^2} \int_{\mathbb{R}^d} \frac{\Gamma_T^{\leftarrow 2}}{\rho_0} = \frac{1}{2\kappa_0^2} \int_{\mathbb{R}^d} \frac{(\boldsymbol{\nabla} \cdot (p_T^\leftarrow \mathscr{E}_T^\leftarrow))^2}{\rho_0} \sim \frac{1}{2m_0^2} \int_{\mathbb{R}^d} \frac{(\boldsymbol{\nabla} \cdot (p_0 \mathscr{E}_T^\leftarrow))^2}{p_0}.
$$

The term $\mathcal{T}$ in Prop. 3.6 is simply the limit of $\mathcal{T}_1$.

### F.2 Remark on the large diffusion limit

**Score function parameterization in constrained score models**

We would like to remark on the parameterization of score function. It is common in literature to directly parameterize $\mathfrak{S}_t$ (4) via some neural network. However, in general, this practice can

rarely guarantee that $\mathfrak{S}_t$ (which is supposed to approximate $\nabla \log p_t$) has the gradient form as well [23, 25]. If we instead parameterize $\log p_t(x) \approx \mathrm{N}_t(x; \theta)$, where $\mathrm{N}.(\cdot; \theta)$ is some (scalar-valued) neural network with parameter $\theta$, we can simply use the following ansatz in training

$$\mathfrak{S}_t(x) = \nabla \mathrm{N}_t(x; \theta).$$

Consequently, the error of score function estimate also has a gradient form:

$$
\begin{aligned}
\epsilon \mathscr{E}_t^{\leftarrow}(x) &\stackrel{(6)}{=} \mathfrak{S}_t^{\leftarrow}(x) - \nabla \log p_t^{\leftarrow}(x) \\
&= \nabla \mathrm{N}_t^{\leftarrow}(x; \theta) - \nabla \log p_t^{\leftarrow}(x) \\
&= \nabla \underbrace{\left( \mathrm{N}_t^{\leftarrow}(x; \theta) - \log p_t^{\leftarrow}(x) \right)}_{=: \varphi}.
\end{aligned}
$$

This validates the discussion in § 3.6. Apart from the apparent benefit in preserving the gradient form, this ansatz can help us identify some interesting structure for the the upper bound of $\lim_{\mathsf{h} \to \infty} L(\mathsf{h})$ in (18), discussed in § 3.6.

**More discussion on the upper bound**

Because the score error $\mathscr{E}_T^{\leftarrow}$ comes from approximating $\nabla \log p_0$, it is reasonable to consider the case $\varphi = \log p_0$, the upper bound (18), or equivalently (19), becomes

$$\mathcal{T} \lesssim \frac{1}{2m_0^2} \int_{\mathbb{R}^d} \left( |\nabla U_0|^2 - \triangle U_0 \right)^2 e^{-U_0}. \tag{41}$$

This quantity heuristically characterizes how difficult the probability distribution $p_0$ can be learn by score-based diffusion models with large diffusion coefficient.

It is of interest to further explore how to utilize the above upper bound to improve the training of score function, or provide theoretical understanding about what types of probability distributions are easy/difficult to learn via score-based SDEs. We shall leave an extensive study of these important and interesting questions for future work. Below, we shall provide a simple example.

**Example: $d$-dimensional Gaussian**

When $p_0 = e^{-U_0} = \mathcal{N}(\mu, \sigma_0^2)$, the quantity $m_0$ (14) is $m_0 = \frac{1}{\sigma_0^2}$. Then this upper bound (18) is

$$\frac{\sigma_0^4}{2} \int_{\mathbb{R}^d} \left( \left| \frac{x - \mu}{\sigma_0^2} \right|^2 - \frac{d}{\sigma_0^2} \right)^2 e^{-U_0} = d.$$

This is compatible with the insight that it is more difficult to learn a high dimensional probability distribution.

# G   Numerical experiment: 1D Gaussian case

We demonstrate and validate main findings via a simple 1D Gaussian. Suppose $f_t(x) = -\frac{1}{2}x$, $g_t = 1$, and the exact data distribution is a Gaussian $p_0 = \mathcal{N}(0, \sigma_0^2)$. The error is quantified by $\mathrm{KL}\left(p_0 || \widetilde{q}_T\right)$. We investigate this error as a function of the magnitude of $h^{\leftarrow}$ (which is chosen as a constant function $h_t^{\leftarrow} = \mathsf{h}$ for any $t \in [0, T]$).

## G.1   Explicit formulas for 1D Gaussian case

From solving the SDE for the forward process, we know that

$$p_t = \mathcal{N}(0, \sigma_t^2), \qquad \sigma_t^2 = \sigma_0^2 e^{-t} + 1 - e^{-t}, \qquad \nabla \log p_t(x) = -\frac{x}{\sigma_t^2}. \tag{42}$$

The backward dynamics (7) on $t \in [0, T]$ is

$$
\begin{cases}
\mathrm{d}\widetilde{Y}_t = \left( \frac{1}{2}\widetilde{Y}_t + \frac{1 + h_t^{\leftarrow 2}}{2} \left( -\frac{\widetilde{Y}_t}{\sigma_{T-t}^2} + \epsilon \mathscr{E}_t^{\leftarrow}(\widetilde{Y}_t) \right) \right) \mathrm{d}t + h_t^{\leftarrow} \, \mathrm{d}W_t, \\
\mathrm{law}(\widetilde{Y}_0) = p_T.
\end{cases}
$$

In this example, there is only one source of error, which is $\mathscr{E}^{\leftarrow}$, as we know explicitly the distribution $p_T$ and we can choose the time step small enough such that numerical discretization error is negligible. Due to the structure of the score function, it is reasonable to consider the following ansatz

$$\mathscr{E}_t^{\leftarrow}(x) = \alpha_t x. \tag{43}$$

This example has explicit formulas: $\widetilde{Y}_T$ is a Gaussian distribution with $\mathbb{E}[\widetilde{Y}_T] = 0$ and

$$\mathrm{Var}(\widetilde{Y}_T) = G_T^{-2}\mathrm{Var}(\widetilde{Y}_0) + \int_0^T G_T^{-2}G_t^2 h_t^{\leftarrow 2}\,\mathrm{d}t,$$

where

$$G_t := \exp\Big(-\int_0^t \frac{1}{2} + \frac{1+h_s^{\leftarrow 2}}{2}\big(-\frac{1}{\sigma_{T-s}^2} + \epsilon\alpha_s\big)\mathrm{d}s\Big).$$

The sample generation error quantified by KL divergence also has an explicit formula:

$$\mathrm{KL}\big(p_0||\widetilde{q}_T\big) = \frac{1}{2}\log\Big(\frac{\mathrm{Var}(\widetilde{Y}_T)}{\sigma_0^2}\Big) + \frac{\sigma_0^2}{2\mathrm{Var}(\widetilde{Y}_T)} - \frac{1}{2}.$$

### G.2 Experiment 1: Fix error magnitude $\epsilon$ and consider various error types

We choose $T = 2$, $h_t^{\leftarrow} = h$ for all $t \in [0, T]$, and the error function is chosen as

$$\epsilon = 0.02, \qquad \mathscr{E}_t^{\leftarrow}(x) = \nabla\log p_t^{\leftarrow}(x) \times \begin{cases} 1 & \text{case 1;} \\ -1 & \text{case 2;} \\ \dfrac{1+\sin(2\pi t/T)}{2} & \text{case 3;} \\ \mathbb{I}_{t<0.95T} & \text{case 4;} \\ \mathbb{I}_{t>0.99T} & \text{case 5.} \end{cases} \tag{44}$$

For these choices, we only perturb the true score function by a bounded prefactor. We can observe that the error $\mathrm{KL}\big(p_0||\widetilde{q}_T\big)$ is a complicated function of h in general in Fig. 6. When we only perturb the score function during the initial period of the generative process (case 4), we can clearly observe that the sampling error decays exponentially fast with respect to $h^2$, which numerically validates Prop. 3.4. When we only perturb the score function near the end of the generative process (case 5), increasing the diffusion coefficient h will actually increase the error, which is predicted by Prop. 3.5. For a general error (case 1, 2, 3), overall we can still expect that increasing diffusion coefficient h will generally suppress the error when $\sigma_0$ is small.

Table 3: Approximated value of $L(h)$ when $h^2 \approx 20$. We can observe that in each column (referring to a specific error type) the value is almost independent of the $\sigma_0$ in particular when $\sigma_0 < 1$.

| $\sigma_0$ | Case 1 | Case 2 | Case 3 |
|---|---|---|---|
| 0.2 | 0.2567 | 0.3032 | 0.0658 |
| 0.4 | 0.2569 | 0.3028 | 0.0636 |
| 0.6 | 0.2570 | 0.3018 | 0.0597 |
| 0.8 | 0.2569 | 0.3023 | 0.0544 |
| 1.5 | 0.2570 | 0.3017 | 0.0321 |
| 2.0 | 0.2564 | 0.3022 | 0.0198 |
| 3.0 | 0.2566 | 0.3020 | 0.0121 |

### G.3 Experiment 2: consider $L(h)$ for various error types

We further numerically approximate $L(h) \equiv L(h, \mathscr{E}^{\leftarrow}, p_0)$ by linear regression and study how $\sigma_0$ (i.e., the data distribution $p_0$), error type $\mathscr{E}^{\leftarrow}$, and diffusion coefficient h affect the leading order term $L(h, \mathscr{E}^{\leftarrow}, p_0)$. Recall that we use $L(h)$ as a simplified notation when $\mathscr{E}^{\leftarrow}$ and $p_0$ are clear from context; see § 3. As a remark, to approximate $L(h)$, we used the leading-order approximation that $\mathrm{KL}\big(p_0||\widetilde{q}_T\big) = L(h)\epsilon^2 + \mathcal{O}\big(\epsilon^3\big)$: we choose a few $\epsilon$ values and use linear regression to estimate $L(h)$.

In Fig. 7, we can clearly observe that $L(\mathsf{h})$ converges to a constant extremely fast when $\mathsf{h}$ increases for cases $\sigma_0 < 1$. The value of $\mathscr{E}_T^{\leftarrow}$ for the case 3 is only $1/2$ of that for case 1 and case 2. By Prop. 3.6, we know that $\lim_{\mathsf{h}\to\infty} L(\mathsf{h})$ for the case 3 should be approximated $1/4$ of that for cases 1 and 2. This is also (approximately) numerically observed in Table 3.

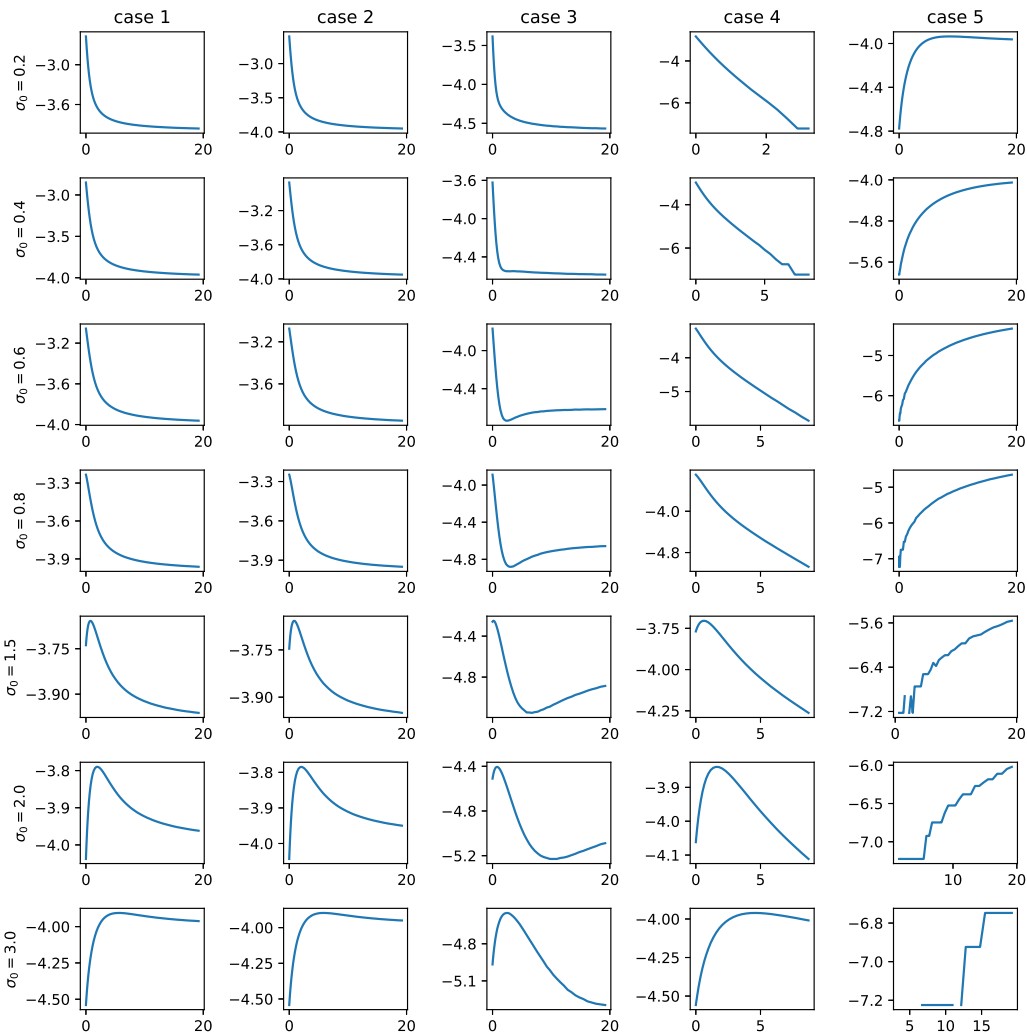

Figure 6: We show $\log_{10}\big(\mathrm{KL}\big(p_0\|q_T\big)\big)$ as a function of $\mathsf{h}^2$ for the 1D Gaussian model. $T = 2$, different $\sigma_0$ and error functions in (44) are considered.

# H   More details about numerical experiments in § 4

In this section, we discuss datasets, network architectures, evaluation metrics, numerical schemes (exponential integrator), and the default weight in denoising score matching.

## H.1   Datasets

- **1D 2-mode Gaussian mixture**: $p_0(x) = \sum_{i=1}^{2} 0.5\mathcal{N}\big(x; (-1.0)^i, 0.01\big)$.
- **2D 4-mode Gaussian mixture**: $p_0(x) = \sum_{i,j=1}^{2} 0.25\mathcal{N}\big(x; ((-1.0)^i, (-1.0)^j), 0.05^2\boldsymbol{I}_2\big)$.
- **Swiss roll**: Swiss roll generates samples by $(x, y) = \big(t\sin(t), t\cos(t)\big)$ with $t$ drawn from the uniform distribution $\mathcal{U}\big(\frac{3\pi}{2}, \frac{9\pi}{2}\big)$.

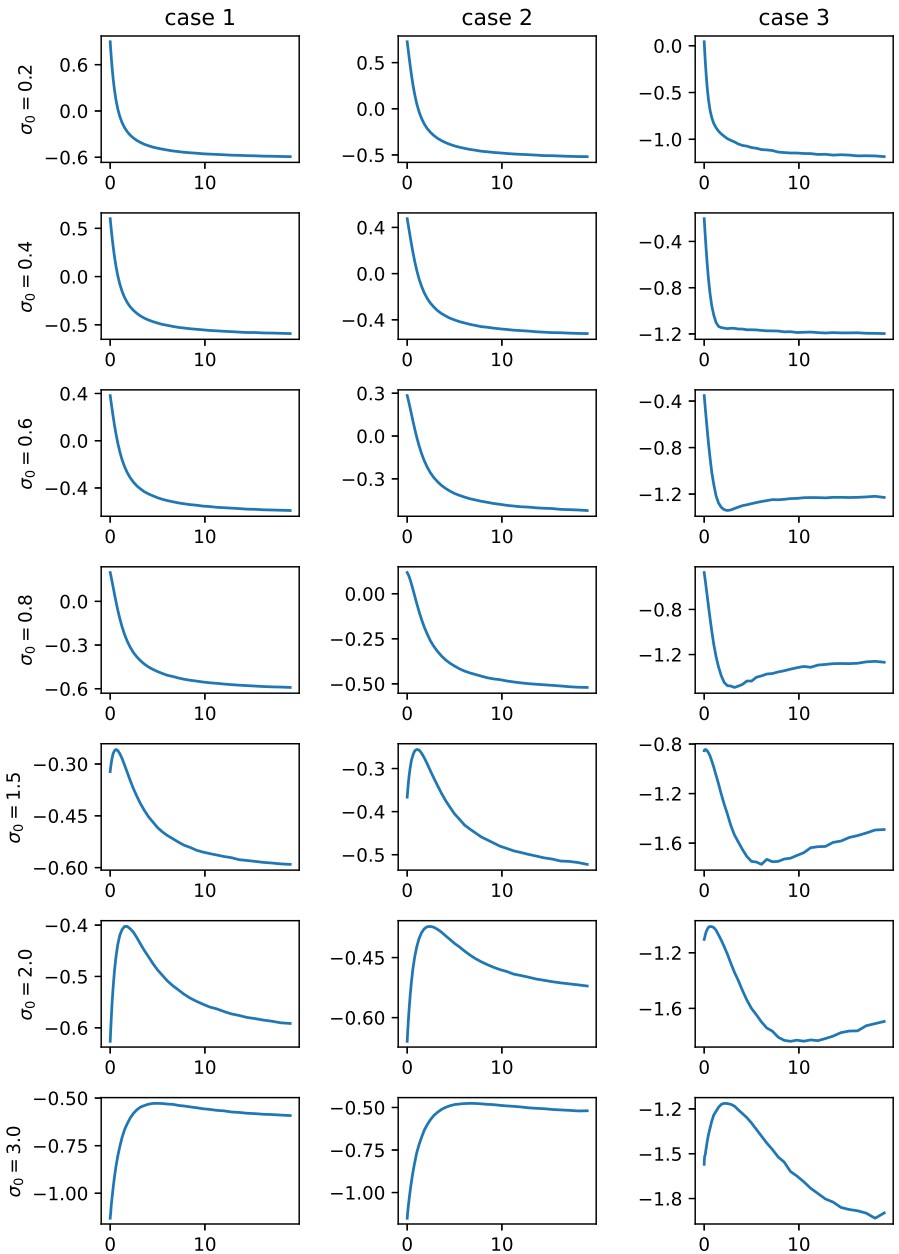

Figure 7: We show $\log_{10}\left(L(\mathsf{h})\right)$ as a function of $\mathsf{h}^2$ for the 1D Gaussian model. $T = 2$, different $\sigma_0$ and error functions in (44) are considered.

- **MNIST**: MNIST [34] contains 60,000 $28 \times 28$ gray-scale images with hand-written digits.
- **CIFAR-10**: CIFAR-10 [22] contains 60,000 $32 \times 32$ RGB images with ten categories.

## H.2 Network architectures and other parameters

For experiments on 1D/2D Gaussian mixtures, the exact scores can be obtained analytically if we use VP-SDE. We set $T = 4$ and $g_t = 1$ for $t \in [0, T]$. For the time discretization when solving the reverse SDE with Euler-Maruyama method, we apply 40,000 steps and 80,000 steps for 1D and 2D Gaussian mixtures, respectively.

For experiments on Swiss roll, we apply a three-layer neural network for score matching, where the width of each layer is set as $50$, $50$, and $2$ and we apply ReLU as the nonlinear activation for two hidden layers. We set $T = 1$ and $g_t = \sqrt{0.1(1 - t) + 20t}$ for $t \in [0, T]$. The learning rate is set as $0.01$ and decays by $0.5$ every 8,000 steps. The batch size is set as 400. We train the neural network for 20,000 steps. For the time discretization when solving the reverse SDE with Euler-Maruyama method, we apply 20,000 steps.

For experiments on MNIST, we apply the net architecture in [17] for score matching, where we use two resolution blocks in U-net and set the multipliers of channels to be one and two. We set $T = 1.4$ and $g_t = \sqrt{0.1(1 - t) + 20t}$ for $t \in [0, T]$. The number of iteration is $20,000$, the batch size is set as $64$. We solve the reverse SDE with exponential integrator; see also Appx. H.4.

For experiments on CIFAR-10, we apply the DDPM++ cont. in [30] as the net architecture, and use their pretrained checkpoint in for score estimation. We use the same setting as [30], i.e., $T = 1$ and $g_t = \sqrt{0.1(1 - t) + 20t}$ for $t \in [0, T]$. For the time discretization when sampling with Euler-Maruyama method, we apply 100, 200, 500, 1000, 2000, 3000 and 4000 steps.

## H.3 Evaluation metrics

For experiments of 1D/2D Gaussian mixtures and Swiss roll, we apply approximated divergences for evaluating the performances. Specifically, we discretize the space into 100 bins in each dimension, then obtain the empirical densities of 10,000 true samples and 10,000 generated samples, and use Jensen-Shannon divergence, Kullback-Leibler divergence and Wasserstein distance between both empirical densities as the metrics for evaluation.

## H.4 Exponential integrator

We shall explain the exponential integrator for (2) with $t \in [0, T]$, i.e., the following equation

$$
\begin{aligned}
\mathrm{d}Y_t &= -f_t^{\leftarrow}(Y_t)\,\mathrm{d}t + \frac{(g_t^{\leftarrow})^2 + (h_t^{\leftarrow})^2}{2} \nabla \log p_t^{\leftarrow}(Y_t)\,\mathrm{d}t + h_t^{\leftarrow}\;\mathrm{d}W_t \\
&= \frac{1}{2}g_t^{\leftarrow 2}Y_t\,\mathrm{d}t + \frac{(g_t^{\leftarrow})^2 + (h_t^{\leftarrow})^2}{2} \nabla \log p_t^{\leftarrow}(Y_t)\,\mathrm{d}t + h_t^{\leftarrow}\;\mathrm{d}W_t\,.
\end{aligned}
$$

Then for any time $t \in [t_k, t_{k+1}]$, and given $\hat{Y}_{t_k}$, we approximate the above dynamics by

$$
\mathrm{d}\hat{Y}_t \approx \frac{1}{2}g_t^{\leftarrow 2}\hat{Y}_t\,\mathrm{d}t + \frac{(g_t^{\leftarrow})^2 + (h_t^{\leftarrow})^2}{2} \nabla \log p_{t_k}^{\leftarrow}(\hat{Y}_{t_k})\,\mathrm{d}t + h_t^{\leftarrow}\;\mathrm{d}W_t\,.
$$

This dynamics is a linear SDE and we can solve it exactly

$$
\hat{Y}_{t_{k+1}} = \underbrace{e^{\frac{1}{2}\int_{t_k}^{t_{k+1}} g_s^{\leftarrow 2}\,\mathrm{d}s}\hat{Y}_{t_k} + \left[\int_{t_k}^{t_{k+1}} e^{\frac{1}{2}\int_t^{t_{k+1}} g_s^{\leftarrow 2}\,\mathrm{d}s}\frac{g_t^{\leftarrow 2} + h_t^{\leftarrow 2}}{2}\,\mathrm{d}t\right]\boldsymbol{S}_k}_{=:\mathscr{T}_1}
$$

$$
+ \int_{t_k}^{t_{k+1}} \underbrace{e^{\frac{1}{2}\int_t^{t_{k+1}} g_s^{\leftarrow 2}\,\mathrm{d}s}h_t^{\leftarrow}}_{:=\mathscr{T}_2}\;\mathrm{d}W_t,
$$

$$
\boldsymbol{S}_k := \nabla \log p_{t_k}^{\leftarrow}(\hat{Y}_{t_k}).
$$

Therefore,

$$\hat{Y}_{t_{k+1}} = \mathscr{T}_1 + \sqrt{\int_{t_k}^{t_{k+1}} \left(\mathscr{T}_2\right)^2 \, \mathrm{d}t} \, Z_k, \qquad Z_k \sim \mathcal{N}(0, I_d).$$

If we pick

$$\begin{cases} h_t^{\leftarrow} = \alpha g_t^{\leftarrow}, & \alpha \in \mathbb{R}^+, \\ g_t = \sqrt{\beta_0 + (\beta_1 - \beta_0)t} \end{cases} \quad \text{which implies that} \quad g_t^{\leftarrow} = \sqrt{\beta_0 + (\beta_1 - \beta_0)(T - t)},$$

then after straightforward calculations,

$$\hat{Y}_{t_{k+1}} = \gamma_k \hat{Y}_{t_k} + (1 + \alpha^2)\left(\gamma_k - 1\right)\boldsymbol{S}_k + \sqrt{\alpha^2\left(\gamma_k^2 - 1\right)}Z_k,$$

where $\delta_k := t_{k+1} - t_k$ and $\gamma_k := \exp\left(\frac{\delta_k\left(2\beta_0 + (2t_k - 2T + \delta_k)(\beta_0 - \beta_1)\right)}{4}\right)$.

## H.5 Default training weight

Given the fixed $X_0$, we can explicitly solve (1) (with the choice $f_t(x) = -\frac{g_t^2}{2}x$):

$$X_t = (G_t)^{-1}X_0 + \int_0^t G_t^{-1}G_s g_s \, \mathrm{d}W_s,$$

where $G_t = \exp\left(\int_0^t \frac{g_s^2}{2} \, \mathrm{d}s\right)$. The standard deviation of the Brownian motion term above is

$$\varpi_t := \sqrt{\int_0^t G_t^{-2}G_s^2 \, g_s^2 \, \mathrm{d}s}.$$

When $g_t = \sqrt{\beta_0 + (\beta_1 - \beta_0)t}$ as used in many literature [30, 17], we can explicitly solve for $\varpi_t$:

$$\varpi_t = \sqrt{1 - \exp\left(-\frac{1}{2}t^2(\beta_1 - \beta_0) - t\beta_0\right)}.$$

In literatures, to balance the noise over time in training the score function, a common practice is to use the default weight $\omega_t = \varpi_t^2$ (5) in the (denoising) score-matching loss function (4).

## H.6 Additional numerical results for experiments in § 4

We present more numerical results on 1D Gaussian mixture, Swiss roll, and MNIST to further verify theoretical results.

**1D Gaussian mixture**

We evaluate the performances of generative models under different values of h for 1D Gaussian mixture, and present the visualization and numerical results in Fig. 8 and Fig. 9, respectively. In Fig. 8, a clear trend shows that with increasing h, the empirical density of generated samples better matches the true density function. This trend is more quantitatively captured in Fig. 9, from which we clearly observe that the distance between the empirical and the true density function decreases to the numerical threshold exponentially fast. Numerical threshold means the error of various distances when $\epsilon = 0$; due to the space discretization when computing various distances, the numerical values of various distances are not exactly zero even when we use the exact score function. However, increasing h can help us to almost reach this limit and this phenomenon is theoretically described in Prop. 3.4.

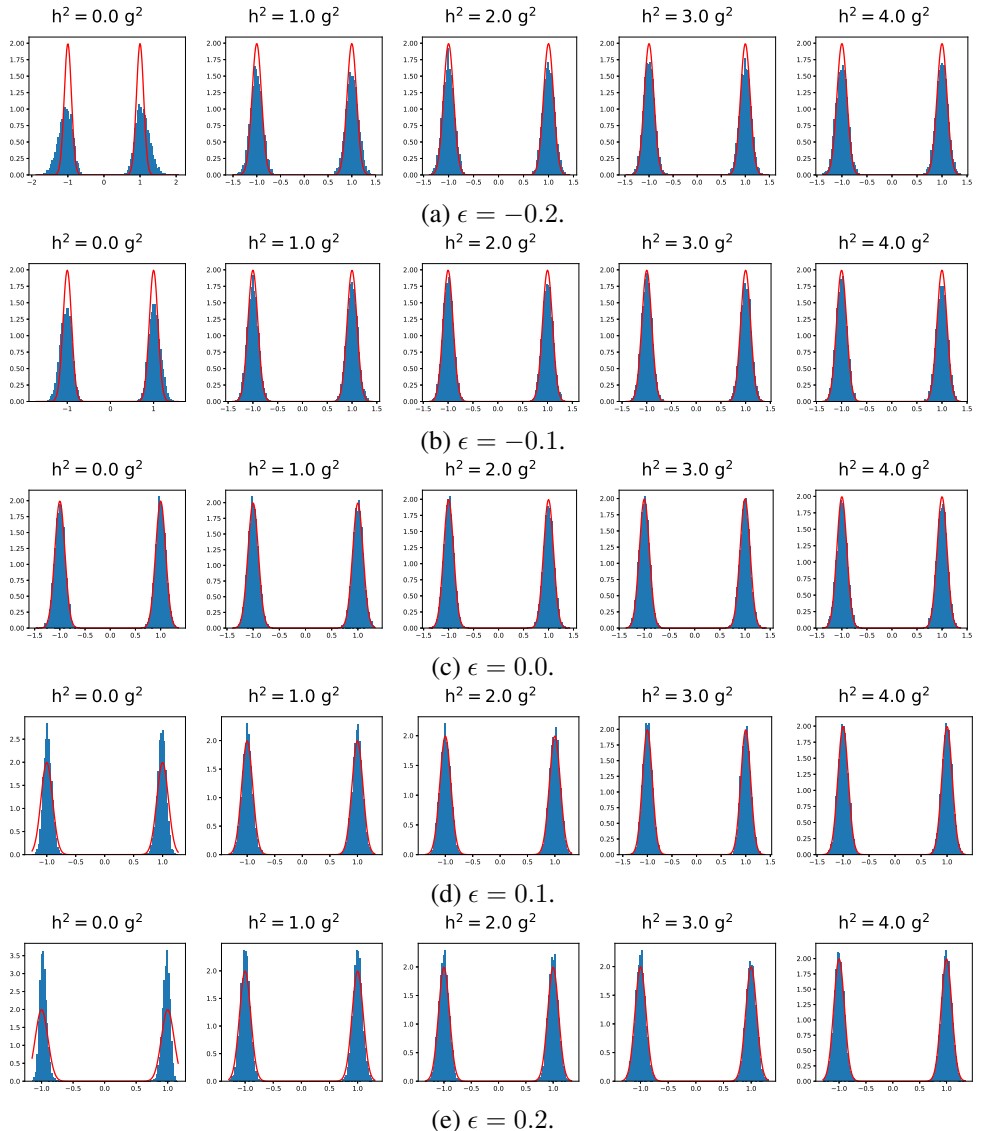

Figure 8: Visualization of 1D GMM, where $\mathscr{E}_t^{\leftarrow}(x) = \nabla \log p_t^{\leftarrow}(x)$. We can observe that increasing h can help to generate samples with a distribution closer to the true $p_0$.

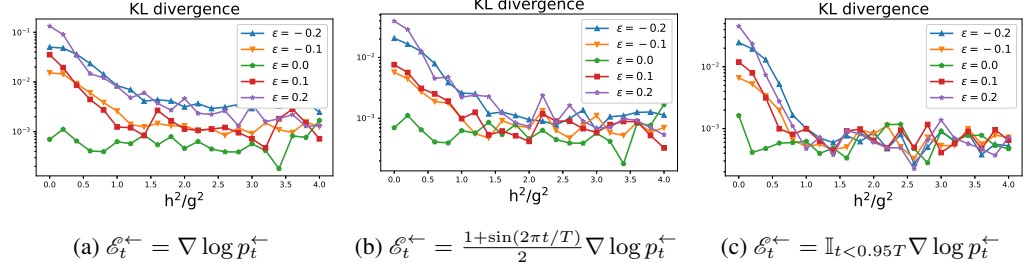

(a) $\mathscr{E}_t^{\leftarrow} = \nabla \log p_t^{\leftarrow}$  (b) $\mathscr{E}_t^{\leftarrow} = \frac{1+\sin(2\pi t/T)}{2} \nabla \log p_t^{\leftarrow}$  (c) $\mathscr{E}_t^{\leftarrow} = \mathbb{I}_{t<0.95T} \nabla \log p_t^{\leftarrow}$

Figure 9: Numerical results of 1D 2-mode Gaussian mixture. We can observe that the distance between the empirical density and the true density function decreases as h increases, when $\mathscr{E}_T^{\leftarrow}$ is not extremely large.

**Swiss roll**

In Fig. 10, we provide additional figures to discuss the effect of time-discretization. When the numerical error is negligible, we can observe that the generative process with a larger h can provide a clearer picture of Swiss roll, as shown in Fig. 10c. However, when the discretization error cannot be ignored, the conclusion may be reversed. Therefore, it is necessary to design and employ more accurate numerical methods for models with large h in order to fully benefit from diffusion models with a large diffusion coefficient.

In Fig. 11, we display JS and KL divergences between the true density $p_0$ and the generated samples. Since the data distribution of Swiss roll is highly localized (data is concentrated on a curve embed in 2D), accurately computing the KL divergence poses a significant numerical challenge. That is why we use Wasserstein distance instead in Fig. 5a. Based on more robust symmetric metrics (i.e., the JS and Wasserstein distance herein), we can observe that a larger h can indeed diminish the error in sample generation.

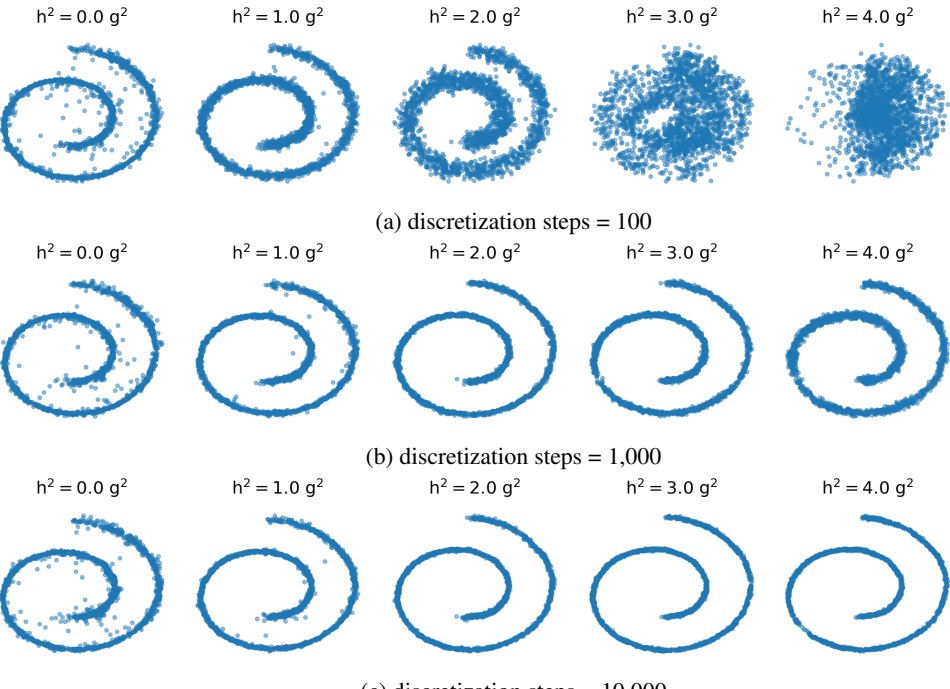

(a) discretization steps = 100

(b) discretization steps = 1,000

(c) discretization steps = 10,000

Figure 10: Visualization results of Swiss roll with different number of time steps.

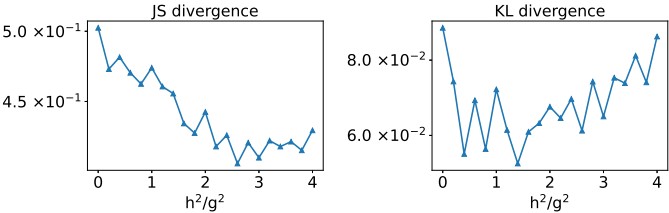

Figure 11: Numerical results of Jensen-Shannon divergence and Kullback-Leibler divergence of Swiss roll.

## MNIST

We conduct further experiments to explore the effect of $h^{\leftarrow}$ and the time discretization steps. It is important to note that if one has an extremely well trained score function, then the effect of $h^{\leftarrow}$ is indeed negligible, as shown in (3). To somewhat magnify the score training error for MNIST (but with a reasonable score function), we increase the time $T$ and use a smaller architecture with fewer parameters; the reason of the occasional failure to generate clear MNIST images in later figures comes from this **deliberate experimental design**.

For an existing pre-trained score function with non-negligible error, we notice that increasing $h^{\leftarrow}$ can almost **ubiquitously** improve the quality of sample generation; see Fig. 12 (as well as Fig. 15 and Fig. 16 under different training setup). This improvement is supported by our theoretical results, particularly Prop. 3.4. Notably, even when choosing $h^{\leftarrow} = 0$ (ODE) and $h^{\leftarrow} = g^{\leftarrow}$ (the default diffusion model in many studies) occasionally fail to generate an image, simply by increasing the magnitude of $h^{\leftarrow}$ (possibly at the cost of more computational resources), we have a larger chance of generating an image with reasonable quality; for instance, see the last row in Fig. 12 (particularly see the last row in Fig. 15). The ODE-based model sometimes fails to generate hand-writing digits even when the step number is $10^3$, whereas the diffusion model (with large $h^{\leftarrow}$) does **not** encounter this issue. This conclusion might be reversed when the time step size is large, which is similarly observed in the Swiss roll.

The computational cost of SDE-based models with large diffusion ($h^{\leftarrow} > g^{\leftarrow}$) is relatively high due to the necessity of a larger number of time steps: a larger $h^{\leftarrow}$ has the similar effect as running Langevin for a larger time horizon as discussed in § 3.3 and a longer time simulation is expected to be more expensive and also its accuracy largely relies on a well-chosen time step. However, this can be offset by the ability to use a lightweight architecture that possibly speeds up the generative process. A detailed comparison of various diffusion-based models with the same computational budget constraint is challenging and is slightly beyond the scope of this work, and we will leave this task to future work.

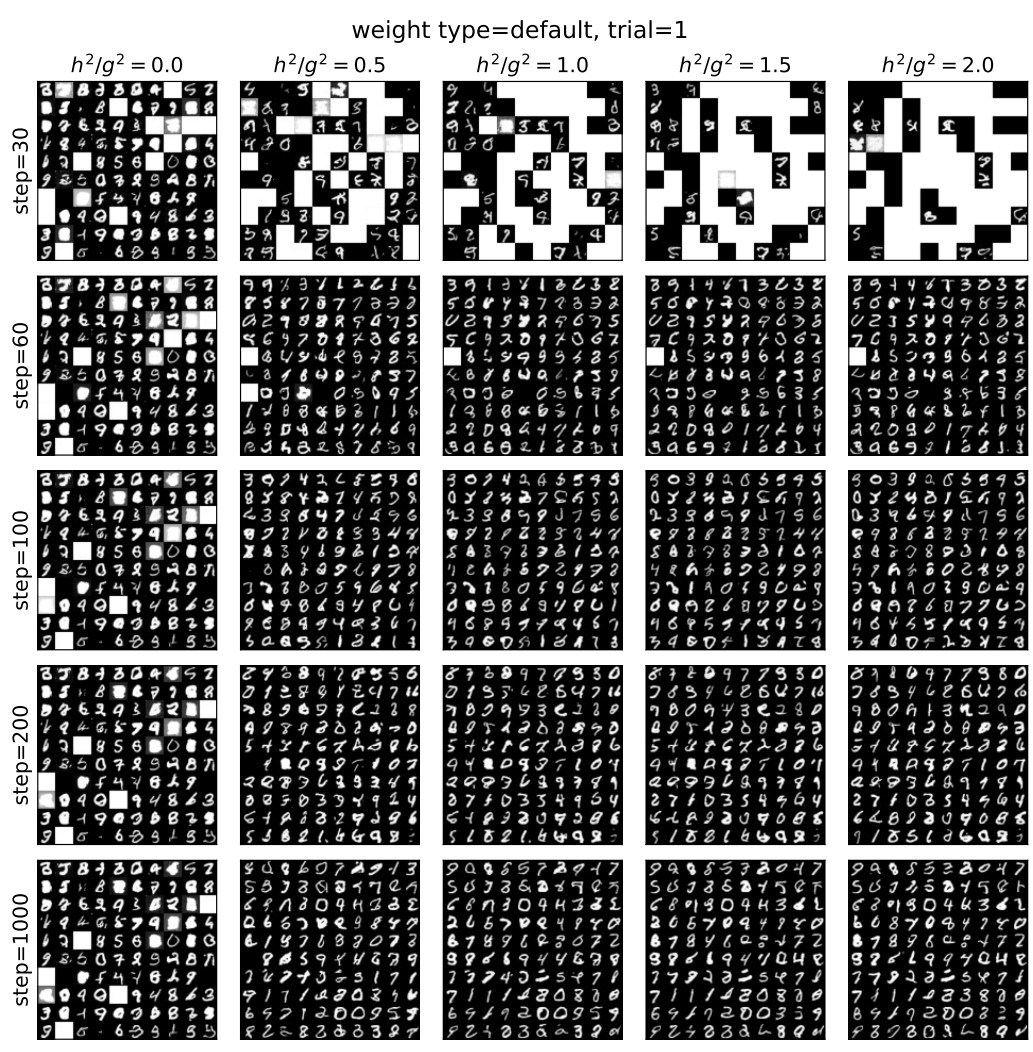

Figure 12: Visualization of sample generation in MNIST where we used **default weight** $\omega$ (5). We use different random seed to ensure robustness and this is the result for trial 1. Results for trial 2 is not included in this work as the overall tendency remains the same.

# I Numerical experiments for adopting different weight in training

We have two reasons to explore the effects of various weight functions in training.

The first reason is that theoretically, any positive scalar-valued functions $\omega_t$ on $(0, T)$ is a valid candidate. This prompts the question of whether such a default weight function (5) is optimal in designing the loss function. We fully acknowledge that the default choice adopted in most literatures is a very effective one. However, the mathematical reason behind it is still not satisfactory in our opinion. This motivates us to ask whether the default choice is really the optimal one, at least in certain circumstances.

The second (and actually the primary reason) comes from our theoretical predictions discussed in § 3.7. As there is a tight connection between the optimal reverse-time generative process and the time-distribution of the score error (see Prop. 3.4, Prop. 3.5 and Prop. 3.6), if we are willing to train or re-train the score function and are interested in using the ODE-based model (for fast sample generation), it appears that we should focus more on the noise's end comparatively. To achieve this goal, namely, to control the distribution of score error, we adopt different weight schemes in the loss function **only** for training: default weight in (5), a data-driven case (more weight in the data side) and a noise-driven case (more weight in the noise side):

$$
\omega_t = \begin{cases} \varpi_t^3 & \text{noise-driven, or simply referred as \textbf{``noise''}}; \\ \dfrac{\varpi_t^2}{0.25 + \varpi_t} & \text{data-driven, or simply referred as \textbf{``data''}}. \end{cases} \tag{45}
$$

There is no theoretical reasons behind the noise-driven and data-driven choices in the last equation. We merely experiment with two reasonable choices and at the same time they are expected to help us control the time distribution of the score error.

After the initial camera-ready version of this work, we notice that such a weight design (in particular the noise-driven weight) has also been studied in [12]. Their experiments seem to provide further evidence about the validity of our arguments. However, we would like to emphasize that the approaches and perspectives towards proposing such a weight design are different: in [12], their conjecture is based on an insightful perception argument, whereas we take a mathematical approach and come up with the design based on the above error accumulation analysis.

## I.1 Our guess

Based on our theoretical results, we guess that if a different weight function can really achieve our expected goal (namely, control the time distribution of the score function), the noise-driven case should be more suitable for ODE models, and the data-driven weight should give us the worst performance for ODE models.

We remark that this conjecture is surely **not universal**, and its validity remains to be fully validated by more benchmark experiments. Nevertheless, our numerical experiments below suggest its potential usefulness and it prompts an interesting question to explore and design the optimal score-matching loss function, which is **rarely** studied in literature. In what follows, we report numerical experiments for MNIST and CIFAR-10.

## I.2 MNIST

We used different weights to train the score function and then visualize their generated samples in Fig. 13. We can clearly observe that the score function trained by the noise-driven weight produces comparatively better samples in ODE models. We plot the denoising score-matching loss for score functions obtained from training with various weights in Fig. 14:

- to make the comparison more straightforward, we visualize the time-distribution of **relative score-matching loss** rather than the absolute value, namely, we demonstrate:

$$
t \mapsto \frac{\mathbb{E}_{X_0 \sim p_0} \mathbb{E}_{X_t \sim p_{t|0}(\cdot|X_0)} \left[ \left\| (\mathfrak{S}^{(i)})_t(X_t) - \nabla \log p_{t|0}(X_t|X_0) \right\|^2 \right]}{\mathbb{E}_{X_0 \sim p_0} \mathbb{E}_{X_t \sim p_{t|0}(\cdot|X_0)} \left[ \left\| (\mathfrak{S}^{(\text{default})})_t(X_t) - \nabla \log p_{t|0}(X_t|X_0) \right\|^2 \right]}, \quad i \in \{\text{noise, data}\}, \tag{46}
$$

and $\mathfrak{S}^{(i)}$ refers to the score function obtained from training using weight $i$ and we use test datasets to approximately represent $p_0$;

- to ensure robustness, we independently conduct two trials with different neural network initializations: For each trial, the initial neural network is the same and the only difference in training is to adopt different weights in the loss function.

In Fig. 14, we can clearly observe that adopting different weights indeed help us to control how the score error is distributed over time as we expect (e.g., noise-driven weight helps us reduce the error near $t \approx T$ and has an opposite effect near $t \approx 0$); as shown in Fig. 13, their numerical performances in terms of sample generation also match our guess above.

We further visualize how the diffusion coefficient $h$ and time step size affect the sample generation quality for score functions obtained by data-driven weight (see Fig. 15) and noise-driven weight (see Fig. 16). The conclusion is the same as in the default weight case. This further validates our theoretical results in § 3.

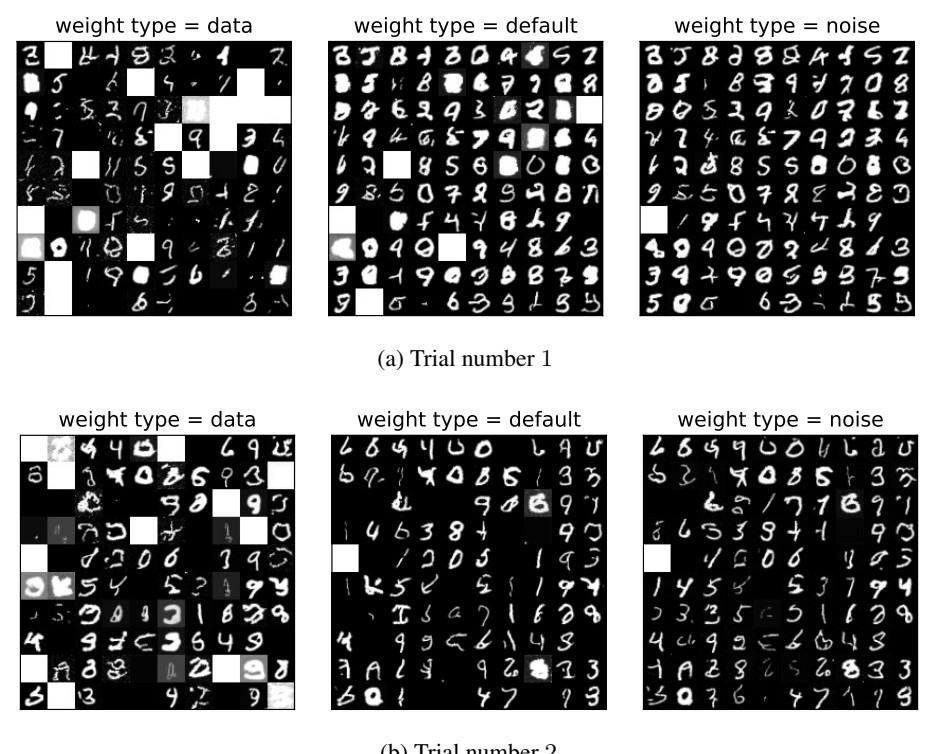

Figure 13: We show generated samples trained for MNIST using three different weight functions in the loss function: we use $h^2/g^2 = 0$ (ODE), time step is $1000$, the trial number is the index for independent random initialization.

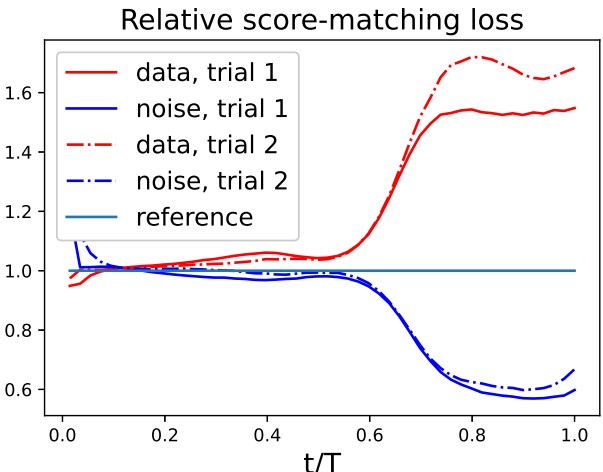

Figure 14: Score-matching loss for data-driven and noise-driven case on MNIST, compared with the default case

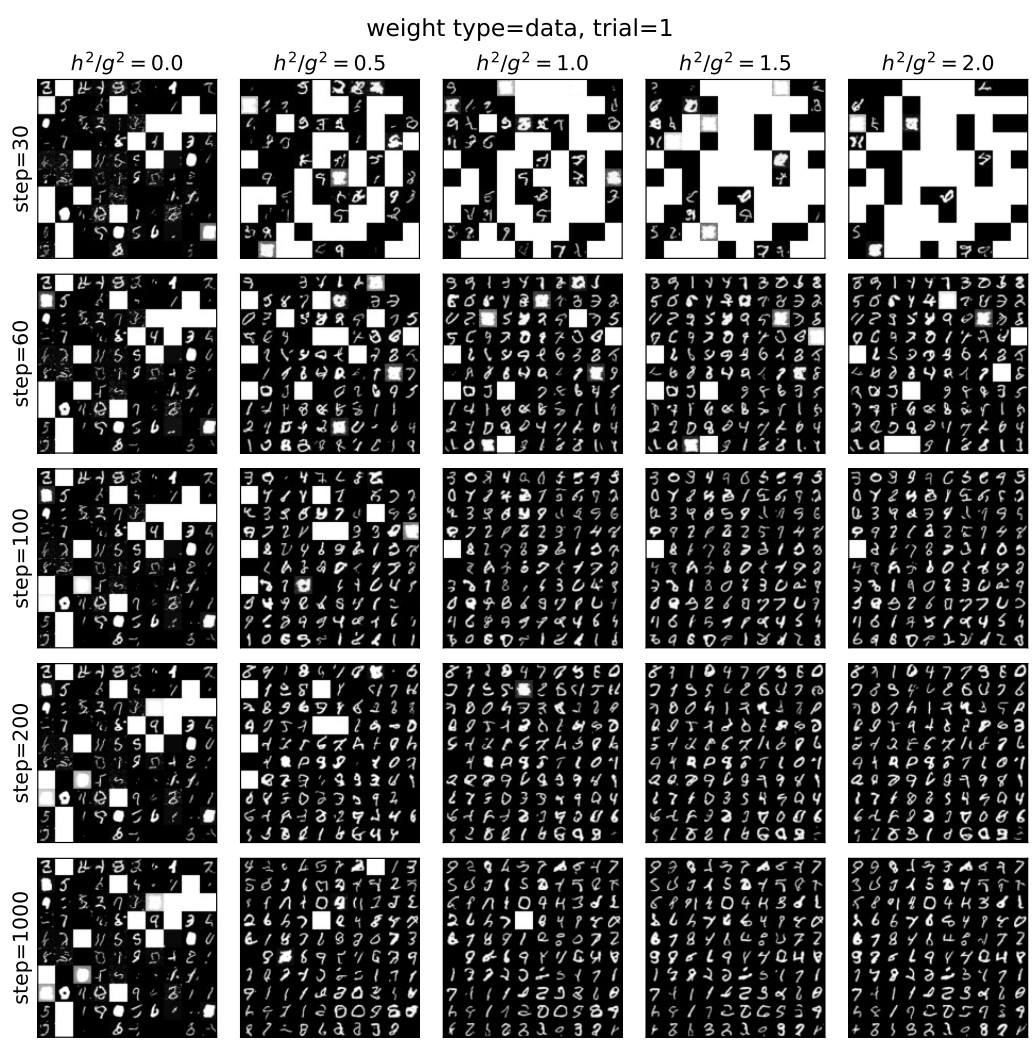

Figure 15: Visualization of sample generation in MNIST where we used **data-driven weight** $\omega$ (45). We use different random seed to ensure robustness and this is the result for trial 1. Results for trial 2 is not included in this work as the overall tendency remains the same.

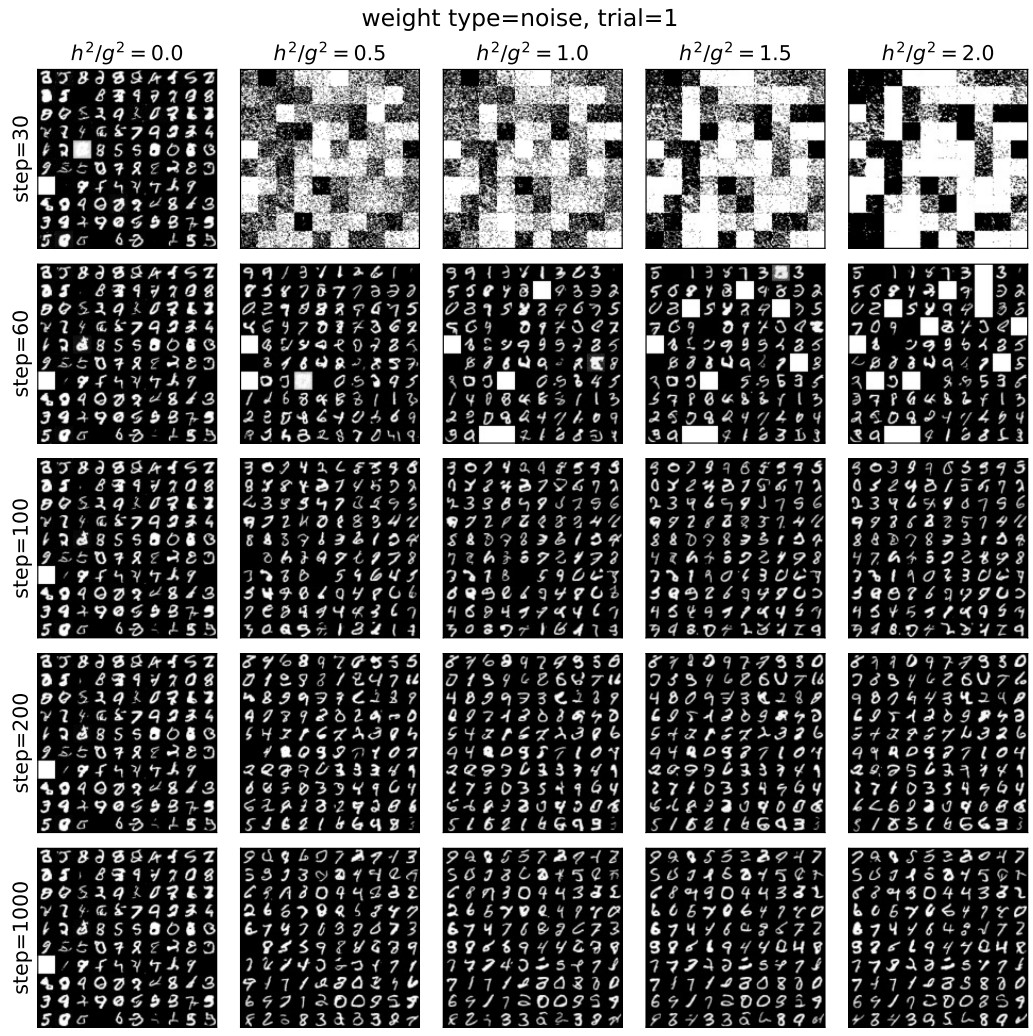

Figure 16: Visualization of sample generation in MNIST where we used **noise-driven weight** $\omega$ (45). We use different random seed to ensure robustness and this is the result for trial 1. Results for trial 2 is not included in this work as the overall tendency remains the same.

### I.3  CIFAR-10

We carry out a similar experiment for CIFAR-10 to test the effect of training weights $\omega_t$: we used the same initialization (referred to as the trial number below) and all other hyper-parameters, except that we employ different weights in score-matching loss (4). The same architecture is used as in [17] for CIFAR-10 and the detailed hyper-parameters can be found in source codes.

In Fig. 17, we observe that overall over the whole training period, the noise-driven weight leads into a score function estimate no worsen than that by the default weight: due to stochastic fluctuations and other uncertainties (in particular if we adopt the mixed precision training), there is no guarantee that the noise-driven one is always better, but the overall tendency is still observable and clear. In Fig. 18, the noise-driven one actually has a slightly larger score-matching loss (SML) than the default one (which could be possibly explained by how we measure the SML in Fig. 18). What is interesting is that score functions trained by the noise-driven weight and the data-driven weight have a similar SML, which both decay at the similar pace; however, the FID values for the score function estimate by noise-driven weight are much smaller than that by the data-driven weight. This apparent gap clearly explains that apart from the total score-matching loss (which does matter), **the time distribution of the score error plays an important role in determining the final sampling error,** echoing our theoretical results in § 3.

For instance, in Fig. 17, if we consider the first experiment (i.e., trial=0) with float16 mixed-precision training (i.e., mixed=True), we notice that relatively near 80k and 120k training iterations, the performance of noise-driven one is much better than the default one, which is consistent with Fig. 19 that the relative loss near $t \approx T$ is more minimised for iterations 80k and 120k, compared with other iteration stages. Moreover, for the same experiment in Fig. 17, the data-driven one has a much worsen FID value at iteration 200k, which is compatible with the increasing relative error near the noise's end (i.e., $t \approx T$) in the last row of Fig. 19. For the remaining three experimental setup (either different initialization or training precision), we notice a similar consistency between how the **time-distribution of the SML behaves** and how **FID values change**. This relation, so far, still cannot be used as a rigorous quantitative indicator to predict one based on the other quantity, but qualitatively, the above explained relationship does appear to be numerically valid and theoretically sound.

**In summary**, if we have two score functions $\mathfrak{S}_1$ and $\mathfrak{S}_2$ from training:

- If the SML for $\mathfrak{S}_1$ is much larger than the SML for $\mathfrak{S}_2$, then we can probably confidently expect that $\mathfrak{S}_2$ is a more accurate estimate.
- However, when the total SML (4) for both are close, then the time-distribution of the score-matching loss together with which generative dynamics is chosen will play a significant role in determining the final sample generation quality, which is probably largely overlooked in current literature as far as we know. A full investigation and in particular whether it is possible to adapt this observation to achieve the state-of-art models will be left to the next stage of research.

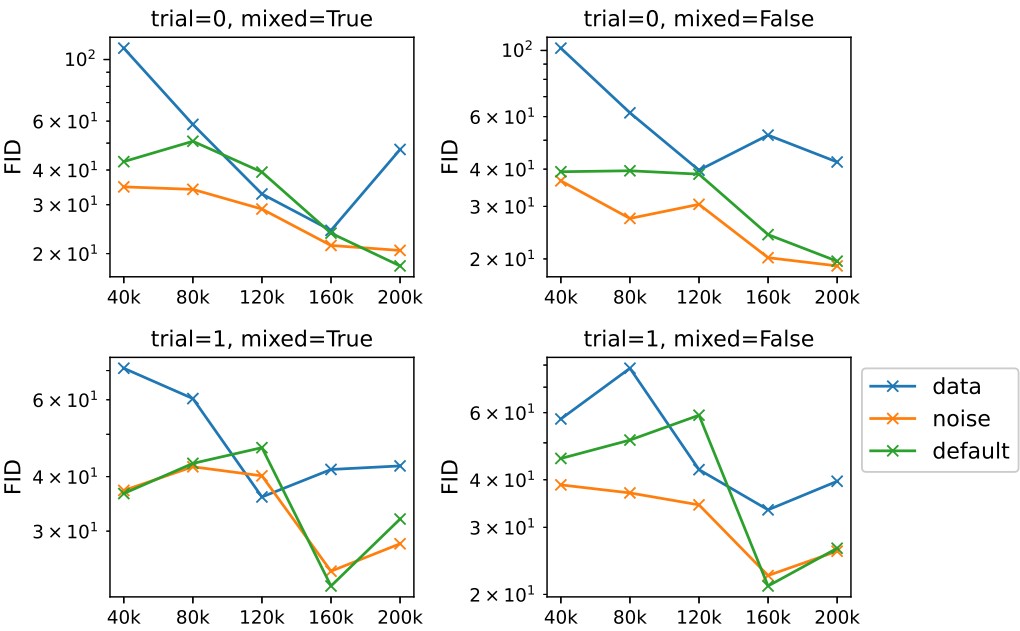

Figure 17: We visualize FIDs of score-training estimates by various weights with respect to the number of training iteration. We consider ODE models (i.e., $h = 0$) when computing FIDs. Trial = 0, 1 refers to different neural network initialization; mixed=True means we use float16 mixed-precision for training; mixed=False means we use 32-bit precision.

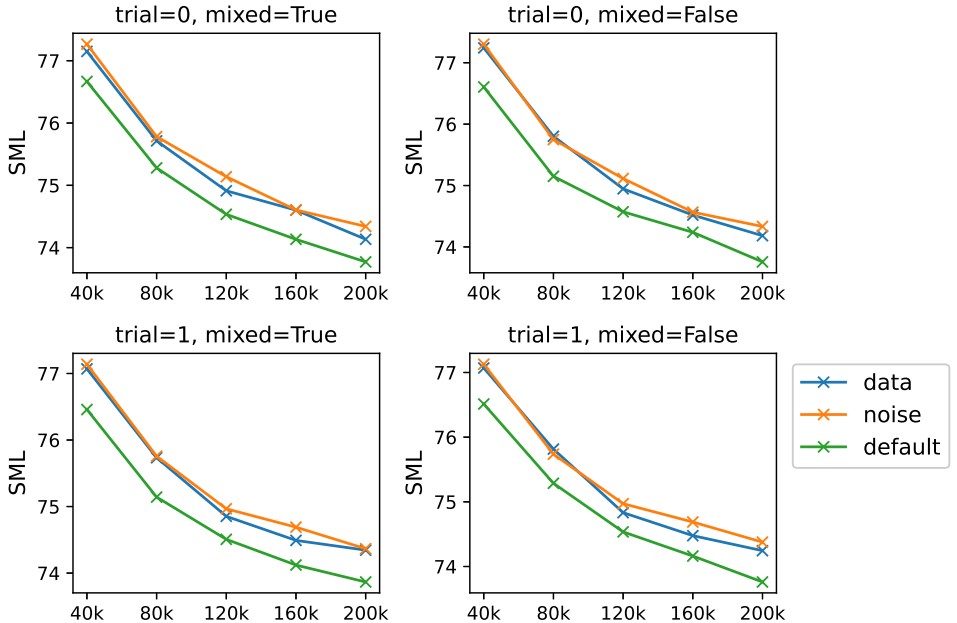

Figure 18: We visualize score-matching loss (4) for score function trained using various weights with respect to the number of training iteration's. Trial = 0, 1 refers to different training initialization; mixed=True means we use float16 mixed-precision for training; mixed=False means we use 32-bit precision. When we compare SML for different score functions above, we consistently use the "default" weight for fairer comparison, even though some score functions are trained using different weight functions; we also use the test dataset to approximately represent $p_0$.

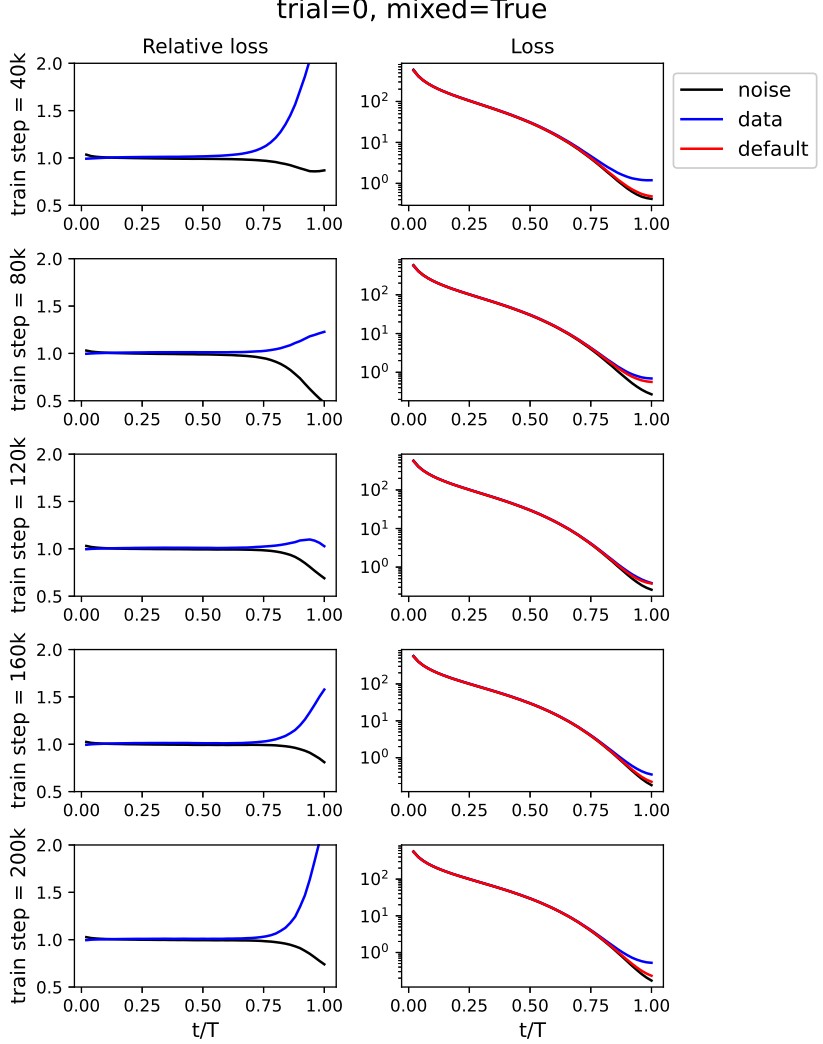

Figure 19: **The first independent experiment with the float16 mixed precision training**: We visualize the relative time distribution of the SML (46) (on the left) and the time distribution of SML (on the right) for various training weights and for various training stages (each row).

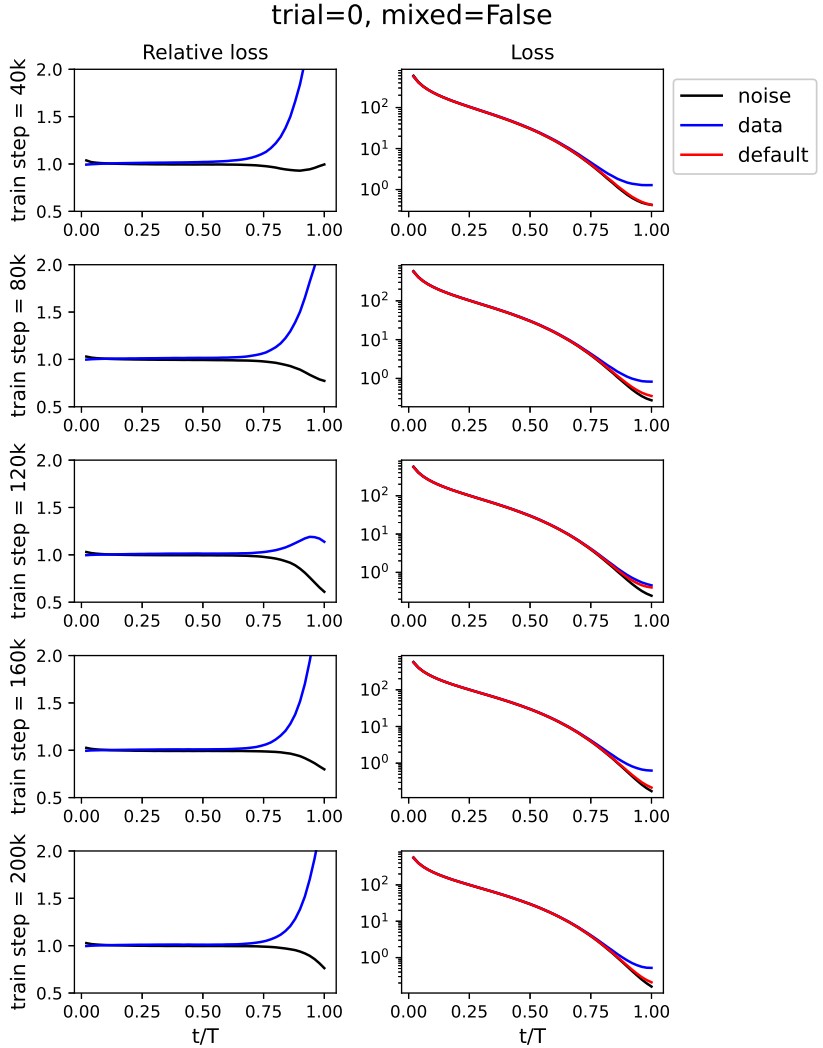

Figure 20: **The first independent experiment with the full 32-bit precision training**: We visualize the relative time distribution of the SML (46) (on the left) and the time distribution of SML (on the right) for various training weights and for various training stages (each row).

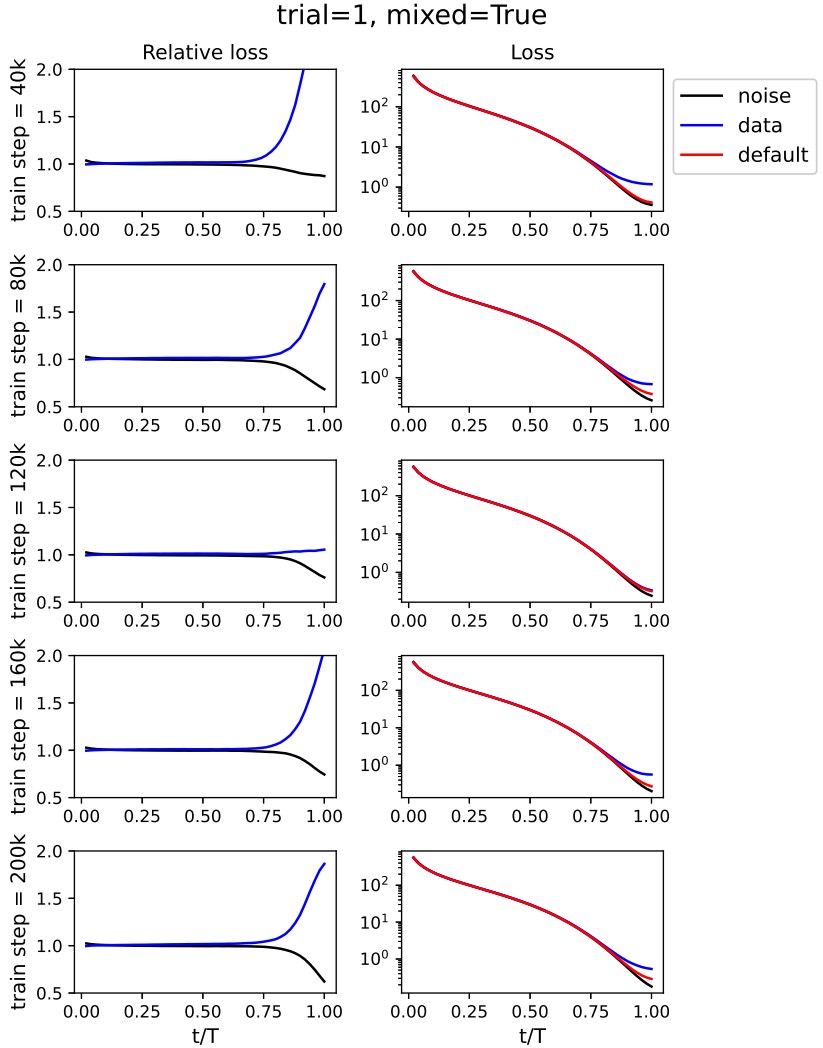

Figure 21: **The second independent experiment with the float16 mixed precision training**: We visualize the relative time distribution of the SML (46) (on the left) and the time distribution of SML (on the right) for various training weights and for various training stages (each row).

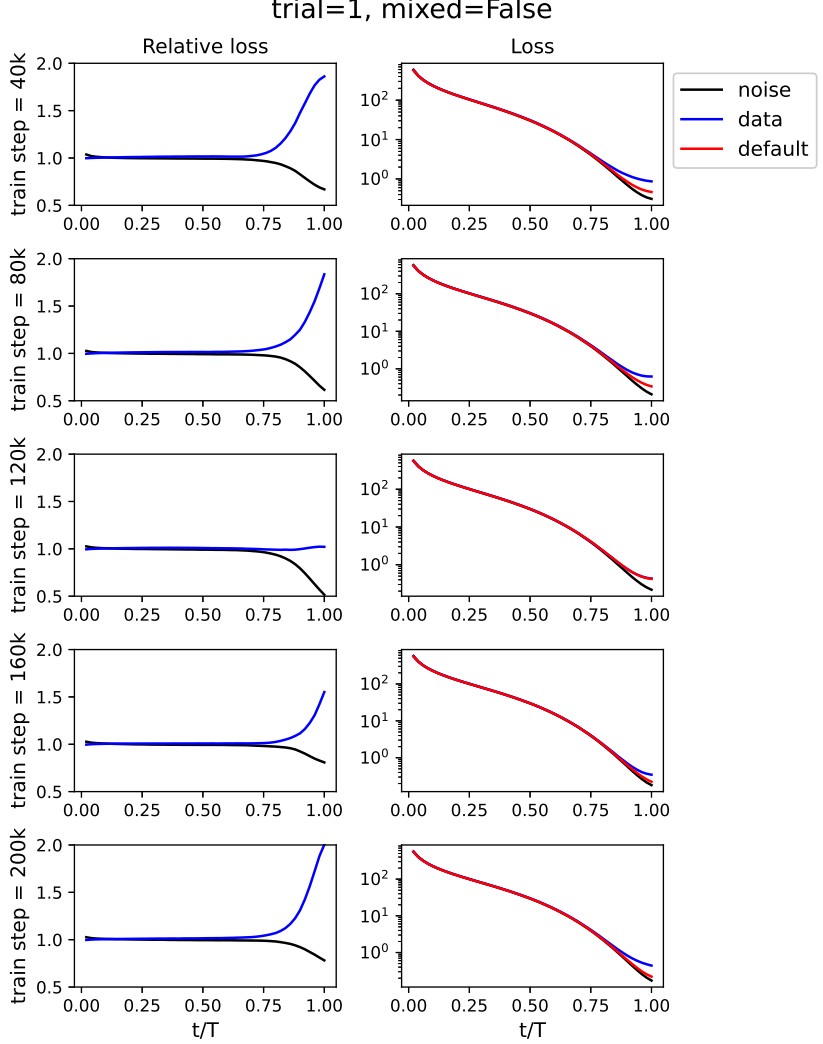

Figure 22: **The second independent experiment with the full 32-bit precision training**: We visualize the relative time distribution of the SML (46) (on the left) and the time distribution of SML (on the right) for various training weights and for various training stages (each row).

