# OpenReview forum: "Exploring the Optimal Choice for Generative Processes in Diffusion Models: Ordinary vs Stochastic Differential Equations"
_NeurIPS.cc/2023/Conference — NeurIPS 2023 poster_

### Official Review · Reviewer_Luhj · 2023-07-02

**Soundness:** 4 excellent
**Presentation:** 2 fair
**Contribution:** 2 fair
**Rating:** 7
**Confidence:** 4

**Summary:**

The paper studies diffusion models, which comprise a class of generative models based on stochastic differential equations. In diffusion models, data is first transformed into Gaussian noise via a stochastic differential equation (usually an Ornstein-Uhlenbeck process), and then a backward process, which turns noise into synthetic data, is learned via the score function. In most works, the diffusion coefficient of the backward process is either chosen to be zero, or to be a constant matching the diffusion term of the forward. The first case is an ODE implementation, while the second case is an SDE corresponding to the exact time reversal of the forward process. However, any function of time can be chosen for the diffusion coefficient of the backward process, and in this work the authors explore the consequences of different choices of diffusion coefficient. In particular, they ask how the optimal choice of diffusion coefficient depends on the error induced by the estimated score function.

The main results of the paper are as follows. Let $h$ denote the diffusion coefficient of the backward process. Assume that the score function estimator has the for $s_t(x)= \nabla \log p_t(x) + \epsilon E_t(x)$, where the first term is the exact score function and the function $E_t(x)$ is a bounded error function. Let $\hat{q}_T$ denote the law of the backward process at time $T$ with $s_t(x)$ in the place of the true score function. Then under certain assumptions on $p_0$, it holds that $$ KL(p_0 \parallel \hat{q}_T = L(h) \epsilon^2 + O(\epsilon^3),$$ where the leading order term $L(h)$ depends on $h$, $p_0$, $E_t(x)$ and $T$. The leading order term $L(h)$ is computed explicitly, which allows the authors to analyze several specific cases when $h$ is constant. In particular, they show that if the error $E_t(x)$ is time-localized (i.e., only nonzero at $t=s$ for some $s > 0$) in the middle of the time interval, the leading order term $L(h)$ decays exponentially fast to zero as $h \rightarrow \infty$. This suggests that the SDE implementation can significantly outperform the ODE implementation in some cases. They also show that for certain forms of the error function, the opposite is true: the ODE implementation has better KL divergence error than the SDE implementation.



--------------
After rebuttal: Thank the authors for the clarifications and comments, which addressed my earlier concerns. I will keep my rating unchanged.

**Strengths:**

Overall, the paper presents a novel and interesting question with some practical implications. The question of choosing the diffusion coefficient in the backward SDE seems largely unstudied because, as the authors mention, the ODE and reverse-time SDE approaches are standard conventions. The fact that they show that different choices of the diffusion coefficient can lead to vastly different performances suggests that the problem is worth studying further.

The authors provide a nice explicit characterization of the leading order term in the KL divergence error, the derivation of which requires a lot of work. Namely, they show that as $h \rightarrow \infty$, the term $L(h)$ converges exponentially fast to a constant $\Tau$ which is upper bounded by $$ \Tau \lesssim \frac{1}{2m_0^2} \int_{\mathbb{R}^d} \frac{(\nabla \cdot (p_0 E_t))^2}{p_0} dx.$$ Here, $m_0$ is the second moment of $p_0$. This suggests the quantity on the right-hand side above plays a significant role in the error analysis of diffusion models, and thus it could be insightful to understand it for different classes of probability distributions.

The authors also leave a lot of room for interesting future work. Most interestingly, the authors pose the question of how to infer the optimal choice of $h$ from data, which could lead to a lot of subsequent research.

**Weaknesses:**

One weakness of the paper is that the upper bound on the constant $\Tau$ is not further explained through discussion or examples. It seems like the upper bound plays a big role in bounding the KL divergence error for diffusion models. It would be very interesting to discuss explicit examples of distributions $p_0$ where the upper bound on $\Tau$ can be explicitly computed or controlled, other than Gaussian distributions. This could be mentioned as a direction for future work. It would also be helpful to describe intuitively what the bound represents about a distribution (what does a large/small $\Tau$ say about a distribution?).

In addition, the numerical experiment on the 2D 4-mode Gaussian mixture could benefit from additional explanation. It doesn’t seem like a machine learning problem, since you already know the score of the distribution from which you are trying to sample. Perhaps the point is simply to illustrate the convergence of $L(h)$ to a constant as $h \rightarrow \infty$, in which case the purpose of the experiment should be stated more clearly.

**Questions:**

1.	Can you explain in what setting, if any, one would ever encounter a score estimator where the error term was localized in time? Or was the purpose of this choice of error term simply to illustrate that different choices of diffusion coefficient can lead to very different performance? It seems like a very strong assumption that would never be realized in practice.
2.	It would be good to state in the main result any dependence on the sub-Gaussian constant $c_U$. As it is stated in the paper, one often works with a mollified version of the empirical distribution, which leads to a tradeoff between bias from smoothing and faster convergence for learning smoother distributions. Thus, rather than treat the constant $c_U$ as hidden, it could be beneficial to discuss how it balances with other terms.
3.	In this paper, only constant choices of the diffusion term $h$ are discussed. Is this just for simplicity? Could it be of interest to consider certain parameterizations of nonconstant functions for $h$? If so, this could be mentioned for future work.

---

> ### Author Rebuttal · Authors · 2023-08-09
>
> Thank you for your appreciation and valuable feedback of our paper.
>
> ### Q1: More understanding about the upper bound.
>
> This suggestion is insightful. We are also interested in exploring the structure of this upper bound. But we are afraid that a simple and heuristic  interpretation of this formula for upper bound is not available. Quantifying the upper bound's structure poses challenges due to the dependence of the score function error, which relies on multiple hyper-parameters and is problem dependent. We've observed some incomplete but interesting aspects regarding this question, and we'll discuss these findings in the revised manuscript.
>
> Consider employing constrained score models, where we parameterize the log probability $\log p_t(x)$ during training, rather than the score function $\nabla\log p_t(x)$. The training process remains unchanged, differing only in how the score function is parameterized via a neural network. In this scenario, the error $\mathcal{E}^\leftarrow_{T} = \nabla\varphi$ for a scalar-valued function $\varphi$ (a result derived from the nature of energy-based models), leading to the following modified upper bound expression:
> $$
> \frac{1}{2m_0^2} \int_{\mathbb{R}^d} (\Delta \varphi - \nabla U_0 \cdot \nabla\varphi)^2 e^{-U_0} = \frac{1}{2m_0^2} \int_{\mathbb{R}^d} (\mathcal{L}^{*}\varphi)^2 e^{-U_0},\qquad p_0 := e^{-U_0}
> $$
>
> where $\mathcal{L}^{*}(\varphi) := \Delta \varphi - \nabla U_0 \cdot \nabla\varphi$. We can readily verify that its adjoint operator $\mathcal{L}(\mu) = \nabla\cdot(\nabla U_0 \mu) + \Delta \mu$ corresponds to the Fokker-Planck generator of the following Langevin dynamics $d X_t = -\nabla U_0(X_t)\ d t + \sqrt{2}\ d W_t$ where $W_t$ is the Brownian motion.
>
> Given that the error $\mathcal{E}^\leftarrow_{T} = \nabla \varphi$ originates from approximating $\nabla\log p_0$, it is logical to consider the scenario where $\varphi = \log p_0$. In this context, the aforementioned upper bound can be expressed as follows:
> $$\mathcal{T} \lesssim \frac{1}{2 m_0^2} \int_{\mathbb{R}^d} \big(|\nabla U_0|^2 - \Delta U_0\big)^2 e^{-U_0}.$$
>
> The above two expressions appear to be slightly more informative than the formula in manuscript.
>
> ### Q2: Gaussian mixture experiment
>
> Thank you for the suggestion. Indeed, the primary use of GMM is for verifying the theory. We will provide clearer clarification of its purpose in the revised version.
>
> ### Q3: About the practicality of pulse shape error.
>
> Thank you for your question. A true pulse shape error might not manifest in real-world scenarios; it is, in essence, a theoretical construct. Nevertheless, we find value in studying it, which also hold practical significance:
> - The time-localized ansatz is a foundational element of our broader findings. It serves as a crucial component in establishing the general result presented in Prop. 3.6, which addresses a generic score error function.
> - The pulse-shape error emerges from analyzing the impact of score estimation errors at each time on the final sample generation error, rather than examining an averaged score-matching loss over time. This concept contributes to a deeper comprehension of the significance of both the overall score-matching loss and the distribution of score function errors across time.
> - To briefly elucidate the origin of the pulse-shape error, we would like to recall Prop. 3.2, where the leading-order term $L$ has a quadratic dependence on $v_T$, which in turn has a linear dependence on $\mathcal{E}^{\leftarrow}_t$. Since $\mathcal{E}^{\leftarrow}_t$ can be decomposed as a linear combination of step functions without loss of generality, it is effectively equivalent to investigating the influence of individual values of $\mathcal{E}^{\leftarrow}_t$ for a fixed $t$ on the final leading-order term $L(h)$—essentially examining a pulse-shape error. We will provide full technical details of this argument in revision.
>
> ### Q4: Scaling of error with respect to $c_U$.
>
> Below are our understandings. We shall discuss three instances where the role of $c_U$ becomes apparent:
> - In Prop. 3.4, $c_U$ explicitly appears in the main result. It influences the lower bound of $\mathsf{h}$ required to ensure exponential decay.
> - In Propositions 3.5 and 3.6, where we address the asymptotic case, the scaling involving $c_U$ becomes harder to track. A lot more effort is required to develop non-asymptotic results and we will defer this to the next stage of research.
> - Another potential context in which $c_U$ could play a role is in determining the upper bound $\mathcal{T}$ as mentioned by the reviewer. Characterizing how $\mathcal{T}$ relates to $p_0$ (and consequently $c_U$) could offer valuable insights. However, one challenge lies in understanding the specific form of the score function error (which is problem-dependent). We acknowledge this intriguing question and plan to explore it further in future research endeavors.
>
> ### Q5: discussion on the constant $h$
>
> The primary reason for the time-independent choice is its simplicity. Furthermore,  the complexity increases significantly for non-asymptotic analysis; when we resort to  asymptotic cases ($h_t = 0$ and $h_t=\infty$), time-dependence becomes less significant. In the case of large diffusion ($h_t \gg 1$), little distinction arises from using time-independent $h_t = \mathsf{h}$. Practicality also guides our choice, as a time-independent $h_t$ is easier to tune: only a single scalar parameter needs adjustment for potential sample generation error reduction. While acknowledging the intrigue of time-dependent cases, we plan to address this in future work, which will be noted and discussed in the revision.

---

> > ### Comment · Area_Chair_wVYj · 2023-08-20
> >
> > Dear reviewer,
> >
> > The author-reviewer discussion period ends in 2 days. Please review the authors' rebuttal and engage with them if you have additional questions or feedback. Your input during the discussion period is valued and helps improve the paper.
> >
> > Thanks, Area Chair

---

### Official Review · Reviewer_pYKm · 2023-07-06

**Soundness:** 4 excellent
**Presentation:** 3 good
**Contribution:** 3 good
**Rating:** 7
**Confidence:** 3

**Summary:**

In this work the authors try to understand the impact of noise in the reverse process of diffusion models in the presence of an approximate score network. In particular, they look at how the Kullback-Leibler divergence between the true data distribution and the denoised distribution evolves w.r.t. the diffusion term $h$ of the backward (generative) process, with an asymptotic expansion of this KL divergence and studying the first non-zero term.
They then show that if the score has a pulse (delta) error early on in the generative sampling, higher values of $h$ are able to recover the density trajectory better, whilst with and error in the late stage of the denoising process it's the opposite.
They empirically back these results on synthetic experiments (With heuristic errors) and on the swiss-roll and MNIST dataset with trained score network.

**Strengths:**

- I believe this paper tackle a really important question, which is the role of noise in the generative process. In particular with the success of flow matching which shows that continuous normalising flows, which are (deterministic) ODE based generative models, can be trained akin to denoising score matching models.
- The analysis seems sounds and rigorous, and the assumptions reasonable.
- The take home message (if I understood correctly) is that apart if the score error is concentrated at the end, the higher the level of noise the better (with infinite computational budget to actually discretise the backward process).
- They link they theoretical findings with empirical results, both on simple settings which is nice to unambiguously be able to conclude, and more realistic settings when the score network is trained.


**Weaknesses:**

- No major weakness, but some of the interpretation of the theoretical results could perhaps be enhanced (cf following questions).
- The fact that higher values of $h$ requires more discretisation steps is something that should be further stressed I believe.


**Questions:**

- Eq 5: missing the score? Or this is the process of the error?
- Section 3.3: typo in title 'Placn'
- Equation 9: This does not assuming $h \gg 0$ right? Worth moving it above in Section 3.2 to avoid any ambiguity perhaps?
- line 194: 'By convection [...]' what does this mean? Doesn't this simply come from the definition of the time-reversal?
- line 196-200: So when $h \rightarrow \infty $ not only $V_t \rightarrow U_t$ but because of the prefactor $h_t^2/2$ in Eq 9 (left) the distribution $\mu_t$ converges extremely fast to to $\mu_0$? Not sure to know what '“almost quasi-static” thermodynamics' refers to. Likely worth expanding a bit more on this in the main paper?
- Can this be seen trough the perspective of Langevin dynamics as a corrector, which would 'project' $\rho_t$ back to $p_t$ at $t>s$?
- line 233: '$L(h)$ will decay to zero exponentially fast' with $h$?
- Section 3.5: Here the setting in that the error accumulates over time in contract with the 'pulse' of Section 3.4. The results of Section 3.5 are that with error at the end of the generative process the smaller $h$ the better, yet if the pulse is also near the end, wouldn't large $h$ help (as the larger $h$ the faster it would correct this er§ror?).
- MNIST: This section is not really self-contained as the weighting functions are in the appendix. What is the motivation for this cubic 'noise' weighting?
- appendix: 'The computational cost of SDE-based models with large diffusion (h > g) is relatively high due to the necessity of a larger number of time steps.' What is the reason for this? That is quite key and worthy of being further developed (in the main text).

**Limitations:**

- Figure 4: It shows that indeed higher values of $h^2/g^2$ can lead to smaller error yet it also requires a lot more step. At $10^3$ steps, which is quite a reasonable number of steps already, the probability flow ODE ($h=0$) perhaps best and the curves starts overlapping around ~2000-3000 steps. So although the theoretical conclusion is that larger values of $h^2/g^2$ is better, for practical implementations there is a trade-off between larger number of discretisation steps (with larger values of $h$) and larger score network architecture.
- The authors look at different types of errors on the score network, both theoretically with pulse-shape error to simplify the analysis, but also experimentally with simple errors (weighting the true score for some time range) which makes it easier to interpret. It would be interesting to look at the empirical error of trained score network (e.g. on image dataset), depending on the weighting $w(t)$ in the loss, but also the scheduling $g(t)$, and in particular its (average) evolution through the denoising time. Is the error spread uniformly (wrt time)? Writing the score as $\epsilon / \sigma(t)$ (with $\sigma(t)$ the marginal variance at time $t$) couldn't one argue that with uniform error in the estimation of noise $epsilon$, the error in the score will blow up near $t=0$ thus the setting where the error is large close to the end of the generative process is the most likely setting (and therefore $h=0$ should work best)? Perhaps this is related with the fact that practitioners stop the denoising process at time $t=\eta$ with $\eta$ a small hyperparameter (e.g. \eta=e-3)?
- Would suggest increasing the fontsize of Figures (e.g. Fig 4)

---

> ### Author Rebuttal · Authors · 2023-08-09
>
> Thank you for taking the time to provide detailed and valuable feedback on our paper. The summary from the reviewer regarding the take-home message is accurate. For a detailed error distribution illustration in score-matching loss over time, see the supplementary PDF. We'll enhance presentation in the revised version based on suggestions. In the following, we address technical queries and concerns.
>
> ### Q1: The fact that higher values of $h$ requires more discretisation steps is something that should be further stressed.
>
> Indeed, if one uses a large $h$ in the generative model, it holds crucial significance for time discretization. We will  give more explicit  statement about this practical limitation for large $h$ in the revised manuscript. We would like to highlight that this work does not say a large $h$ is favored in practice: it discussed both small and large $h$, but under different cases of score error.  Our analysis also indicates under what circumstance, the use of ODE model is preferred, which seems more efficient with less NFE.
>
> ### Q2: Missing of score in Eq. 5
>
> Thanks for pointing out this typo. In the second line below eq (5), ``$\mathfrak{S}^{\leftarrow}_t = \epsilon\mathcal{E}_t^{\leftarrow}$'' should be $\mathfrak{S}^{\leftarrow}_t = \nabla\log p_t^{\leftarrow} + \epsilon\mathcal{E}_t^{\leftarrow}$. We will correct this.
>
> ### Q3: About eq. (9)
>
> We don't assume that $h^{\leftarrow}_t \gg 1$ (we only need $h^{\leftarrow}_t > 0$) in Eq. (9). We will remove $h^{\leftarrow}_t \gg 1$ in the title of Sec. 3.3 to avoid confusion.
>
> ### Q4: In the line 194
>
> We mean "by notational convention". We'll rephrase it to "by the notation of time-reversal".
>
> ### Q5: About Lines 196-200 and quasi-static dynamics
>
> Indeed, rapid convergence is unattainable unless the prefactor $\frac{(h^{\leftarrow}_t)^2}{2}\to\infty$.
> We use "quasi-static" to indicate that the system will remain in proximity to the equilibrium $\rho_t^{\leftarrow}$ at time $t$, regardless of pulse perturbation. We will incorporate the following explanation in the revision: For any distribution ${\mu}_t^{\leftarrow}$ (even quite deviating from $\rho_t^{\leftarrow}$),  over a brief time $\Delta t$ slightly exceeding $\mathcal{O}(1/(h_t^{\leftarrow})^{2})$, we can anticipate that
>
>  $\mu_{t+\Delta t}^{\leftarrow}$ is close to $\rho_{t+\Delta t}^{\leftarrow},$
> assuming the evolution of $\mu_s^{\leftarrow}$ aligns with $\mathcal{L}_{s}^{(h^{\leftarrow})}$.
>
> ### Q6: Relation to the Langevin corrector
>
> This resembles a Langevin corrector: both our findings and the Langevin corrector are blessed by the same convergence property. However, nuanced distinctions exist: In literature, the Langevin corrector addresses errors from discretization schemes, without referring to the score estimation errors, whereas our work deals with diminishing effect from score error.
>
> ### Q7: In line 233
>
> Indeed, it decays exponentially fast with respect to $\mathsf{h}$.
>
> ### Q8: Regarding Sec 3.5
>
> When the pulse occurs near (but not at) the generative process's end, larger $h$ leads to smaller sampling errors (as in Sec 3.4). For instance, an error like $\mathbb{I}_{[T-2\delta, T-\delta]}(t) E(x)$,
> where $\delta>0$ is fixed and $E:\mathbb{R}^d\to\mathbb{R}^d$ still belongs to the case in Sec. 3.4. However, the findings of Sec 3.4 become inapplicable if $\mathcal{E}_T^{\leftarrow} \neq 0$, addressing the subtleties in Sec 3.5.
>
> We isolate the key aspect of $\mathcal{E}^{\leftarrow}$ as the following $\mathbb{I}_{[T-a, T]}(t)E(x)$
> with a small $a \ll 1$. Larger $h$ may not be advantageous here and the tricky part is the complex scaling relationship between $a$ and $\mathsf{h}$. For simplicity, let us say $a = \frac{1}{\mathsf{h}^2}$.
>
> Examining the score function error $\mathcal{E}^\leftarrow_t$ within $t\in [T-a, T]$, the effective time of Langevin dynamics is only $\mathcal{O}\big(a \frac{\mathsf{h}^2}{2}\big) = \mathcal{O}(1)$, insufficient to return to equilibrium and reduce score estimate error. The $\frac{\mathsf{h}^2}{2}$ scaling comes from the prefactor in eq (9). Heuristically, the key distinct is that the SDE, with constant diffusion $\mathsf{h}$, can address error $\mathcal{E}_{t}^\leftarrow$ in the interval $[0,T-\frac{c}{\mathsf{h}^2}]$ (section 3.4), but not in $[T-\frac{c}{\mathsf{h}^2}, T]$ (section 3.5), where $c$ is a constant irrelevant herein.
>
> ### Q9: The cubic training weight
>
> Please refer to our global discussion regarding the relationship between training weight, score error, dynamics selection, and numerical schemes. Our analysis suggests that one might enhance ODE via improving its score training near the noise end, and cubic weight function, increasing monotonically from 0 to 1 as $t\in [0,T]$, reduces data-side loss contribution compared to the quadratic default in literature (i.e., this prioritizes noise-side effects in training loss). For more details, see the supplementary PDF.
>
> ### Q10: Computational cost of large $h$
> The rationale for introducing additional discretization steps stems from the amplified magnitude of both drift and diffusion terms (see Eq. (6)) when using larger $h$.  Enhancing the differential equation's magnitude is akin to prolonging dynamics, inevitably resulting in larger errors in most cases. This necessitates the choice of an improved scheme or a reduction in step size.
>
> ### Q11: Distribution of loss over time and connection to the training weight
>
> Supplementary figures depict score-matching loss over time. We explored loss distribution with varying training weights $\omega_t$. Investigating the impact of $g$ practically remains an interesting future endeavor. Errors are unevenly distributed across time, especially pronounced at the data side under conventional training loss. It appears that the current training weight scheme favors ODE over SDE. Indeed, such a score error distribution over time also connects to the adoption of the early-stopping techniques.

---

> > ### Comment · Reviewer_pYKm · 2023-08-18
> > **response**
> >
> > Thanks for the detailed response and clarifications!

---

### Official Review · Reviewer_GpQ2 · 2023-07-07

**Soundness:** 4 excellent
**Presentation:** 4 excellent
**Contribution:** 3 good
**Rating:** 7
**Confidence:** 4

**Summary:**

The authors focus on reverse diffusion process in the presence of non-negligible error in the score function, and estimate KL divergence between the data distribution and the distribution generated by reverse process. They analyze how the KL divergence varies with the diffusion coefficient and demonstrate that a large diffusion coefficient is beneficial when the error of score function is concentrated near beginning of generation process, while a small diffusion coefficient is beneficial in the opposite case. The findings are validated through numerical experiments on Gaussian mixture model and MNIST data set.

**Strengths:**

The paper introduces a novel analysis of how error accumulation occurs in diffusion models when the error in the score function is not negligible. This is in contrast to conventional analysis that bound error between target and generated distributions when the error in the score function is small.
The analysis is only employed in exploring the diffusion coefficient in reverse generative process. However, the analysis has potential implications for various applications where the correction or modification of score function is important. For instance, more theoretical treatments could be possible on image editing(classifier-free-guidance), fair generation(discriminator-guidance), or so on.

**Weaknesses:**

In Line 178, relation between the assumption and low-dimensionality of data distribution require further explanation to enhance understanding.

**Questions:**

Regarding the scaling hyperparameter in the loss function of conventional diffusion model, which aims to balance the error at different time of the generative process, can the practical choice of the scaling be associated with the theoretical results presented in this work?

**Limitations:**

The authors acknowledge limitation of their work. One additional limitation is lack of large-scale experiments to further validate the findings.

---

> ### Author Rebuttal · Authors · 2023-08-09
>
> Thank you for taking the time to provide valuable feedback on our paper.
>
> ### Q1: Need to clarify low-dimensional data distribution.
>
> We acknowledge that the previous description lacks informative detail, and we aim to provide further elucidation as follows:
> Given that many realistic datasets exhibit effective compact support, it becomes possible to identify a constant denoted as $\widetilde{C}$, such that the distribution follows the inequality:
> $$
> \rho_0(x) \le \widetilde{C} \exp(-|x|^2/2),\qquad \forall x.
> $$
> Consequently, we can deduce:
> $$
> e^{-U_0(x)} \le \widetilde{C} \exp(-|x|^2/2)\ \implies U_0(x) \ge \frac{|x|^2}{2} - \log{\widetilde{C}}.
> $$
> This assertion corresponds to the third condition in Assumption 3.1.
>
> To delve deeper, in situations where $\rho_0$ is predominantly supported within $B_0(R)$, it is feasible to ascertain an adequately large $\widetilde{C}$ such that $\rho_0(x) \le \widetilde{C} e^{-|x|^2/2}$ within $B_0(R)$. Outside of this domain, the aforementioned assumption suggests that the decay surpasses that of a standard Gaussian distribution. In instances where data conforms to a low-dimensional subspace, a common practice is to pump in certain noise for the degenerate coordinate, e.g., approximate the distribution in the degenerate coordinate via $N(0, \sigma^2)$ with small $\sigma$ but NOT infinitesimally small. The above equation can hold true by selecting an appropriate $\widetilde{C}$. We will incorporate these clarifications in our manuscript to enhance accuracy and clarity.
>
> ### Q2: The practical implication of the score-training weight hyperparameter based on our analysis
>
> Our theoretical analysis indicates that for a large diffusion SDE, the sample quality is much less unaffected by score error occurring near the noise end, while the quality  is quite sensitive to score error near the data side ($p_0$). In contrast, for the ODE model,    the conclusion is reversed. This finding immediately suggests a practical hint for enhancing ODE flow about how to train the score function by adjusting the  weight  to favor ODEs:  one would like less error in the noise end and  the  noise-driven  weight (stated in the Appendix of manuscript, which use larger weights near the noise for score training) will likely enhance ODE flows.
>
> Our previous submission clearly demonstrated the effectiveness of this practical strategy  for MNIST (in Fig 5). The supplementary PDF features loss function distributions under various training weight functions, verifying our conjecture. We've also included initial CIFAR-10 results, subject to further exploration and incorporation in the revised version.
>
> ### Q3: about limitations for larger datasets
>
> We have conducted  preliminary assessments on CIFAR-10, and the findings are documented in the supplementary PDF. And our numerical outcomes on CIFAR-10 echo those observed in the MNIST dataset. We will undertake a more comprehensive exploration toward  a broader array of results to be incorporated in  the forthcoming revised manuscript.

---

> > ### Comment · Area_Chair_wVYj · 2023-08-20
> >
> > Dear reviewer,
> >
> > The author-reviewer discussion period ends in 2 days. Please review the authors' rebuttal and engage with them if you have additional questions or feedback. Your input during the discussion period is valued and helps improve the paper.
> >
> > Thanks, Area Chair

---

> ### Comment · Reviewer_GpQ2 · 2023-08-20
>
> I have chosen to maintain my original evaluation.
>
> While I acknowledge the valid concerns by reviewer LAed, I believe this work offers a valuable theoretical contribution. I suggest that the authors include a discussion with reviewer LAed to address their concerns and acknowledge the limitation of this work.
>
> Additional suggestions:
> - I do not think CIFAR10 alone is large scale experiments. Therefore, I strongly urge the authors to include more experiments in the revised version. Even if some of the experimental results do not align perfectly with the theory, they would not diminish the value of this work and could instead serve as limitations that could open up future research directions.
> - To enhance the accessibility of your paper to a wider range of readers, consider adding intuitive figures that visually elucidate the implications of your theorems.

---

### Official Review · Reviewer_LAed · 2023-07-13

**Soundness:** 2 fair
**Presentation:** 2 fair
**Contribution:** 1 poor
**Rating:** 3
**Confidence:** 4

**Summary:**

The paper provides a theoretical analysis of the estimation error of SDE and ODE methods along with some numerical experiments.

**Strengths:**

By perturbing the score function, the authors study how the estimation error changes in ODE and SDE methods.

**Weaknesses:**

It is commonly known that the sample generation error consists of three parts: discretization error, estimation error, and initialization error, see e.g. [1, 2]. It is also commonly known that usually the ODE works better with fewer NFE and SDE works better with more NFE due to the interplay between discretization error and estimation error. The estimation error might not even dominates the total error. I don't think only studying the continuous time model provides very useful insight into how to choose those two methods optimally. There is a significant lack of connection between this theory and what happens in reality. Please comment on this.

[1] Chen et al., Sampling is as easy as learning the score: theory for diffusion models with minimal data assumptions


[2] Lee et al., Convergence of score-based generative modeling for general data distributions

**Questions:**

see weaknesses.

**Limitations:**

yes

---

> ### Author Rebuttal · Authors · 2023-08-09
>
> Thank you for taking the time to provide feedback on our paper.
>
> It's noteworthy that our findings align with certain factual aspects you raised. In the ensuing discussion, we will predominantly focus on two pivotal matters:
> - the significance of refining score function estimation, with the intention to surpass existing conventions;
> - the inherent naturalness and illuminative quality of the continuous-time model within our framework.
>
> ### Clarification of our results.
> We agree with the reviewer's observation that ODE models tend to perform better with fewer function evaluations (NFE), while SDE models excel with higher NFE—a crucial finding. This prompts the question: why does increased NFE potentially boost SDE models over ODE models **whose framework lies in the continuous-time model**, and under what specific conditions? This forms the core of our study.
>
> Importantly, there is no definitive explanation for this phenomenon according to current literature. Our research is fueled by this recognized but mathematically unverified insight. Our objective is to bridge this gap by delving into underlying dynamics, illuminating the conditions that empower SDE models to surpass ODE models with heightened NFE.
>
> ### Score function estimation is important
> It's crucial to emphasize that the estimation error in score training holds comparable, if not greater, significance than the design of an accurate discretization scheme for the inference process. While the reviewer correctly notes that the estimation error may not always dominate the total error, this holds true for numerous scenarios. Notably, situations with limited training data or a preference for lighter-size architectures (e.g., for memory conservation) fall within this category. Score function estimation forms the nucleus of score-based diffusion models, and we believe it's essential to refrain from assuming that training error can be easily minimized or disregarded in most circumstances.
>
> ### The continuous-time model is useful.
> We acknowledge the reviewer's valid point that a continuous-time model does not offer a finalized solution. Nonetheless, despite these acknowledged gaps, the continuous-time model retains its utility:
> -  **Many seminal works are essentially based on continuous-time models.** It's notable that prevailing theoretical studies heavily focus on the continuous-time model or are built upon it. The references highlighted by the reviewer also lean toward continuous-time analysis rather than discrete-time mappings. When we mention the continuous-time model, we encompass both a fully continuous-time representation and a discretized model using a highly accurate numerical scheme. This contrasts with a fully discrete model like DDPM with $10$-NFE or optimal transport maps. Furthermore, we spotlight the seminal work [Score-Based Generative Modeling through Stochastic Differential Equations, ICLR 2021] focusing on continuous-time models. This exemplifies the continued relevance and influence of this framework within the research community.
> - **Continuous-time model is still useful.** If the reviewer acknowledges the significance of comprehending the mentioned wisdom—namely, the interplay between time-discretization error and score estimation error—we believe that a comprehensive response necessitates an understanding of two pedagogical scenarios: (1) absence of score estimate error, addressed in numerous numerical analysis textbooks; (2) absence of discretization error, requiring an exploration of continuous-time models.
> Given the limited analyses exploring score error's impact on the inference process, our study initiates this inquiry as far as we know, and validates it through numerical experiments. It's crucial to clarify that we don't directly extend findings from continuous-time models to real-world scenarios. Nonetheless, these findings hold significance. For instance, our numerical experiments indicate that for ODE models, the score error during initial inference stages significantly affects final sample errors— typically not a big challenge for SDE models (proved in Sec. 3.4). Prioritizing score error refinement in the noise end could potentially enhance sample generation quality. This insight is illustrated with examples like MNIST and supported by preliminary CIFAR-10 experiments (see the additional PDF).
> - **The perception of discretization error being uncontrollable perhaps has evolved.** Efficient numerical schemes for inference have been thoroughly explored through collaborative research efforts. Notable examples encompass DDIM, gDDIM, and several others. The control of numerical discretization error has evolved from a formidable challenge to a manageable one. Our raised question yet still lacks comprehensive systematic exploration, as far as we know.
>
> ### Future direction.
> We have indeed undertaken some efforts to employ analogous techniques utilized in this paper to comprehend a discrete-time model, characterized by a modest number of function evaluations (e.g., around 10-20 NFE).

---

> > ### Comment · Reviewer_LAed · 2023-08-10
> >
> > I understand the authors are trying to justify why studying the continuous time model is necessary. However, the reasons are not convincing.
> >
> > 1. "(1) absence of score estimate error, addressed in numerous numerical analysis textbooks; (2) absence of discretization error, requiring an exploration of continuous-time models"
> >
> > The authors seem to suggest that it is common to assume one error is absent and study another one solely. That might be the case in numerical analysis textbooks, but it is not what people do at conferences like this, as far as I know. I'd be really interested to see some references that solely study one of the three errors: score function estimation error, discretization error, or initialization error. In fact, for example, [1] does not neglect the score function estimation error.
> >
> > 2. "The perception of discretization error being uncontrollable perhaps has evolved"
> >
> > My point was not the discretization error being uncontrollable. What I was trying to say was there was no evidence suggesting the discretization error is negligible, compared to the score function estimation error. For example, let's assume the total error is $E=A+B+C$, where $A$ the is score function estimation error, $B$ is the discretization error, and $C$ is the initialization error. I agree with the authors that $B$ is bounded under certain conditions, but this doesn't mean $B\leq A$. To make this study useful, we need some results like $B\ll A$.
> >
> > 3. I think it might be more appropriate to compare this work with, for example, [2], instead of Song's work at ICLR. In [2], they provide a unified framework for flow models and diffusion models using continuous time stochastic processes. Notably, the framework they proposed is new. Given the high similarities in both techniques and results between those two works, and considering this work is only a small subset of [2], I decided to keep the rating for now. I am glad to have more discussions if there are still misunderstandings or confusion.
> >
> > [1] Chen et al., Sampling is as easy as learning the score: theory for diffusion models with minimal data assumptions, ICLR, 2023.
> >
> > [2] Albergo et al.,  Stochastic Interpolants: A Unifying Framework for Flows and Diffusions, 2023

---

> > > ### Author Response · Authors · 2023-08-15
> > >
> > > ### About comment #1 and #2
> > > These two comments are again about whether a continuous-time model without directly working on the discrete-time setting is still of sufficient scientific contribution or not.  In fact, the work of (Albergo et al., Stochastic Interpolants 2023) that the reviewer just appraised in the feedback is entirely based on the continuous-time setting.
> > >
> > > We believe that it is more beneficial if focusing on the *contributions* of the continuous-time  model, especially when the conclusions are both rigorous in theory and  also backed up by substantial numerical examples, as we did in our original submission and the new supplementary PDF.
> > >
> > > Additionally, considering feedbacks from other reviewers, we maintain that utilizing a continuous-time model isn't inherently a defect in our paper.
> > >
> > > ### About comment #3
> > > Thanks for presenting  this argument that did not appear in the initial review report. We are glad to respond and make clarifications, despite that this comment is a bit confusing for us to comprehend. Please allow us to explain based on our understanding.
> > >
> > > *“Given the high similarities in both techniques and results between those two works, and considering this work is only a small subset of [2]”*
> > >
> > > + “the high similarities in both techniques and results between those two works”.
> > >
> > > We are quite puzzled by why those two works from others (Albergo et al., Stochastic Interpolants 2023) and (Song et al. Score-based generative modeling through SDE. ICLR 2021) matter to our own work here and becomes one of reasons our own work gets low rated.
> > > In our appraisals of these two works, both are excellent and insightful works by utilizing the continuous-time models, and each has its own perspectives toward the generative models: one used the time-reversed SDE and the other directly parametrized a path between two random variables.
> > > + “considering this work is only a small subset of [2]”
> > >
> > > Here [2] refers to (Albergo et al., Stochastic Interpolants 2023) which first appeared at arxiv on 15 Mar 2023 (two months before our submission to NeurIPS).   Our original submission has mentioned this reference on page 2.  Even though these contemporaneous works are in the large field of diffusion and generative models, they clearly focused on different problems and adopted entirely different techniques.  We strongly disagree that our work is only a small subset of [2].
> > >
> > > +  **The problems solved in our work and [2] are different:** Stochastic Interpolants (2023) derives the continuity equations governing a stochastic interpolant by identifying the underlying velocity field. Our problem here is that within the standard score-based diffusion model, how the KL accuracy is impacted by the score error, under different diffusion coefficients.
> > > + **The techniques are different:** Theorem 2.21 in (Stochastic Interpolants 2023) is to directly estimate the KL error by two loss functions for their velocity field and score function.  Our approach works on the first variation (functional derivative) of the KL error w.r.t. the perturbation of score.  The detailed mathematical techniques and skills behind these works are also completely different.
> > >  + **Our main result is not covered by (Stochastic Interpolants 2023):** In stochastic interpolants (2023), even though Section 2.4 shows the upper bound of sampling error in the setting of stochastic interpolant minimizing two vector fields and Section 3.2 has some discussion on deterministic vs stochastic generative models, there is indeed no similar conclusion for the score-based diffusion model as our main results here. Our results indicate that it is not only the loss of the score function matters, but *the distribution (in time) of the score errors matters for the choice of the diffusion coefficient*.

---

> > > ### Comment · Area_Chair_wVYj · 2023-08-18
> > >
> > > 1. Can reviewer LAed please elaborate on why this work is only a small subset of [2], in light of the differences listed in the authors' responses? It would be most helpful if you can provide some examples of "high similarities in both techniques and results".
> > >
> > > 2. As noted by the authors, [2] is posted to arxiv two months before the NeurIPS deadline. I think it is fair to consider these contemporary works. Does reviewer LAed's opinion on the merits of this paper change *if we discount potential similarities to [2]*?
> > >
> > > Thank you for your input!

---

### Official Review · Reviewer_wiU7 · 2023-07-25

**Soundness:** 2 fair
**Presentation:** 2 fair
**Contribution:** 2 fair
**Rating:** 3
**Confidence:** 1

**Summary:**

The paper explores the difference between ODE-based probability flow and SDE-based diffusion models when score training errors are present. Specifically, they investigate how setting the generative diffusion coefficient h impacts sample quality.

**Strengths:**

- The question the authors are trying to answer seems to be a good research question.
- It's nice to see experiments validating some of their assumptions (Fig 1.).


**Weaknesses:**

- As is, the paper is very difficult to follow.
- Many equations are presented, but few of them are explained.
- With so many variables presented in the paper, it is difficult to keep track of all of them, making the paper challenging to comprehend.


**Questions:**

N/A

**Limitations:**

- The authors briefly address potential negative societal impacts of their work.
- I did not see any mention of limitations of their analysis.

---

> ### Author Rebuttal · Authors · 2023-08-09
>
> Thank you for taking the time to provide feedback on our paper. We value your insights regarding
> the presentation of equations and variables. We are dedicated to enhancing the comprehensibility of the
> revised manuscript by incorporating more detailed explanations and motivations. It’s important to note
> that this paper serves a technical purpose, primarily centered around mathematical and asymptotic
> analyses. As such, a substantial number of equations and variables are often unavoidable to precisely
> conveying our intended message, which may require a high level of understanding of mathematical
> analysis. We acknowledge that we have already dedicated Appendix A to define clearly the most
> important variables in the last submitted manuscript, to hopefully facilitate a more streamlined
> tracking of the variables.
>
> Regrettably, we cannot concur with the outlined limitations in the review report:
>  + We have indeed addressed the potential social impact in our paper, as we believe it aligns
> with the submission requirements, particularly when considering the potential enhancements
> to train generative models. As a result, we want to clarify that the negative societal impact
> should not be regarded as a limitation of our study.
> +  We want to emphasize that we have taken explicit steps to address certain potential limitations.
> This includes clearly outlining our assumptions, highlighting the interests of incorporating low-
> dimensional manifold information in the summary section (a step we hadn’t previously taken
> in this work), and explicitly indicating that certain theoretical results hold under asymptotic
> conditions. We believe that these aspects could be construed as limitations in our analysis,
> and we have already presented and discussed limitations.

---

> > ### Comment · Area_Chair_wVYj · 2023-08-20
> >
> > Dear reviewer,
> >
> > The author-reviewer discussion period ends in 2 days. Please review the authors' rebuttal and engage with them if you have additional questions or feedback. Your input during the discussion period is valued and helps improve the paper.
> >
> > Thanks, Area Chair

---

> > ### Comment · Reviewer_wiU7 · 2023-08-21
> >
> > I understand that the nature of the paper requires a large amount of equations and variables and I agree that adding more explanations/motivations will enhance the readability of the paper. I appreciate the authors revisiting this. I thank the authors for pointing out areas in which they addressed the limitations and potential societal impacts of their work.
> >
> > With these, I am comfortable moving my rating to a 5.

---

### Author Rebuttal · Authors · 2023-08-09

We would like to thank all reviewers for comments and suggestions. We are glad by the positive reception of the proposed question, and we hold respect and appreciation for the critical and diverse perspectives provided.

### About practicality and motivation:
A concern or question is about the discretization error (or practicality). It's well-acknowledged that the interplay between score function estimation error and discretization error collectively shapes the ultimate quality of sample generation.
Our overarching perspective encompasses four sequential facets:
1. We choose hyper-parameters for training.
2. Training scheme introduces score estimation error.
3. Score error guides dynamics selection.
4. Subsequently, a numerical scheme needs to be determined.

Rapid advances in inference solvers and diverse algorithmic forms pose many challenges in directly integrating discretization error into optimal dynamics study. Our current focus centers on the interplay between score training error and noise-level within SDE (ODE or large diffusion), i.e., the relation between (2) and (3) above. This is necessary in order to comprehend the whole four facets. Our objective is not to favor specific dynamics but to reveal their relationship and underlying mechanism. This approach may offer a practical avenue to explore optimal dynamics selection under finite NFE and a fixed numerical scheme, which awaits future research pursuits to integrate more factors.

### Explanation of supplementary PDF:
Regarding the supplementary PDF, we highlight the following: our theoretical analysis indicates that large diffusion SDE is unaffected by score error near the noise end (start of inference process) and is sensitive to score error near the data side ($p_0$), with the conclusion reversed for ODEs. This suggests the potential for enhancing ODE flow through training weight adjustment: less error in the noise end (namely, $N(0,I_d)$) is  beneficial for ODEs. Our previous submission demonstrated this for MNIST. Additionally, the supplementary PDF features how score-matching loss function distributions with respect to time under various training weight functions, in line with the above theoretical conjecture. We've also included initial CIFAR-10 results, subject to further exploration and incorporation in the revised version.

---

### Decision · Program_Chairs · 2023-09-21

**Decision:**

Accept (poster)

**Comment:**

This paper studies the effect of diffusion coefficient on the accumulation of score approximation error, and in turn the error of generated samples as measured in KL divergence. There has been many excellent theory papers in the past few years on diffusion generative models, but most of these focus on discretization error, while simply assuming that the score approximation error is bounded by a constant. On the other hand, the error of score-approximation is equally important, but much less understood.

The authors establish a connection between the scale of Gaussian noise in the reverse process, and the final error due to score-approximation. Their analysis shows that SDE can outperform ODE in certain cases, such as when the score approximation is localized to the middle of the generative process. The theory is supported by experiments on some simple synthetic data, as well as on MNIST and CIFAR-10.

There are limitations, for instance, much of the analysis is asymptotic, and  assumptions such as time-localized error are not realistic, and the experiments focuses on relatively simple datasets. Nonetheless, I think the analysis in this paper is an important contribution for understanding score error accumulation, both for theorists, as well as for practitioners in tuning parameters of the reverse process.

Finally, I note that I largely discounted the reviews by wiU7 and LAed; the former because the criticisms raised are largely orthogonal to the quality of the paper; the latter because many excellent papers have indeed focused on analyzing only the discretization error, or the approximation error. For instance, the best existing bounds on discretization error are completely unchanged whether we assume the score error is 0 or bounded by some constant.